# CASCADED FLOW MATCHING FOR HETEROGENEOUS TABULAR DATA WITH MIXED-TYPE FEATURES

## ABSTRACT

Advances in generative modeling have recently been adapted to heterogeneous tabular data. However, generating mixed-type features that combine discrete values with an otherwise continuous distribution remains challenging. We advance the state-of-the-art in diffusion-based generative models for heterogeneous tabular data with a cascaded approach. As such, we conceptualize categorical variables and numerical features as low- and high-resolution representations of a tabular data row. We derive a feature-wise low-resolution representation of numerical features that allows the direct incorporation of mixed-type features including missing values or discrete outcomes with non-zero probability mass. This coarse information is leveraged to guide the high-resolution flow matching model via a novel conditional probability path. We prove that this lowers the transport costs of the flow matching model. The results illustrate that our cascaded pipeline generates more realistic samples and learns the details of distributions more accurately.

## 1 INTRODUCTION

Advancements in the field of generative modeling – rooted in seminal contributions on diffusion models (Sohl-Dickstein et al., 2015; Ho et al., 2020), score-based modeling (Song et al., 2021) and flow matching (Albergo & Vanden-Eijnden, 2023; Lipman et al., 2023; Liu et al., 2023) – have yielded state-of-the-art results across a broad range of complex data modalities. However, progress in adapting these models to the domain of heterogeneous tabular data has remained limited. Given the ubiquity of tabular data in both research and industry – from the social sciences, medicine, to finance in the form of questionnaires, surveys, census data or electronic health records (Borisov et al., 2022; Hernandez et al., 2022; Assefa et al., 2021) – the ability to generate realistic tabular datasets is as crucial as generating images or videos.

Several diffusion-based models for heterogeneous tabular data generation have been introduced (Kim et al., 2023; Kotelnikov et al., 2023; Zhang et al., 2024b; Lee et al., 2023; Mueller et al., 2025; Shi et al., 2025), each with a different solution to the main challenge of integrating numerical and categorical features. However, none of them explicitly accommodates features that combine both categorical and continuous characteristics (Zhao et al., 2021). Such *mixed-type* features are unique to tabular data (Li et al., 2025) and hold significant practical relevance. Prominent examples include censored and inflated features, or numerical features with missing values. Particularly, in cases of informative absence of data, missing values can carry important signals for downstream statistical analysis. As such, a generative model should not merely impute or learn from missing values, but be able to generate them as part of realistic synthetic samples. Thus, the inability of existing approaches to faithfully generate mixed-type features significantly limits their practical utility.

In this paper, we propose TabCascade, a novel cascaded flow matching framework for heterogeneous tabular data with features exhibiting a mixture of categorical and continuous distributions. Within this cascaded framework, numerical details are generated conditional on a coarse-grained representation of the high-fidelity data. Accordingly, we conceptualize categorical variables as low-resolution and numerical features as high-resolution representations of a tabular data row. We explore discretization methods such as distributional regression and Gaussian mixture models to construct a categorical low-resolution approximation of the numerical features. TabCascade first learns the joint distribution of categorical and discretized numerical data as low-resolution information. Subsequently, numerical data is generated conditionally on the low-resolution model's output. This allows TabCascade to focus its ca-

pacity to where it is mostly needed: to generate details, as opposed to coarse categorical data, which we show is relatively easy to learn. We base the high-resolution model on a conditional probability path guided by low-resolution information, thereby introducing a data-dependent coupling that reduces the transport costs between source and target distributions of high-resolution data. Further, we endow it with learnable time schedules conditioned on low-resolution information. Based on some criteria, we choose the categorical part of the CDTD model (Mueller et al., 2025) as our low-resolution component.

The cascaded pipeline gives a natural way of incorporating mixed-type features by letting the model first decide on their categorical part and filling in continuous values only when necessary. Our results show that this benefits the realism of the generated samples substantially and that TabCascade learns the details of the distributions much more accurately than the current state-of-the-art methods.

In sum, we make several contributions towards more efficient and effective models for tabular data:

• To the best of our knowledge, we propose the first diffusion-based model to address mixed-type feature generation, i.e., features that each follow a mixture of categorical and continuous distributions. In practice, this includes inflated, censored, and – most importantly – missing values in numerical features. Our framework naturally extends to any value type that warrants distinct treatment from its continuous counterpart.

• We decompose the tabular data generation task into low- and high-resolution parts. From this, we propose a novel cascaded flow matching framework. We design a guided conditional probability path to model high-resolution data.

• The use of feature-type tailored models sidesteps the challenge of balancing type-specific losses, and thereby prevents the unintended weighting of features during training, prevalent in previous works altogether.

• Accounting for low-resolution information in the generation of numerical details not only boosts sample quality and fidelity but also improves model convergence.

## 2   RELATED WORK

**Diffusion models for tabular data.**    The main challenge for tabular data generation is the effective integration of heterogeneous (i.e., numerical and categorical) features. TabDDPM (Kotelnikov et al., 2023) and CoDi (Lee et al., 2023) combine multinomial diffusion (Hoogeboom et al., 2021) and DDPM (Sohl-Dickstein et al., 2015; Ho et al., 2020); STaSY (Kim et al., 2023) treats one-hot encoded categorical data as numerical; and TabSyn (Zhang et al., 2024b) adopts latent diffusion to embed both feature types into a continuous space. Despite its popularity in other domains, latent diffusion has proven less effective for heterogeneous tabular data compared to models defined directly in data space (Mueller et al., 2025). More recent models, such as TabDiff (Shi et al., 2025) and CDTD (Mueller et al., 2025) learn noise schedules alongside the diffusion model to accommodate the feature heterogeneity in tabular data. These models integrate score matching (Song et al., 2021; Karras et al., 2022) with either masked diffusion (Sahoo et al., 2024) or score interpolation (Dieleman et al., 2022), respectively. While most of these models can be easily adapted to be *trainable* on data containing missing values, in their original state none of them can *generate* missing values in numerical features.

**Exploitation of low-resolution information.**    Cascaded diffusion models (Ho et al., 2022) for super-resolution images define a sequence of diffusion models, where higher resolution models are conditioned on the lower resolution model's outputs. This divide-and-conquer strategy has been successfully used in Google's Imagen model (Saharia et al., 2022) for the generation of high-fidelity images, and can be further refined with data-dependent couplings (Albergo et al., 2024). Instead, Tang et al. (2024) improve sample quality with the combination of a hybrid autoregressive transformer that encodes images into both categorical and continuous tokens. Sahoo et al. (2023) introduce auxiliary latent variables to learn a latent lower resolution structure among images in order to learn pixel-wise conditional noise schedules. This allows the model to adjust the noise in the forward process dependent on low-resolution information of an image. Neural flow diffusion models (Bartosh et al., 2024) generalize this by learning the entire forward process. More generally, Pandey et al. (2022) and Kouzelis et al. (2025) show that combining low-level image details with high-level semantic features improves training efficiency and sample quality. However, the lack of a clear notion of 'resolution' in tabular data makes it difficult to apply the same principle directly.

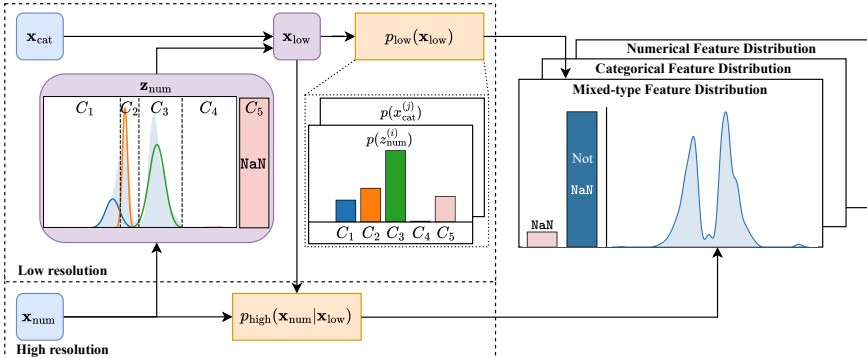

Figure 1: Overview of TabCascade for the task of generating missing values. We first derive a categorical, low-resolution representation $\mathbf{z}_{\text{num}}$ from $\mathbf{x}_{\text{num}}$. We form $\mathbf{x}_{\text{low}} = (\mathbf{x}_{\text{cat}}, \mathbf{z}_{\text{num}})$, and model $\mathbf{x}_{\text{cat}}$ and $\mathbf{z}_{\text{num}}$ jointly with $p_{\text{low}}$. Learning $p_{\text{high}}$, the distribution of high-resolution, numerical features, is simplified by conditioning on $\mathbf{x}_{\text{low}}$, which benefits sample quality. During generation, the explicit availability of the discrete state $\mathbf{z}_{\text{num}}$ enables the model to naturally handle mixed-type feature distributions. This approach generalizes to arbitrary (and multiple) discrete states.

## 3 PROBLEM STATEMENT

**Goal.** Let $\mathcal{D}_{\text{train}} = \{\mathbf{x}_i\}_{i=1}^{N}$ denote a tabular dataset with i.i.d. observations $\mathbf{x} = (\mathbf{x}_{\text{cat}}, \mathbf{x}_{\text{num}})$ drawn from an unknown distribution $p_{\text{data}}(\mathbf{x}_{\text{cat}}, \mathbf{x}_{\text{num}})$. Further, let $\mathbf{x}_{\text{cat}} = (x_{\text{cat}}^{(j)})_{j=1}^{K_{\text{cat}}}$ with $x_{\text{cat}}^{(j)} \in \{0, \dots, C_j\}$ represent the $K_{\text{cat}}$ categorical (including binary) features; and $\mathbf{x}_{\text{num}} \in \mathbb{R}^{K_{\text{num}}}$ the $K_{\text{num}}$ numerical features. The objective is to learn a (parameterized) joint distribution $p^{\boldsymbol{\theta}}(\mathbf{x}_{\text{cat}}, \mathbf{x}_{\text{num}}) \approx p_{\text{data}}(\mathbf{x}_{\text{cat}}, \mathbf{x}_{\text{num}})$ to generate new samples $\mathbf{x}^* = (\mathbf{x}_{\text{cat}}^*, \mathbf{x}_{\text{num}}^*) \sim p^{\boldsymbol{\theta}}(\mathbf{x}_{\text{cat}}, \mathbf{x}_{\text{num}})$ that match the statistical properties of the training data. In practice, $\mathbf{x}_{\text{num}}$ can also be of mixed-type, e.g., a numerical feature including missing values, or a variable following a continuous distribution with point masses at certain outcomes. Such a mixed-type nature differs considerably from the purely continuous distributions typically considered in diffusion-based generative models.

**Inflated values.** Exemplary, let $x_{\text{mixed}}$ be a mixed-type feature with a single inflated value at $v$. Its univariate density is $p(x_{\text{mixed}}) = \pi_v \cdot \delta_v(x_{\text{mixed}}) + (1 - \pi_v) \cdot p_{\text{cont}}(x_{\text{mixed}})$, where $\pi_v$ is the probability mass at $v$, $p_{\text{cont}}$ is a continuous density, and $\delta_v$ is the Dirac delta function centered at $v$. Zero-inflated features ($v = 0$) are common in practice and often carry contextual information: a working time of zero hours in economic survey data may indicate unemployment; in medical data, a drug dosage of zero may indicate the absence of treatment. While existing diffusion models can, in principle, generate such inflated values, they do not explicitly account for this structure. As the distribution becomes more complex, assigning precise probability mass exactly at $v$ becomes increasingly difficult. This setup trivially extends to multiple inflated values, making the discrete distribution categorical instead of binary.

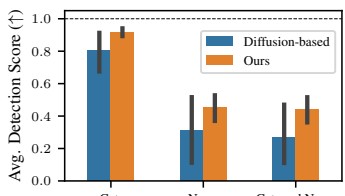

Figure 2: Average detection scores across all datasets and ten sampling seeds (for a single training run) computed on categorical and numerical generated sample subsets.

**Missing values.** Likewise, the discrete state in a mixed-type feature can represent missingness. Let $m = 1$ if feature $x_{\text{mixed}}$ is missing, and $m = 0$ otherwise. Then, the observed data is $x_{\text{mixed}} = (1 - m) \odot x_{\text{num}}^{(\text{latent})} + m \odot \texttt{NaN}$ with a latent variable $x_{\text{num}}^{(\text{latent})}$. Generally, the missingness indicator $m$ may depend on both observed and unobserved parts of the data row. The generative model must therefore also be able to infer $p(m|\mathbf{x}_{\text{num}}, \mathbf{x}_{\text{num}}^{(\text{latent})})$ for all features (Little & Rubin, 2019). This formulation is particularly relevant in domains where missing values carry information: missing answers in psychological questionnaires may point towards certain personality traits; missing values in medical datasets might indicate reluctance to disclose information. Previous diffusion models for tabular data can be *trained* on numerical features with missing values, but are not designed to *generate* such instances.

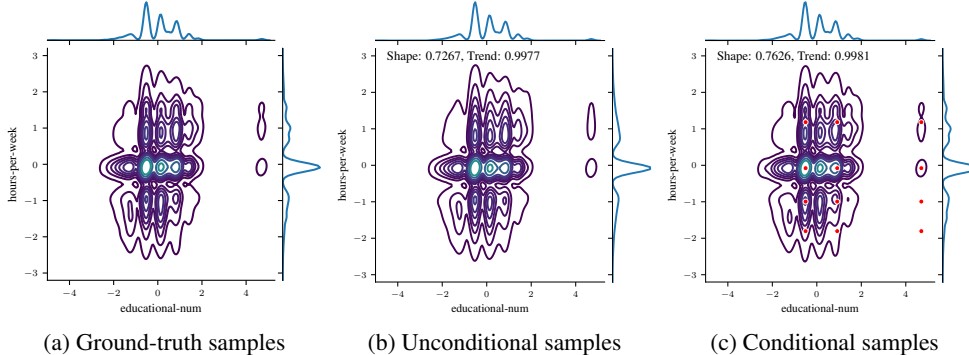

(a) Ground-truth samples  (b) Unconditional samples  (c) Conditional samples

Figure 3: Example from `adult` (hours-per-week, years of education) illustrating the effectiveness of low-resolution conditioning in guiding generation and improving details. (a) Samples from $p_{\text{data}}$. (b) Samples from the unconditional CDTD trained on the two features. (c) Samples from the CDTD conditional on categorical, low-resolution information learned from feature-specific, shallow distributional regression trees. The red dots indicate the means of the possible combinations of components. Shape and Trend metrics are estimated as an average over five sampling seeds.

**The simplicity of learning categorical features.** The premise of existing models for tabular data is to generate $\mathbf{x}_{\text{cat}}$ and $\mathbf{x}_{\text{num}}$ jointly. However, the generation performance is not equal across the two feature types. Empirical evidence in Figure 2 shows that the detection score (averaged over all datasets and diffusion-based models) estimated only on $\mathbf{x}_{\text{cat}}$ exceeds the score obtained for only $\mathbf{x}_{\text{num}}$ substantially. Thus, on average, $\mathbf{x}_{\text{num}}$ is more difficult to learn and accurately generate than $\mathbf{x}_{\text{cat}}$. Figure 16 in the Appendix shows the detailed results per model. This observation motivates the divide-and-conquer approach of our model: first generating the easier component, $\mathbf{x}_{\text{cat}}$, and afterwards the more difficult part $\mathbf{x}_{\text{num}}$ conditional on $\mathbf{x}_{\text{cat}}$ to improve sample quality, as shown in improved detection scores in Figure 2.

**The benefits of conditional generation.** Conditional generation is known to improve sample quality. Unlike images, for which text captions are available as conditioning information, tabular data lacks similar signals. In Figure 3, we investigate the use of distributional trees (Schlosser et al., 2019) to generate a feature-wise clustering of data points which is then used as the conditioning signal in a CDTD model that learns a bivariate distribution. Qualitatively, the conditional model learns the details, i.e., low density areas, of the distribution more accurately. This is also reflected in improved Shape and Trend metrics indicating improved sample quality.

**The pitfall of imbalanced losses.** The heterogeneity of tabular features requires careful alignment of different losses to avoid implicit weighting of feature importance (Ma et al., 2020). For tabular data, Mueller et al. (2025) derive the means of achieving such a balancing from first principles in their CDTD model. Yet, importance parity between $\mathbf{x}_{\text{cat}}$ and $\mathbf{x}_{\text{num}}$ does not necessarily translate into better overall sample quality. For illustration, we train CDTD on the `adult` data using a grid of 14 relative loss weights for categorical features. Figure 4 shows that the detection score can be improved by increasing the relative weight of the categorical losses. In practice, however, models tend to be too large to effectively tune such hyperparameters. Our novel cascaded flow matching model avoids such balancing issues entirely, without requiring any tuning of relative loss weights.

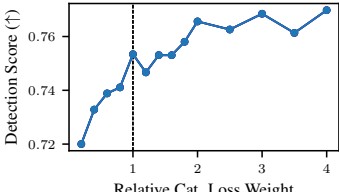

Figure 4: Detection score as a function of the relative loss weight of categorical features (from the `adult` dataset) in CDTD. The vertical line indicates the default.

## 4 CASCADED FLOW MATCHING FOR TABULAR DATA

In the following, we introduce TabCascade, a cascaded flow matching model for heterogeneous tabular data with mixed-type features. An overview is given in Figure 1. First, we outline the general framework and motivate the proposed decomposition into low- and high-resolution information. We lever-

age the low-resolution structure to learn feature-specific probability paths to improve the generation of $\mathbf{x}_{\text{num}}$. In addition to a high-resolution flow matching model, we adopt an efficient low-resolution model and demonstrate how a low-resolution representation of $\mathbf{x}_{\text{num}}$ can be derived in practice.

## 4.1 Cascaded framework

**Tabular data resolution.** In images, resolution refers to the level of visual detail, typically expressed in terms of the total number of pixels. Tabular data lacks a comparable notion of resolution. Building on Figure 2 and the idea that coarse information is easier to learn than details, we link data resolution in tabular datasets to feature types, that is, we treat $\mathbf{x}_{\text{cat}}$ as low-resolution information and $\mathbf{x}_{\text{num}}$ as high-resolution information. We assume that there exists a low-resolution representation of $x_{\text{num}}^{(i)}$ denoted $z_{\text{num}}^{(i)}$. For each data row, $\mathbf{x} = (\mathbf{x}_{\text{cat}}, \mathbf{x}_{\text{num}})$, we construct a low-resolution counterpart, $\mathbf{x}_{\text{low}} = (\mathbf{x}_{\text{cat}}, \mathbf{z}_{\text{num}})$, where $\mathbf{z}_{\text{num}} = [z_{\text{num}}^{(i)}]_{i=1}^{K_{\text{num}}}$ and each $z_{\text{num}}^{(i)}$ is a categorical low-resolution latent representation of $x_{\text{num}}^{(i)}$.

**Cascaded structure.** Accordingly, we define the cascading pipeline (Ho et al., 2022) as a sequence of a low-resolution model followed by a high-resolution model:

$$p(\mathbf{x}_{\text{cat}}, \mathbf{x}_{\text{num}}) = \sum_{\mathbf{z}_{\text{num}} \in \mathcal{Z}} p_{\text{high}}(\mathbf{x}_{\text{num}}|\mathbf{z}_{\text{num}}, \mathbf{x}_{\text{cat}})\, p_{\text{low}}(\mathbf{z}_{\text{num}}, \mathbf{x}_{\text{cat}}). \tag{1}$$

Thus, it resembles a latent variable model, with the latent variable $\mathbf{z}_{\text{num}}$ generated jointly with $\mathbf{x}_{\text{cat}}$. This factorization simplifies learning the joint distribution: The generation of $\mathbf{x}_{\text{cat}}$ is informed by coarse information about $\mathbf{x}_{\text{num}}$, which enables the model to capture dependencies across feature types effectively. Additionally, conditioning on the information in $\mathbf{z}_{\text{num}}$ eases learning $p_{\text{high}}$ and generating $\mathbf{x}_{\text{num}}$. From the chain rule of entropy, we know that $\mathbb{H}(\mathbf{x}_{\text{num}}|\mathbf{z}_{\text{num}}, \mathbf{x}_{\text{cat}}) < \mathbb{H}(\mathbf{x}_{\text{num}}|\mathbf{x}_{\text{cat}})$ if $\mathbf{x}_{\text{num}} \not\perp \mathbf{z}_{\text{num}}$. We therefore aim to infer an informative $\mathbf{z}_{\text{num}}$ such that $p(\mathbf{x}_{\text{num}}|\mathbf{x}_{\text{low}})$ and $p(\mathbf{x}_{\text{low}})$ are easier to learn than the joint distribution $p(\mathbf{x}_{\text{num}}, \mathbf{x}_{\text{cat}})$.

**Mixed-type features.** We use ancestral sampling to sample from $p(\mathbf{x}_{\text{cat}}, \mathbf{x}_{\text{num}})$: we first sample $\mathbf{z}_{\text{num}}, \mathbf{x}_{\text{cat}} \sim p_{\text{low}}^{\boldsymbol{\theta}}(\mathbf{z}_{\text{num}}, \mathbf{x}_{\text{cat}})$, and then $\mathbf{x}_{\text{num}} \sim p_{\text{high}}^{\boldsymbol{\theta}}(\mathbf{x}_{\text{num}}|\mathbf{z}_{\text{num}}, \mathbf{x}_{\text{cat}})$. Since we defined $z_{\text{num}}^{(i)}$ to be categorical, this procedure allows us to directly accommodate mixed-type features. Let $\texttt{NaN}$ and $v_{\text{infl}}$ be the missing and inflated states of $x_{\text{num}}^{(i)}$, respectively. We encode these as separate categories $c_{\text{miss}}$ and $c_{\text{infl}}$ in $z_{\text{num}}^{(i)}$. Thus, we construct

$$x_{\text{num}}^{(i)} = \mathbb{I}(z_{\text{num}}^{(i)} = c_{\text{miss}}) \cdot \texttt{NaN} + \mathbb{I}(z_{\text{num}}^{(i)} = c_{\text{infl}}) \cdot v_{\text{infl}} + \mathbb{I}(z_{\text{num}}^{(i)} \notin \{c_{\text{miss}}, c_{\text{infl}}\}) \cdot \tilde{x}_{\text{num}}^{(i)}, \tag{2}$$

where $\mathbb{I}(\cdot)$ is the indicator function and $\tilde{x}_{\text{num}}^{(i)} = [\tilde{\mathbf{x}}_{\text{num}}]_i$ with $\tilde{\mathbf{x}}_{\text{num}} \sim p_{\text{high}}^{\boldsymbol{\theta}}(\mathbf{x}_{\text{num}}|\mathbf{z}_{\text{num}}, \mathbf{x}_{\text{cat}})$. Once $\mathbf{z}_{\text{num}}$ indicates a category of interest, we can substitute $\tilde{x}_{\text{num}}^{(i)}$ with the inflated or missing state. Intuitively, the model first decides on the coarse structure and only fills in the details when necessary. Therefore, inflatedness and missingness is entirely determined by $p_{\text{low}}^{\boldsymbol{\theta}}$. We can thus mask the corresponding instances when training $p_{\text{high}}^{\boldsymbol{\theta}}$ to free up model capacity. This setup trivially extends to an arbitrary mixed-type structure, for instance, with multiple inflated values.

## 4.2 High-resolution model

We build our model top-down and first introduce the high resolution model $p_{\text{high}}^{\boldsymbol{\theta}}$. For brevity, let $\mathbf{x}_1 = \mathbf{x}_{\text{num}}$ and $\mathbf{z} = \mathbf{z}_{\text{num}}$ and assume $\mathbf{z}$ is observed such that $\mathbf{x}_{\text{low}} = (\mathbf{x}_{\text{cat}}, \mathbf{z})$ and $\mathbf{x}_1, \mathbf{x}_{\text{low}} \sim p_{\text{data}}^*$. We rely on flow matching (Lipman et al., 2023; Albergo & Vanden-Eijnden, 2023; Liu et al., 2023) to learn $p_{\text{high}}^{\boldsymbol{\theta}}$, i.e., we use an ODE $\mathrm{d}\mathbf{x}_t = \mathbf{u}_t(\mathbf{x}_t)\mathrm{d}t$, with a time-dependent vector field $\mathbf{u}_t$ for $t \in [0, 1]$, to transform samples from a source distribution $\mathbf{x}_0 \sim p_0$ to the distribution of interest $\mathbf{x}_1 \sim p_1 = \sum_{\mathbf{x}_{\text{low}} \in \mathcal{X}_{\text{low}}} p_{\text{data}}^*$ via a probability path $p_t$. The goal of flow matching is to learn a vector field $\mathbf{u}_t^{\boldsymbol{\theta}}$ which generates a flow $\Psi_t(\mathbf{x}_0) = \mathbf{x}_t \sim p_t$ such that $\Psi_0(\mathbf{x}_0) = \mathbf{x}_0 \sim p_0$ and $\Psi_1(\mathbf{x}_0) = \mathbf{x}_1 \sim p_1$. Below, we derive a novel *guided* conditional vector field $\mathbf{u}_t(\mathbf{x}_t|\mathbf{x}_1, \mathbf{x}_{\text{low}})$ which uses $\mathbf{x}_{\text{low}}$ to simplify and improve the generation of $\mathbf{x}_1$.

**Guided conditional probability path.** The construction of a suitable ODE requires to design a conditional probability path $p_t(\mathbf{x}_t|\mathbf{x}_1)$. Particularly popular is the linear path, i.e., $\mathbf{x}_t = t\mathbf{x}_1 + (1-t)\mathbf{x}_0$ with $\mathbf{x}_0 \sim \mathcal{N}(\mathbf{0}, \mathbf{I})$. To account for the high feature heterogeneity, we introduce a novel conditional probability path which is guided by feature-specific time schedules and source distributions to exploit our knowledge of $\mathbf{x}_{\text{low}}$.

First, we define a time schedule $\boldsymbol{\gamma}_t(\mathbf{x}_{\text{low}}) : t \to [0, 1]^{K_{\text{num}}}$ which uses $\mathbf{x}_{\text{low}}$ to construct feature-specific non-linear paths of least resistance in $t$. We constrain $\boldsymbol{\gamma}_t(\mathbf{x}_{\text{low}})$ to be monotonically increasing and to satisfy $\boldsymbol{\gamma}_0 = 0$ and $\boldsymbol{\gamma}_1 = 1$. As an efficient parameterization that allows for a closed-form time derivative $\dot{\boldsymbol{\gamma}}_t$, we use a fifth-degree polynomial in $t$ with the parameters provided by a neural network (Sahoo et al., 2023, see Appendix G for details).

Second, we utilize our knowledge of $\mathbf{z}$ to move $\mathbf{x}_0$ closer to the target $\mathbf{x}_1$ with *data-dependent couplings* (Albergo et al., 2024). The coarse information about $\mathbf{x}_1$ in $\mathbf{z}$ determines $\boldsymbol{\mu} \in \mathbb{R}^{K_{\text{num}}}$ and $\boldsymbol{\sigma}(\mathbf{z}) := (\sigma_1(z^{(1)}), \ldots, \sigma_k(z^{(k)}))^{\mathsf{T}}$ with $\sigma_i(z^{(i)}) \in \mathbb{R}_+$ of the feature-specific source distributions such that

$$\mathbf{x}_0 = \boldsymbol{\mu}(\mathbf{z}) + \boldsymbol{\sigma}(\mathbf{z}) \odot \epsilon, \text{ with } \epsilon \sim \mathcal{N}(\mathbf{0}, \mathbf{I}), \tag{3}$$

where $\odot$ indicates element-wise multiplication.

We factorize feature-wise and thus, we re-write the induced coupling as

$$p(\mathbf{x}_0, \mathbf{x}_1) = \sum_{\mathbf{z} \in \mathcal{Z}} p(\mathbf{x}_0|\mathbf{z})p(\mathbf{z}|\mathbf{x}_1)p(\mathbf{x}_1) = \prod_i \sum_{z^{(i)} \in \mathcal{Z}^{(i)}} p(x_0^{(i)}|z^{(i)})p(z^{(i)}|x_1^{(i)})p(\mathbf{x}_1), \tag{4}$$

with $p(x_0^{(i)}|z^{(i)}) = \mathcal{N}(\mu(z^{(i)}), \sigma^2(z^{(i)}))$ similar to a Gaussian component in a mixture model and parameters selected based on $z^{(i)}$. Hence, we first draw $\mathbf{x}_1 \sim p(\mathbf{x}_1)$, retrieve $z^{(i)}$ for each $x_1^{(i)}$ feature-wise, and then sample $x_0^{(i)}$ from the corresponding $p(x_0^{(i)}|z^{(i)})$. Intuitively, we use $z^{(i)}$ to construct a coupling such that each $x_0^{(i)}$ is already located in the proximity of its target $x_1^{(i)}$. These innovations induce a *guided conditional probability path* $p_t(\mathbf{x}_t|\mathbf{x}_1, \mathbf{x}_{\text{low}})$ such that

$$\mathbf{x}_t = \boldsymbol{\gamma}_t(\mathbf{x}_{\text{low}})\mathbf{x}_1 + (1 - \boldsymbol{\gamma}_t(\mathbf{x}_{\text{low}}))[\boldsymbol{\mu}(\mathbf{z}) + \boldsymbol{\sigma}(\mathbf{z}) \odot \epsilon] \sim p_t(\mathbf{x}_t|\mathbf{x}_1, \mathbf{x}_{\text{low}}). \tag{5}$$

This defines the probability path in an augmented space such that the samples take group-conditioned paths, with the groups defined by $\mathbf{x}_{\text{low}}$. Since we impose $\boldsymbol{\gamma}_1 = 1$ and $\boldsymbol{\gamma}_0 = 0$, we obtain $p_0(\mathbf{x}_t|\mathbf{x}_1, \mathbf{x}_{\text{low}}) = p(\mathbf{x}_0|\mathbf{z})$ and $p_1(\mathbf{x}_t|\mathbf{x}_1, \mathbf{x}_{\text{low}}) = \delta_{\mathbf{x}_1}(\mathbf{x}_t)$. Thus, $p_t(\mathbf{x}_t|\mathbf{x}_1, \mathbf{x}_{\text{low}})$ defines a valid conditional probability path.

**Guided conditional vector field.** Our knowledge of $p_t(\mathbf{x}_t|\mathbf{x}_1, \mathbf{x}_{\text{low}})$ allows us to apply Theorem 3 from Lipman et al. (2023) to derive the guided conditional vector field (see Appendix A.1) as

$$\mathbf{u}_t(\mathbf{x}_t|\mathbf{x}_1, \mathbf{x}_{\text{low}}) = \frac{\dot{\boldsymbol{\gamma}}_t(\mathbf{x}_{\text{low}})(\mathbf{x}_1 - \mathbf{x}_t)}{1 - \boldsymbol{\gamma}_t(\mathbf{x}_{\text{low}})}. \tag{6}$$

By substituting Equation (5) in Equation (6) (see Appendix A.1.1), we obtain the target in the conditional flow matching (CFM; Lipman et al., 2023) loss

$$\mathcal{L}_{\text{CFM}} = \mathbb{E}_{t \sim [0,1], (\mathbf{x}_1, \mathbf{x}_{\text{low}}) \sim p^*_{\text{data}}, \epsilon \sim \mathcal{N}(\mathbf{0}, \mathbf{I})} \left[||\mathbf{u}_t^{\boldsymbol{\theta}}(\mathbf{x}_t|\mathbf{x}_{\text{low}}) - \dot{\boldsymbol{\gamma}}_t(\mathbf{x}_{\text{low}})(\mathbf{x}_1 - [\boldsymbol{\mu}(\mathbf{z}) + \boldsymbol{\sigma}(\mathbf{z}) \odot \epsilon])||_2^2\right], \tag{7}$$

with the velocity field $\mathbf{u}_t^{\boldsymbol{\theta}}(\mathbf{x}_t|\mathbf{x}_{\text{low}}) = \dot{\boldsymbol{\gamma}}_t(\mathbf{x}_{\text{low}})F_{\boldsymbol{\theta}}(\mathbf{x}_t, \mathbf{x}_{\text{low}}, t)$ parameterized by a neural network $F_{\boldsymbol{\theta}}$ conditioned on $\mathbf{x}_{\text{low}}$. We mask missing or inflated value entries, as these are inferred from $p_{\text{low}}^{\boldsymbol{\theta}}$ based on Equation (2). Hence, $p_{\text{high}}^{\boldsymbol{\theta}}$ mostly learns feature dependencies and details. Note that, for $\boldsymbol{\gamma}_t = t \cdot \mathbf{1}, \boldsymbol{\mu} = \mathbf{0}$ and $\boldsymbol{\sigma}(\mathbf{x}_{\text{low}}) = \mathbf{1}$, we recover the typical loss from a flow matching model with linear paths. Having trained $\mathbf{u}_t^{\boldsymbol{\theta}}$, we simulate $d\mathbf{x}_t = \mathbf{u}_t^{\boldsymbol{\theta}}(\mathbf{x}_t|\mathbf{x}_{\text{low}})dt$ starting from $\mathbf{x}_0 \sim p(\mathbf{x}_0|\mathbf{z})$ to sample from $p_1$. The cascaded pipeline ensures that $\mathbf{x}_{\text{low}}$ will be available during generation.

### 4.3 LOW-RESOLUTION REPRESENTATION

So far, we have not discussed how we derive $\mathbf{z}$ and how we determine $\boldsymbol{\mu}(\mathbf{z})$ and $\boldsymbol{\sigma}(\mathbf{z})$. First, we note that $z^{(i)}$ must be categorical and only summarizes information about $x_1^{(i)}$. Second, to minimize the noise introduced to the training process of the flow models, we aim to learn feature-specific

encoders $\text{Enc}_i(x_1^{(i)}) \; \forall i$ to output $z^{(i)}$ during data pre-processing. Finally, we want to learn $\mu$ and $\sigma^2$ of $p(x_0^{(i)}|z^{(i)})$ from $z^{(i)}$. Based on these requirements, we propose two different encoders, a Gaussian mixture model (GMM; Bishop, 2006) and a distributional regression tree (DT; Schlosser et al., 2019). For details on the encoders, we refer to Appendix F.

Each model efficiently learns to approximate $p(x_1^{(i)})$ with $K_i$ Gaussian components $p_k(x_1^{(i)}) = \mathcal{N}(\mu_k, \sigma_k^2) \; \forall k \in \{1, \ldots, K_i\}$. For the GMM, we set $z^{(i)} = \arg\max_k \log w_k p_k(x_1^{(i)})$ with mixture weights $w_k$; for the DT, $z^{(i)} = \text{Tree}(x_1^{(i)})$ is the index of the terminal leaf node $x_1^{(i)}$ is allocated to. Our encoder choices allow us to directly use $\mu_k$ and $\sigma_k^2$ to parameterize $p(x_0^{(i)}|z^{(i)})$ in Equation (4) without any additional learning. If $\sigma_k^2 \approx 0$, we treat $\mu_k$ as a inflated value and account for it explicitly. Missing values are removed before fitting the encoder but afterwards added as a separate category $c_{\text{miss}}$ to $z^{(i)}$. Intuitively, we select $p(x_0^{(i)}|z^{(i)})$ to be the Gaussian component that the encoder suggests has most likely generated the data point $x_1^{(i)}$. This moves the source distribution $p(\mathbf{x}_0|\mathbf{z})$ closer to the target distribution $p(\mathbf{x}_1)$, which benefits both training and sampling by reducing the transport cost. We provide a proof below. Compared to, e.g., minibatch Optimal Transport couplings (Tong et al., 2024), our method comes at no additional costs, aside from obtaining $\mathbf{z}$.

**Theorem 1** (Data-dependent coupling lowers transport costs). *Let $\mathbf{z}$ be derived using a DT encoder. Then, our data-dependent coupling (see Equation (4)) yields lower transport costs than an independent coupling.*

*Proof.* See Appendix A.1.2. □

### 4.4 LOW RESOLUTION MODEL

The main requirements for the low-resolution model $p_{\text{low}}^{\boldsymbol{\theta}}$ to learn $p_{\text{low}}$ are that the model generates categorical data efficiently and accurately (and accommodates arbitrary cardinalities). A strength of our framework is that *any generative model for categorical can be used*. For comparative purposes we choose the CDTD model (Mueller et al., 2025), which has been shown to be both efficient and effective at modeling high cardinality features.

## 5 EXPERIMENTS

### 5.1 EXPERIMENTAL SETUP

We evaluate TabCascade across a diverse set of generative models and on multiple popular benchmark datasets. Additionally, we conduct ablation studies to, among others, investigate the value of the individual components of our proposed framework. The implementation details for TabCascade are detailed in Appendix D.

**Baselines.** We benchmark TabCascade against several state-of-the-art generative models for tabular data. These include CTGAN (Xu et al., 2019), TVAE (Xu et al., 2019), the tree-based ARF Watson et al. (2023) as well as the diffusion-based architectures TabDDPM (Kotelnikov et al., 2023), TabSyn (Zhang et al., 2024b), TabDiff (Shi et al., 2025) and CDTD (Mueller et al., 2025).[1] [2] [3] For a fair comparison, we align all models as consistently as possible. Since none of the models natively supports missing data generation, we augment each with a simple encoding-based mechanism for missing value simulation, as described in Section C. Results are aggregated over three training

---

[1] We do not consider ForestDiffusion (Jolicoeur-Martineau et al., 2024) due to its severe lack of efficiency. On `adult`, the default hyperparameters lead to a several hours long training time, which substantially exceeds the training budget of all other diffusion-based models. Similar to Mueller et al. (2025), we therefore deem it prohibitively expensive to include.

[2] We do not benchmark against SMOTE (Chawla et al., 2002) as Mueller et al. (2025) showed their inefficiency and subpar performance for medium to large datasets compared to diffusion-based models. The reason is its reliance on identifying nearest neighbors.

[3] We acknowledge the existence of additional model classes that could be used for the generative task, such as foundational models (e.g., Lin et al., 2025). However, due to our contribution being diffusion-specific, we aim for a fair, comprehensive comparison to *diffusion-based* models.

and ten sampling seeds. The training seeds also affect the missingness simulation. Details on the implementations are provided in Appendix C.

**Evaluation metrics.** We evaluate all models on a broad set of standard metrics for synthetic tabular data (for details, see Appendix E). We consider Shape, Wasserstein distance (WD), Jensen-Shannon divergence (JSD), Trend and detection scores to illustrate the quality of the uni-, bi-variate and joint densities. In addition, we evaluate the performance of the synthetic relative to the real training data on downstream tasks, also known as machine learning efficiency (MLE). Further results on fidelity, coverage and diversity are provided by the $\alpha$-Precision, $\beta$-Recall and DCR share metrics. Since our goal is to approximate the true distribution and provide a fair comparison to existing baselines, we are, similar to the baselines, not concerned with privacy considerations. For completeness, we do provide scores for a membership inference attack (MIA). However, any privacy guarantees would require the adoption of additional techniques, such as differential privacy, in practice. We provide modular code on all evaluation metrics to make future research on tabular data generation easier and more comparable.

**Datasets.** We benchmark on a diverse set of six popular tabular datasets: `adult`, `beijing`, `default`, `diabetes`, `news` and `shoppers` (see also Kotelnikov et al., 2023; Zhang et al., 2024b; Mueller et al., 2025; Shi et al., 2025). The selected datasets include inflated values. The missing values are added (10%) via a simulated MNAR mechanism (Muzellec et al., 2020; Zhao et al., 2023; Zhang et al., 2024a). We utilize the associated regression or classification tasks to evaluate machine learning efficiency for each dataset. For details on the datasets and the simulation, see Appendix B.

## 5.2 RESULTS

Table 1 summarizes all results averaged accross all datasets as well as training and sampling seeds. TabDDPM produced NaNs for the `diabetes` and `news` datasets, in these cases, we assigned it the lowest (i.e., worst) score among the remaining models. Detailed results are given in Appendix J. We provide training and sampling times in Appendix L and the learned time schedules per dataset in Appendix H.

Table 1: Average results across datasets and seeds. The best, row-wise result is indicated in **bold**, the second best is underlined. Shape (num) and Shape (cat) were computed on numerical and categorical features, respectively. Trend (mixed) only considers dependencies across feature types.

| Metric | ARF | TVAE | CTGAN | TabDDPM | TabSyn | TabDiff | CDTD | Ours (DT) |
|---|---|---|---|---|---|---|---|---|
| Detection Score | $0.145_{\pm0.141}$ | $0.059_{\pm0.074}$ | $0.043_{\pm0.035}$ | $0.203_{\pm0.270}$ | $0.110_{\pm0.157}$ | $\underline{0.284}_{\pm0.269}$ | $0.231_{\pm0.219}$ | $\mathbf{0.437}_{\pm0.338}$ |
| Shape | $0.952_{\pm0.029}$ | $0.891_{\pm0.028}$ | $0.911_{\pm0.008}$ | $0.931_{\pm0.055}$ | $0.926_{\pm0.039}$ | $\underline{0.968}_{\pm0.021}$ | $0.962_{\pm0.019}$ | $\mathbf{0.978}_{\pm0.015}$ |
| Shape (cat) | $\mathbf{0.996}_{\pm0.002}$ | $0.889_{\pm0.037}$ | $0.920_{\pm0.035}$ | $0.942_{\pm0.050}$ | $0.948_{\pm0.030}$ | $\underline{0.990}_{\pm0.011}$ | $0.988_{\pm0.004}$ | $0.989_{\pm0.004}$ |
| Shape (num) | $0.921_{\pm0.031}$ | $0.887_{\pm0.033}$ | $0.909_{\pm0.008}$ | $0.924_{\pm0.064}$ | $0.920_{\pm0.045}$ | $\underline{0.959}_{\pm0.027}$ | $0.943_{\pm0.027}$ | $\mathbf{0.974}_{\pm0.020}$ |
| WD (num) | $0.026_{\pm0.026}$ | $0.027_{\pm0.012}$ | $0.024_{\pm0.010}$ | $0.119_{\pm0.268}$ | $0.127_{\pm0.264}$ | $\underline{0.011}_{\pm0.016}$ | $0.017_{\pm0.018}$ | $\mathbf{0.007}_{\pm0.008}$ |
| JSD (cat) | $\mathbf{0.020}_{\pm0.009}$ | $0.174_{\pm0.076}$ | $0.118_{\pm0.031}$ | $0.090_{\pm0.086}$ | $0.075_{\pm0.037}$ | $\underline{0.023}_{\pm0.012}$ | $0.025_{\pm0.009}$ | $0.025_{\pm0.010}$ |
| Trend | $0.961_{\pm0.014}$ | $0.853_{\pm0.073}$ | $0.847_{\pm0.062}$ | $0.915_{\pm0.084}$ | $0.910_{\pm0.048}$ | $\underline{0.969}_{\pm0.018}$ | $0.957_{\pm0.026}$ | $\mathbf{0.969}_{\pm0.019}$ |
| Trend (mixed) | $0.942_{\pm0.020}$ | $0.763_{\pm0.080}$ | $0.739_{\pm0.049}$ | $0.867_{\pm0.117}$ | $0.873_{\pm0.053}$ | $\mathbf{0.954}_{\pm0.021}$ | $0.918_{\pm0.052}$ | $\underline{0.945}_{\pm0.038}$ |
| MLE | $0.056_{\pm0.043}$ | $0.089_{\pm0.100}$ | $0.100_{\pm0.085}$ | $0.577_{\pm1.328}$ | $0.597_{\pm1.318}$ | $\underline{0.034}_{\pm0.029}$ | $0.046_{\pm0.052}$ | $\mathbf{0.025}_{\pm0.019}$ |
| $\alpha$-Precision | $0.953_{\pm0.033}$ | $0.595_{\pm0.319}$ | $0.861_{\pm0.057}$ | $0.661_{\pm0.365}$ | $0.880_{\pm0.140}$ | $0.954_{\pm0.063}$ | $\underline{0.963}_{\pm0.055}$ | $\mathbf{0.972}_{\pm0.032}$ |
| $\beta$-Recall | $0.322_{\pm0.115}$ | $0.209_{\pm0.152}$ | $0.243_{\pm0.089}$ | $0.391_{\pm0.242}$ | $0.261_{\pm0.133}$ | $0.408_{\pm0.085}$ | $\mathbf{0.570}_{\pm0.095}$ | $\underline{0.563}_{\pm0.073}$ |
| DCR Share | $0.800_{\pm0.010}$ | $0.804_{\pm0.013}$ | $\underline{0.781}_{\pm0.003}$ | $0.817_{\pm0.025}$ | $\mathbf{0.779}_{\pm0.002}$ | $0.783_{\pm0.004}$ | $0.858_{\pm0.050}$ | $0.850_{\pm0.051}$ |
| MIA Score | $0.987_{\pm0.008}$ | $0.987_{\pm0.006}$ | $\mathbf{0.991}_{\pm0.002}$ | $0.981_{\pm0.008}$ | $\underline{0.989}_{\pm0.007}$ | $0.987_{\pm0.008}$ | $0.984_{\pm0.011}$ | $0.971_{\pm0.018}$ |

**Substantially improved realism.** The detection score is our main metric of interest since it evaluates the realism of the whole joint distribution of the synthetic data. TabCascade with a DT encoder leads to *substantially* more realistic samples. Figure 2 illustrates the benefit of our cascaded pipeline compared the average of the diffusion-based models.

**Improved or competitive univariate densities.** Metrics reflecting the quality of the univariate densities, i.e., Shape, WD and JSD, indicate that TabCascade's ability to explicitly incorporate mixed-type feature distributions improves the sample quality for numerical features over the baselines. For categorical features, it performs competitively and similar to CDTD, which is caused by our choice of using CDTD for $p_{\text{low}}^{\boldsymbol{\theta}}$. TabCascade achieves this performance despite $p_{\text{low}}^{\boldsymbol{\theta}}$ being actually

much smaller in parameter count compared to the baselines, as we split parameters between $p^{\boldsymbol{\theta}}_{\text{low}}$ and $p^{\boldsymbol{\theta}}_{\text{high}}$. This again supports our motivation that categorical data distributions tend to be easier to learn. We also want to highlight TabCascade yields better results than TabDiff with on average less than 50% of the training costs (see Appendix L). In principle, further performance gains could be realized by using a different model, such as ARF, as $p^{\boldsymbol{\theta}}_{\text{low}}$.

**Accurate dependencies despite cascaded pipeline.**  A cascaded pipeline could make it more difficult for the model to learn dependencies across feature types. However, our introduction of $\mathbf{z}_{\text{num}}$ alleviates this issues. This can be seen in the very competitive Trend scores. Despite a cascaded pipeline, the Trend (mixed) score, which evaluates the bivariate dependencies across feature types only, remains at a very high level. We provide qualitative comparisons of bivariate densities in Appendix I which further illustrate that TabCascade fits the details of distributions more accurately.

**Improved downstream task performance.**  The structure of TabCascade allows a great focus on the details of distributions (see Figure 3), this can benefit the utility of data. The achieved MLE score indicates that the utility of the synthetic data is very high when used as a drop-in replacement for the true data in a downstream task.

**Competitive fidelity, diversity and coverage.**  The greater focus on details naturally translates into greater sample fidelity, as highlighted by the $\alpha$-Precision. The coverage, evaluated by the $\beta$-Recall score, remains competitive to CDTD. Moving samples to more precise areas in the data space comes with the downside of reduced diversity in terms of a lower DCR share. Still, it remains at the level of CDTD, while improving most other metrics considerably.

**The privacy trade-off.**  There is an obvious trade-off between more accurate samples and privacy. To make any privacy guarantees in practice, additional context-dependent mechanisms, like differential privacy, are required. For completeness, we show that privacy, as measured by MIA, remains on a high level but is slightly lower than the baselines.

## 5.3 ABLATION STUDIES

Table 2: Average ablation results over all datasets and seeds. The best, column-wise result is indicated in **bold**, the second best is underlined. Changing from CDTD to a flow matching (FM) high-resolution model implies *independent* coupling and *linear* paths. Grey represents the full TabCascade (DT).

| | Shape | Shape (num) | WD (num) | Trend | Trend (mixed) | Detection Score | MLE | $\alpha$-Precision | $\beta$-Recall | DCR Share | MIA Score |
|---|---|---|---|---|---|---|---|---|---|---|---|
| CDTD | $0.962_{\pm0.019}$ | $0.943_{\pm0.027}$ | $0.017_{\pm0.018}$ | $0.957_{\pm0.026}$ | $0.918_{\pm0.052}$ | $0.231_{\pm0.219}$ | $0.046_{\pm0.052}$ | $0.963_{\pm0.055}$ | $\mathbf{0.570}_{\pm0.095}$ | $0.858_{\pm0.050}$ | $0.984_{\pm0.011}$ |
| + cascade | $0.961_{\pm0.041}$ | $0.945_{\pm0.052}$ | $0.069_{\pm0.154}$ | $0.962_{\pm0.024}$ | $0.931_{\pm0.047}$ | $0.370_{\pm0.299}$ | $0.048_{\pm0.061}$ | $0.939_{\pm0.117}$ | $0.515_{\pm0.205}$ | $0.875_{\pm0.063}$ | $0.978_{\pm0.017}$ |
| + latents $\mathbf{z}_{\text{num}}$ (DT) | $0.917_{\pm0.057}$ | $0.873_{\pm0.096}$ | $0.018_{\pm0.010}$ | $0.882_{\pm0.073}$ | $0.767_{\pm0.105}$ | $0.138_{\pm0.202}$ | $0.074_{\pm0.063}$ | $\underline{0.973}_{\pm0.034}$ | $0.486_{\pm0.116}$ | $\mathbf{0.841}_{\pm0.046}$ | $\mathbf{0.990}_{\pm0.004}$ |
| change to FM | $0.975_{\pm0.016}$ | $0.971_{\pm0.020}$ | $\mathbf{0.007}_{\pm0.007}$ | $0.962_{\pm0.027}$ | $0.935_{\pm0.050}$ | $0.396_{\pm0.296}$ | $\mathbf{0.021}_{\pm0.012}$ | $0.969_{\pm0.037}$ | $0.563_{\pm0.074}$ | $0.851_{\pm0.051}$ | $0.974_{\pm0.016}$ |
| + data dep. coupling | $\underline{0.977}_{\pm0.015}$ | $\underline{0.974}_{\pm0.020}$ | $0.007_{\pm0.009}$ | $\mathbf{0.969}_{\pm0.016}$ | $\mathbf{0.947}_{\pm0.029}$ | $\underline{0.434}_{\pm0.337}$ | $0.024_{\pm0.018}$ | $0.970_{\pm0.034}$ | $0.563_{\pm0.074}$ | $0.851_{\pm0.051}$ | $0.971_{\pm0.018}$ |
| + non-linear paths | $\mathbf{0.978}_{\pm0.015}$ | $\mathbf{0.974}_{\pm0.020}$ | $0.007_{\pm0.008}$ | $0.969_{\pm0.019}$ | $0.945_{\pm0.038}$ | $\mathbf{0.437}_{\pm0.338}$ | $0.025_{\pm0.019}$ | $0.972_{\pm0.032}$ | $\underline{0.563}_{\pm0.073}$ | $0.850_{\pm0.051}$ | $0.971_{\pm0.018}$ |
| switch DT to GMM | $0.961_{\pm0.016}$ | $0.941_{\pm0.015}$ | $0.015_{\pm0.006}$ | $0.957_{\pm0.017}$ | $0.923_{\pm0.025}$ | $0.233_{\pm0.189}$ | $0.029_{\pm0.012}$ | $\mathbf{0.976}_{\pm0.014}$ | $0.535_{\pm0.065}$ | $\underline{0.850}_{\pm0.053}$ | $\underline{0.985}_{\pm0.010}$ |

**Effect of cascaded pipeline and latents.**  First, we investigate the benefit of our main innovations. Table 2 compares the average performances of the vanilla CDTD (Mueller et al., 2025), to a model adds the cascaded pipeline, i.e., specifies $p(\mathbf{x}_{\text{cat}})\,p(\mathbf{x}_{\text{num}}|\mathbf{x}_{\text{cat}})$, and a model that adds $\mathbf{z}_{\text{num}}$ to define $p(\mathbf{x}_{\text{cat}}, \mathbf{z}_{\text{num}})\,p(\mathbf{x}_{\text{num}}|\mathbf{x}_{\text{cat}}, \mathbf{z}_{\text{num}})$, including the relevant masking of losses. All hyperparameters were held constant. The results show that the CDTD model itself can already benefit from the cascaded structure. Adding the latents, however, without the further improvements of TabCascade, leads to a substantial drop in sample quality. This may be caused by CDTD relying on learnable noise schedules that aim for the diffusion losses to develop linearly in time. Adding highly informative signal, like $\mathbf{z}_{\text{num}}$ makes this goal more difficult for the model, such that the learnable noise schedules actually become a hindrance.

**Effect of data dep. coupling and learnable, non-linear paths.**  To reap the benefits of introducing $\mathbf{z}_{\text{num}}$, the TabCascade adds data-dependent coupling and learnable, non-linear paths. Both improve the realism of the univariate and joint densities as well as the dependencies among features over a vanilla flow matching (FM) model with linear paths and independent coupling. The effect of adding

non-linear paths can be interpreted as subtle. However, we want to emphasize that our specification is strictly more flexibly than fixed, linear paths. If it benefits $\mathcal{L}_{\text{CFM}}$, the learnable time schedule can become linear, see Appendix H for illustrations.

**DT vs. GMM encoder.** The DT encoder consistently outperforms the GMM encoder. This is because the DT encoder induces a finer granularity into $\mathbf{z}_{\text{num}}$, i.e., it estimates more Gaussian components. For instance, for the `adult` data, DT on average encodes 65.5 groups, whereas GMM only finds 12.5 on average. In addition, the reduced overlap in the Gaussian components estimated by the DT encoder (see Appendix F) may benefit the generative model by providing more effective clustering of samples.

Table 3: Ablation results for changing the missingness rate averaged over all datasets, sampling seeds and a single training seed. WD cannot be estimated due to too many missings at the very high missingness rate of $p = 0.50$, so we excluded it here. The missingness rate of $p = 0.10$ was used for the main results.

| | TabCascade | | | TabDiff | | |
|---|---|---|---|---|---|---|
| | $p = 0.10$ | $p = 0.25$ | $p = 0.50$ | $p = 0.10$ | $p = 0.25$ | $p = 0.50$ |
| Shape | $0.978_{\pm 0.015}$ | $0.973_{\pm 0.017}$ | $0.966_{\pm 0.024}$ | $0.968_{\pm 0.021}$ | $0.958_{\pm 0.031}$ | $0.950_{\pm 0.048}$ |
| Shape (cat) | $0.989_{\pm 0.004}$ | $0.988_{\pm 0.005}$ | $0.985_{\pm 0.006}$ | $0.990_{\pm 0.010}$ | $0.986_{\pm 0.018}$ | $0.985_{\pm 0.022}$ |
| Shape (num) | $0.975_{\pm 0.020}$ | $0.969_{\pm 0.022}$ | $0.958_{\pm 0.030}$ | $0.959_{\pm 0.027}$ | $0.947_{\pm 0.038}$ | $0.932_{\pm 0.061}$ |
| WD (num) | $0.006_{\pm 0.007}$ | $0.173_{\pm 0.405}$ | - | $0.010_{\pm 0.013}$ | $0.176_{\pm 0.404}$ | - |
| JSD (cat) | $0.025_{\pm 0.010}$ | $0.149_{\pm 0.267}$ | $0.248_{\pm 0.272}$ | $0.024_{\pm 0.011}$ | $0.155_{\pm 0.264}$ | $0.249_{\pm 0.271}$ |
| Trend | $0.969_{\pm 0.017}$ | $0.966_{\pm 0.018}$ | $0.966_{\pm 0.016}$ | $0.969_{\pm 0.018}$ | $0.961_{\pm 0.026}$ | $0.961_{\pm 0.034}$ |
| Trend (mixed) | $0.943_{\pm 0.037}$ | $0.942_{\pm 0.035}$ | $0.948_{\pm 0.030}$ | $0.954_{\pm 0.024}$ | $0.948_{\pm 0.029}$ | $0.952_{\pm 0.036}$ |
| Detection Score | $0.445_{\pm 0.349}$ | $0.454_{\pm 0.324}$ | $0.505_{\pm 0.301}$ | $0.281_{\pm 0.264}$ | $0.267_{\pm 0.227}$ | $0.352_{\pm 0.270}$ |
| MLE | $0.017_{\pm 0.013}$ | $0.021_{\pm 0.013}$ | $0.023_{\pm 0.016}$ | $0.034_{\pm 0.028}$ | $0.039_{\pm 0.031}$ | $0.035_{\pm 0.027}$ |
| $\alpha$-Precision | $0.972_{\pm 0.035}$ | $0.956_{\pm 0.079}$ | $0.950_{\pm 0.056}$ | $0.955_{\pm 0.058}$ | $0.936_{\pm 0.106}$ | $0.920_{\pm 0.164}$ |
| $\beta$-Recall | $0.562_{\pm 0.077}$ | $0.572_{\pm 0.089}$ | $0.599_{\pm 0.069}$ | $0.408_{\pm 0.085}$ | $0.429_{\pm 0.098}$ | $0.419_{\pm 0.110}$ |
| DCR Share | $0.849_{\pm 0.050}$ | $0.852_{\pm 0.051}$ | $0.868_{\pm 0.057}$ | $0.784_{\pm 0.004}$ | $0.782_{\pm 0.004}$ | $0.786_{\pm 0.005}$ |
| MIA Score | $0.971_{\pm 0.019}$ | $0.972_{\pm 0.017}$ | $0.969_{\pm 0.017}$ | $0.986_{\pm 0.007}$ | $0.987_{\pm 0.005}$ | $0.982_{\pm 0.011}$ |

**Effect of missingness rate.** To investigate the effect of increasing the rate of simulated missings from the default of $p = 0.10$ to $p = 0.25$ and $p = 0.50$, we compare TabCascade to TabDiff, which performs well overall in the main results. The results in Table 3 confirm the general pattern seen in Table 1. The relative performance gain of using TabCascade over TabDiff stays consistent as we increase $p$. In particular, most metrics barely worsen, despite the significant increase in missings in the training data.

**Additional ablation on the DT encoder depth.** We provide ablation results for the effect of the maximum depth of DT encoder in Appendix K. This includes an analysis of the effect of an increase in encoder complexity on the proportion of masked inputs to $p_{\text{high}}^{\boldsymbol{\theta}}$.

## 6 CONCLUSION

In this paper, we introduced TabCascade, a cascaded flow matching model that generates high-resolution, numerical features based on their low-resolution encoding and categorical features. The model builds on a novel conditional probability path guided by low-resolution information and combines it with feature-specific, learnable time schedules that enable non-linear paths. This framework allows the direct accommodation of mixed-type features and provably lowers the transport costs. The extensive experiments we conducted demonstrate TabCascade's enhanced ability to generate realistic samples and learn the details of the distribution. However, we emphasize that our benchmarks are mostly limited to *diffusion-based* models. Other model classes, such as foundational models (e.g., Lin et al., 2025) or SMOTE (Chawla et al., 2002) are not included. Also, due to computational constraints, we only consider a set of six very popular benchmark datasets in the field. Hence, we suggest a more extensive comparison including additional model classes and additional, possibly more difficult to model datasets, e.g., based on those proposed by (McElfresh et al., 2024)

for regression / classification tasks, as future work. The latter may become even more important in the future, since we already see near-ceiling performance of most models on some of our metrics. We also leave questions about how to generalize the cascaded framework to other data modalities, how to adopt it for data imputation, and how to integrate privacy guarantees for future work. To further improve sample quality, our cascaded framework could also be combined with an autoregressive low-resolution model. Lastly, for efficiency gains, the number of parameters in the high-resolution model could be optimized depending on the number of numerical features and the proportion of their masked entries.

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

# A APPENDIX

## A.1 DERIVATION OF THE GUIDED CONDITIONAL VECTOR FIELD FOR THE HIGH-RESOLUTION MODEL

Theorem 3 in Lipman et al. (2023) proves that if the Gaussian conditional probability path is of the form $p_t(\mathbf{x}_t|\mathbf{x}_1) = \mathcal{N}(\boldsymbol{\mu}_t(\mathbf{x}_1), \sigma_t^2(\mathbf{x}_1)\mathbf{I})$ then the unique vector field that generates the flow $\Psi_t$ has the form:

$$\mathbf{u}_t(\mathbf{x}_t|\mathbf{x}_1) = \frac{\dot{\sigma}_t(\mathbf{x}_1)}{\sigma_t(\mathbf{x}_1)}(\mathbf{x}_t - \boldsymbol{\mu}_t(\mathbf{x}_1)) + \dot{\boldsymbol{\mu}}_t(\mathbf{x}_1). \tag{8}$$

In Equation (5), we implicitly define the guided conditional probability path as

$$\mathbf{x}_t = \boldsymbol{\gamma}_t(\mathbf{x}_{\text{low}})\mathbf{x}_1 + (1 - \boldsymbol{\gamma}_t(\mathbf{x}_{\text{low}}))[\boldsymbol{\mu}(\mathbf{z}) + \boldsymbol{\sigma}(\mathbf{z}) \odot \epsilon].$$

This induces the probability path

$$p_t(\mathbf{x}_t|\mathbf{x}_1, \mathbf{x}_{\text{low}}) = \mathcal{N}(\boldsymbol{\mu}_t(\mathbf{x}_1, \mathbf{x}_{\text{low}}), \text{diag}(\boldsymbol{\sigma}_t^2(\mathbf{x}_1, \mathbf{x}_{\text{low}}))), \tag{9}$$

with

$$\boldsymbol{\mu}_t(\mathbf{x}_1, \mathbf{x}_{\text{low}}) = \boldsymbol{\gamma}_t(\mathbf{x}_{\text{low}})\mathbf{x}_1 + (1 - \boldsymbol{\gamma}_t(\mathbf{x}_{\text{low}}))\boldsymbol{\mu}(\mathbf{z}), \tag{10}$$

and

$$\boldsymbol{\sigma}_t(\mathbf{x}_1, \mathbf{x}_{\text{low}}) = (1 - \boldsymbol{\gamma}_t(\mathbf{x}_{\text{low}}))\boldsymbol{\sigma}(\mathbf{x}_{\text{low}}), \tag{11}$$

since $\mathbf{x}_1$ and $\mathbf{x}_{\text{low}}$ are fixed.

With some abuse of notation, we let $\sigma_t$ and $\dot{\sigma}_t$ be vectors and interpret any multiplication and division operations element-wise. This is valid, since Theorem 3 would also apply to each element in $\mathbf{x}_t$ separately, and we are specifying a diagonal covariance matrix. The time-derivatives are given by

$$\dot{\boldsymbol{\mu}}_t(\mathbf{x}_1, \mathbf{x}_{\text{low}}) = \dot{\boldsymbol{\gamma}}_t(\mathbf{x}_{\text{low}})(\mathbf{x}_1 - \boldsymbol{\mu}(\mathbf{z})) \text{ and } \dot{\boldsymbol{\sigma}}_t(\mathbf{x}_1, \mathbf{x}_{\text{low}}) = -\dot{\boldsymbol{\gamma}}_t(\mathbf{x}_{\text{low}})\boldsymbol{\sigma}(\mathbf{z}). \tag{12}$$

Plugging into Theorem 3 and (for brevity) omitting the dependence of $\boldsymbol{\gamma}_t$, $\boldsymbol{\mu}$ and $\boldsymbol{\sigma}$ on $\mathbf{x}_{\text{low}}$ (and therefore also $\mathbf{z}$), we derive the conditional vector field as

$$\begin{aligned}
\mathbf{u}_t(\mathbf{x}_t|\mathbf{x}_1, \mathbf{x}_{\text{low}}) &= \frac{-\dot{\boldsymbol{\gamma}}_t\boldsymbol{\sigma}}{(1 - \boldsymbol{\gamma}_t)\boldsymbol{\sigma}}(\mathbf{x}_t - [\boldsymbol{\gamma}_t\mathbf{x}_1 + (1 - \boldsymbol{\gamma}_t)\boldsymbol{\mu}]) + \dot{\boldsymbol{\gamma}}_t(\mathbf{x}_1 - \boldsymbol{\mu}) \\
&= \frac{-\dot{\boldsymbol{\gamma}}_t}{1 - \boldsymbol{\gamma}_t}(\mathbf{x}_t - \boldsymbol{\gamma}_t\mathbf{x}_1 - (1 - \boldsymbol{\gamma}_t)\boldsymbol{\mu} - (1 - \boldsymbol{\gamma}_t)\mathbf{x}_1 + (1 - \boldsymbol{\gamma}_t)\boldsymbol{\mu}) \\
&= \frac{\dot{\boldsymbol{\gamma}}_t(\mathbf{x}_1 - \mathbf{x}_t)}{1 - \boldsymbol{\gamma}_t}.
\end{aligned}$$

### A.1.1 DERIVATION OF THE TRAINING TARGET FOR THE HIGH-RESOLUTION MODEL

To derive the training target, We plug Equation (5) into Equation (6) to get

$$\begin{aligned}
\mathbf{u}_t(\mathbf{x}_t|\mathbf{x}_1, \mathbf{x}_{\text{low}}) &= \frac{\dot{\boldsymbol{\gamma}}_t(\mathbf{x}_{\text{low}})(\mathbf{x}_1 - \mathbf{x}_t)}{1 - \boldsymbol{\gamma}_t(\mathbf{x}_{\text{low}})} \\
&= \frac{\dot{\boldsymbol{\gamma}}_t(\mathbf{x}_{\text{low}})}{1 - \boldsymbol{\gamma}_t(\mathbf{x}_{\text{low}})}\left((1 - \boldsymbol{\gamma}_t(\mathbf{x}_{\text{low}}))\mathbf{x}_1 - (1 - \boldsymbol{\gamma}_t(\mathbf{x}_{\text{low}}))[\boldsymbol{\mu}(\mathbf{z}) + \boldsymbol{\sigma}(\mathbf{z}) \odot \epsilon]\right) \\
&= \dot{\boldsymbol{\gamma}}_t(\mathbf{x}_{\text{low}})\left(\mathbf{x}_1 - [\boldsymbol{\mu}(\mathbf{z}) + \boldsymbol{\sigma}(\mathbf{z}) \odot \epsilon]\right),
\end{aligned}$$

which is the scaled difference between ground-truth sample $\mathbf{x}_1$ and source sample $\mathbf{x}_0$ from our data-dependent source distribution.

### A.1.2 PROOF: DT ENCODER LOWERS TRANSPORT COST BOUND

Proposition 3.1 by Albergo et al. (2024) shows that for a probability flow defined as

$$\Psi_t(\mathbf{x}_0) = \alpha_t\mathbf{x}_1 + \beta_t\mathbf{x}_0 \in \mathbb{R}^{K_{\text{num}}},$$

such that $\Psi_0(\mathbf{x}_0) = \mathbf{x}_0 \sim p_0$ and $\Psi_1(\mathbf{x}_0) = \mathbf{x}_1 \sim p_1$, the transport costs are upper bound by

$$\mathbb{E}_{\mathbf{x}_0 \sim p_0}\left[||\Psi_1(\mathbf{x}_0) - \mathbf{x}_0||^2\right] \leq \int_0^1 \mathbb{E}[||\dot{\Psi}_t||^2]\mathrm{d}t < \infty. \tag{13}$$

Minimizing the left-hand side implies finding the optimal transport plan as defined by Benamou & Brenier (2000), corresponding to the minimum Wasserstein-2 distance between $p_0$ and $p_1$. However, to show that a specific coupling $p^*(\mathbf{x}_0, \mathbf{x}_1)$ induces less transport costs, it suffices to show that $\mathbb{E}[||\dot{\Psi}_t||^2]$ is smaller under the new coupling than the original independent coupling $p(\mathbf{x}_0, \mathbf{x}_1)$.

Below, we show that our proposed data-dependent coupling leads to provable lower transport costs when using a distributional tree as the encoder.

Our high resolution model defines $\Psi_t(\mathbf{x}_0) = \boldsymbol{\gamma}_t \mathbf{x}_1 + (1 - \boldsymbol{\gamma}_t)\mathbf{x}_0$ such that $\dot{\Psi}_t = \dot{\boldsymbol{\gamma}}_t(\mathbf{x}_1 - \mathbf{x}_0)$.

We need to show that

$$\int_{\mathbb{R}^{2d}} ||\dot{\Psi}_t||^2 p(\mathbf{x}_0, \mathbf{x}_1)\mathrm{d}\mathbf{x}_0 \mathrm{d}\mathbf{x}_1 \leq \int_{\mathbb{R}^{2d}} ||\dot{\Psi}_t||^2 p(\mathbf{x}_0)p(\mathbf{x}_1)\mathrm{d}\mathbf{x}_0 \mathrm{d}\mathbf{x}_1,$$

where $p(\mathbf{x}_0, \mathbf{x}_1)$ is our data-dependent coupling from Equation (4) where $\mathbf{z}$ is derived by the DT encoder.

First, for the independent coupling, i.e., $p(\mathbf{x}_0, \mathbf{x}_1) = p(\mathbf{x}_0)p(\mathbf{x}_1)$, the expectation is taken over $\mathbf{x}_0 \sim p(\mathbf{x}_0) = \mathcal{N}(\mathbf{0}, \mathbf{I})$ and $\mathbf{x}_1 \sim p_1$ such that

$$\mathbb{E}[||\dot{\Psi}_t||^2] = \mathbb{E}[||\dot{\boldsymbol{\gamma}}_t(\mathbf{x}_1 - \mathbf{x}_0)||^2]$$
$$= \dot{\boldsymbol{\gamma}}_t^2\left[\mathbb{E}[||\mathbf{x}_1||^2 + ||\mathbf{x}_0||^2 - 2\mathbf{x}_1^\mathsf{T}\mathbf{x}_0]\right]$$
$$= \dot{\boldsymbol{\gamma}}_t^2\left[\mathbb{E}[||\mathbf{x}_1||^2] + K_{\text{num}}\right],$$

where we used that $\mathrm{Var}[X] = \mathbb{E}[X^2] - \mathbb{E}[X]^2$ and $\mathrm{Cov}[X, Y] = \mathbb{E}[XY] - \mathbb{E}[X]\mathbb{E}[Y]$. We can deconstruct the expression into a sum over the $K_{\text{num}}$ features $x_1^{(i)}$:

$$\mathbb{E}[||\dot{\Psi}_t||^2] = \dot{\boldsymbol{\gamma}}_t^2 \sum_i^{K_{\text{num}}}\left[\mathbb{E}[(x_1^{(i)})^2]\right] + \dot{\boldsymbol{\gamma}}_t^2 \sum_i^{K_{\text{num}}}\left[\mathbb{E}[1]\right]. \tag{14}$$

For our data-dependent coupling, we have $p(\mathbf{x}_0, \mathbf{x}_1) = \sum_{\mathbf{z} \in \mathcal{Z}} p(\mathbf{x}_0|\mathbf{z})p(\mathbf{z}|\mathbf{x}_1)p(\mathbf{x}_1)$ from Equation (4) such that (from Equation (3)):

$$\mathbf{x}_0 = \boldsymbol{\mu}(\mathbf{z}) + \boldsymbol{\sigma}(\mathbf{z}) \odot \epsilon \text{ with } \epsilon \sim \mathcal{N}(\mathbf{0}, \mathbf{I}).$$

Since $\mathbf{z} = f(\mathbf{x}_1)$ is a deterministic function of $\mathbf{x}_1$, we only take the expectation over $\mathbf{x}_1$ and $\epsilon$ to derive

$$\mathbb{E}[||\dot{\Psi}_t||^2] = \mathbb{E}[||\dot{\boldsymbol{\gamma}}_t(\mathbf{x}_1 - \mathbf{x}_0)||^2]$$
$$= \dot{\boldsymbol{\gamma}}_t^2 \mathbb{E}[||(\mathbf{x}_1 - \boldsymbol{\mu}(f(\mathbf{x}_1)) - \boldsymbol{\sigma}(f(\mathbf{x}_1)) \odot \epsilon)||^2]$$

We can deconstruct this expression as a sum over $K_{\text{num}}$ features $x_1^{(i)}$ as

$$\mathbb{E}[||\dot{\Psi}_t||^2] = \dot{\boldsymbol{\gamma}}_t^2 \mathbb{E}\left[\sum_i^{K_{\text{num}}}\left(x_1^{(i)} - \mu(f(x_1^{(i)})) - \sigma(f(x_1^{(i)}))\varepsilon^{(i)}\right)^2\right]$$

$$= \dot{\boldsymbol{\gamma}}_t^2 \mathbb{E}\sum_i^{K_{\text{num}}}\left[\left(x_1^{(i)} - \mu(f(x_1^{(i)}))\right)^2 + \left(\sigma(f(x_1^{(i)}))\varepsilon^{(i)}\right)^2 - 2\left(x_1^{(i)} - \mu(f(x_1^{(i)}))\right)\sigma(f(x_1^{(i)}))\varepsilon^{(i)}\right]$$

$$= \dot{\boldsymbol{\gamma}}_t^2 \sum_i^{K_{\text{num}}}\left[\mathbb{E}\left(x_1^{(i)} - \mu(f(x_1^{(i)}))\right)^2 + \mathbb{E}\left(\sigma(f(x_1^{(i)}))^2(\varepsilon^{(i)})^2\right)\right],$$

since $x_1^{(i)} \perp \epsilon^{(i)}$ which implies

$$\mathbb{E}\left(x_1^{(i)} - \mu(f(x_1^{(i)}))\right)\sigma(f(x_1^{(i)}))\varepsilon^{(i)} = \mathbb{E}[x_1^{(i)}\sigma(f(x_1^{(i)}))\varepsilon^{(i)}] - \mathbb{E}[\mu(f(x_1^{(i)}))\sigma(f(x_1^{(i)}))\varepsilon^{(i)}]$$
$$= \mathbb{E}[x_1^{(i)}\sigma(f(x_1^{(i)}))]\mathbb{E}[\varepsilon^{(i)}] - \mathbb{E}[\mu(f(x_1^{(i)}))\sigma(f(x_1^{(i)}))]\mathbb{E}[\varepsilon^{(i)}]$$
$$= 0,$$

as $\mathbb{E}[\varepsilon^{(i)}] = 0$. Using $\text{Var}[\varepsilon^{(i)}] = \mathbb{E}[(\varepsilon^{(i)})^2] - \mathbb{E}[\varepsilon^{(i)}]^2 = 1$, we can further derive

$$\mathbb{E}[||\dot{\Psi}_t||^2] = \dot{\gamma}_t^2 \sum_i^{K_{\text{num}}} \left[ \mathbb{E}\left[ \left(x_1^{(i)} - \mu(f(x_1^{(i)}))\right)^2 \right] \right] + \dot{\gamma}_t^2 \sum_i^{K_{\text{num}}} \left[ \mathbb{E}\left[ \sigma(f(x_1^{(i)}))^2 \right] \right]. \tag{15}$$

If we now compare Equation (14) and Equation (15), we recognize that to that $\mathbb{E}[||\dot{\Psi}_t||^2]$ is smaller under our data-dependent coupling, it suffices to show that feature-wise that

$$\mathbb{E}\left[ \left(x_1^{(i)} - \mu(f(x_1^{(i)}))\right)^2 \right] \leq \mathbb{E}[(x_1^{(i)})^2] = 1, \tag{16}$$

since we standardize $x_1^{(i)}$ to zero mean, unit variance and

$$\mathbb{E}\left[ \sigma(f(x_1^{(i)}))^2 \right] \leq \mathbb{E}[1] = 1. \tag{17}$$

Note that if we are using the DT encoder, $f(x_1^{(i)})$ simply indicates in which of the $K_i$ terminal leafs the observation falls. The $k$th terminal leaf reflects an interval $[\tau_{k-1}^{(i)}, \tau_k^{(i)}]$ on the real line. Based on all observations falling into the $k$th interval, DT learns a Gaussian distribution with parameters $\mu_k$ and $\sigma_k$. This allows us to rewrite Equation (16) as

$$\mathbb{E}\left[ \left(x_1^{(i)} - \mu(f(x_1^{(i)}))\right)^2 \right] = \sum_{k=1}^{K_i} \text{Pr}(\tau_{k-1}^{(i)} < x_1^{(i)} \leq \tau_k^{(i)}) \underbrace{\mathbb{E}_{x_1^{(i)}|x_1^{(i)} \in [\tau_{k-1}^{(i)}, \tau_k^{(i)}]} \left[ \left(x_1^{(i)} - \mu_k\right)^2 \right]}_{\text{MSE in } k\text{th interval}}.$$

For each interval, the DT encoder learns the optimal $\mu_k$ by maximizing the likelihood, i.e., minimizing the mean squared error *within the $k$th interval*, which is equivalent to the expectation on the right-hand side. We assign the *optimal $\mu_k$*, i.e., the MSE is necessarily lower than choosing $\mu_k = 0$ in the case of an independent coupling. This proves that Equation (16) holds.

For proofing the second condition in Equation (17), we only need to show $\sigma(f(x_1^{(i)})^2 \leq 1$ for all $x_1^{(i)}$. That is, the variance of the terminal leaf in which $x_1^{(i)}$ falls should be at most one for all possible $x_1^{(i)}$. This directly follows from the fact that we separate observations into smaller groups based on the intervals determined by the DT encoder. Note that $[\tau_{k-1}^{(i)}, \tau_k^{(i)}] \leq \text{supp}(x_1^{(i)})$ for all $k$, which implies $\sigma_k^2 \leq 1$ for all $k$.

Since both sufficient conditions in Equation (16) and Equation (17) are proven to hold, we conclude that

$$\dot{\gamma}_t^2 \mathbb{E}[||(\mathbf{x}_1 - \boldsymbol{\mu}(f(\mathbf{x}_1)) - \boldsymbol{\sigma}(f(\mathbf{x}_1)) \odot \epsilon)||^2] \leq \dot{\gamma}_t^2 \left[ \mathbb{E}[||\mathbf{x}_1||^2] + K_{\text{num}} \right], \tag{18}$$

i.e., our data-dependent coupling based on the DT encoder is able to achieve a lower transport cost bound than the independent coupling.

## B  BENCHMARK DATASETS

Our selected benchmark datasets are highly diverse, particularly in cardinality of categorical features (see Table 4), and have been used extensively in previous research (Kotelnikov et al., 2023; Mueller et al., 2025; Shi et al., 2025; Tiwald et al., 2025; Zhang et al., 2024b). All datasets are publicly accessible and licensed under creative commons. We randomly split each dataset into 70/10/20 training, validation and test sets. Numerical features in $\mathbf{x}_{\text{num}}$ are quantile transformed and standardized, following the usual practice for tabular data generation.

**Missing value simulation.**  First, we remove any rows with missing values in the target, to ensure that a valid estimation of the Machine Learning Efficiency metric, or in any of the numerical features. This gives us full control over the missingness proportion and mechanism. To simulate missingness, we adopt the approach from prior imputation studies (see e.g., Muzellec et al., 2020; Zhao et al., 2023; Zhang et al., 2024a). Note that missing values in categorical features are trivial to deal with by simply encoding them as a separate category. In the following, we therefore focus exclusively on generating missing values in numerical features.

Table 4: Overview of the selected experimental datasets. We count the target towards the respective features. The minimum and maximum number of categories are taken over all categorical features.

| Dataset | License | Prediction task | Total no. observations | No. of features categorical | No. of features continuous | No. of categories min | No. of categories max |
|---|---|---|---|---|---|---|---|
| adult (Becker & Kohavi, 1996) | CC BY 4.0 | binary class. | 48 842 | 9 | 6 | 2 | 42 |
| beijing (Chen, 2015) | CC BY 4.0 | regression | 41 757 | 1 | 10 | 4 | 4 |
| default (Yeh, 2009) | CC BY 4.0 | binary class. | 30 000 | 10 | 14 | 2 | 11 |
| diabetes (Clore et al., 2014) | CC BY 4.0 | binary class. | 101 766 | 29 | 8 | 2 | 523 |
| news (Fernandes et al., 2015) | CC BY 4.0 | regression | 39 644 | 14 | 46 | 2 | 2 |
| shoppers (Sakar et al., 2019) | CC BY 4.0 | binary class. | 12 330 | 8 | 10 | 2 | 20 |

We choose to simulate missing values under a missing not at random (MNAR) mechanism, as it combines a missing at random (MAR), $p(\mathbf{m}|\mathbf{x}_{num}, \mathbf{x}_{num}^{(latent)}) = p(\mathbf{m}|\mathbf{x}_{num})$, with a missing completely at random (MCAR), $p(\mathbf{m}|\mathbf{x}_{num}, \mathbf{x}_{num}^{(latent)}) = p(\mathbf{m})$, mechanism (see Little & Rubin, 2019). Following prior work (Muzellec et al., 2020; Zhao et al., 2023; Zhang et al., 2024a), we simulate missing values using a two-step procedure. First, under a MAR mechanism, we randomly select 30% of the numerical and categorical features as inputs to a randomly initialized logistic model, to determine the missingness probabilities for the remaining numerical features. The model's coefficients are scaled to preserve variance, and the bias term is adjusted via line search to achieve a 10% missing rate. Second, we apply an MCAR mechanism by setting 10% of the logistic model's input features (including selected categorical ones) to missing. Thus, the missingness introduced by the MAR mechanism may be explained by values which now have been masked by the MCAR mechanism, making them latent to them model. Throughout, we ensure that we do not introduce any missings to the target, to ensure that we can determine the Machine Learning Efficiency metric. Introducing non-trivial missings increases the complexity of the joint distribution, both in terms of dimensions and dependencies, and makes the job for the generative models more difficult.

## C IMPLEMENTATIONS

We benchmark TabCascade against recent state-of-the-art generative models, many of which are diffusion-based. To ensure that the benchmarks are fair, we align the models as much as possible. For diffusion-based models, we use the same MLP-based architecture with the same bottleneck dimension. The MLP contains a projection layer onto the bottleneck dimension (256-dimensional), five fully connected layers, and an output layer. The only differences stem from variations in the required inputs or outputs, which make certain minor model-specific changes to the MLP necessary, e.g., CDTD requires predicted logits for categorical features. For all models, we use the same time encoder based on positional embeddings with a subsequent 2-layer MLP. For non-diffusion-based models, we try to align the layer dimensions. In any case, similar to Mueller et al. (2025) we scale each model to a total of $\approx 3$ million parameters on the adult dataset (when simulating missing values according to the MNAR mechanism) and train it for 30 000 steps with a batch size of 4096. For diffusion-based models, we limit the maximum training time to 30 minutes to increase model comparability. We use the same data pre-processing pipeline for all models and add model-specific pre-processing steps where necessary. For diffusion-based models, we mostly align the sampling steps to 200. One exception is TabDDPM, which builds on DDPM and therefore requires more sampling steps (default = 1000). A second exception is TabDiff, for which we adopt the authors' suggestion of 50 sampling steps. Otherwise, TabDiff sampling will take an order of magnitude more time than other models, in particular for larger datasets. When available, we follow the default hyperparameters provided by the authors or the package / code documentation. We run all experiments using PyTorch 2.7.1 and TensorFloat32 using a MIG instance on an A100 GPU. All code and configuration files are made available to ensure reproducibility.

Below, we briefly elaborate on each baseline model and its implementation:

**ARF** (Watson et al., 2023) – a generative model that is based on a random forest for density estimation. The implementation is available at https://github.com/bips-hb/arfpy and licensed under the MIT license. We use package version 0.1.1. For training, we utilize 16 CPU cores and 20 trees as suggested in the paper.

**CTGAN** (Xu et al., 2019) – one of the most popular GAN-based models for tabular data. The implementation is available as part of the Synthetic Data Vault (Patki et al., 2016) at `https://github.com/sdv-dev/CTGAN` and licensed under the Business Source License 1.1. We use package version 0.11.0. The architecture dimensions are adjusted to be comparable to MLP used for the diffusion-based models. The model requires that the batch size is divisible by 10. Therefore, we adjust the default batch size of 4096 downwards accordingly.

**TVAE** (Xu et al., 2019) – a VAE-based model for tabular data. The implementation is available as part of the Synthetic Data Vault (Patki et al., 2016) at `https://github.com/sdv-dev/CTGAN` and licensed under the Business Source License 1.1. We use package version 0.11.0. The architecture dimensions are adjusted to be comparable to MLP used for the diffusion-based models.

**TabDDPM** (Kotelnikov et al., 2023) – a diffusion-based generative model for tabular data that combines multinomial diffusion (Hoogeboom et al., 2021) and DDPM (Sohl-Dickstein et al., 2015; Ho et al., 2020). We base our code on the official implementation available at `https://github.com/yandex-research/tab-ddpm` under the MIT license. However, we adjust the model to allow for unconditional generation in case of classification tasks.

**TabSyn** (Zhang et al., 2024b) – a latent diffusion model that first learns a transformer-based VAE to map mixed-type data to a continuous latent space. The diffusion model is then trained on that latent space. Note that despite TabSyn utilizing a separately trained encoder, this does *not* result in a lower-dimensional latent space and therefore, does not speed up sampling. We use the official code available at `https://github.com/amazon-science/tabsyn` under the Apache 2.0 license. We leave the transformer-based VAE unchanged and scale only the MLP.

**TabDiff** (Shi et al., 2025) – a continuous time diffusion model that combines score matching (Song et al., 2021; Karras et al., 2022) with masked diffusion (Sahoo et al., 2024) and learnable, feature-specific noise schedules. Originally, it relies on transformer-based encoder and decoder parts, which we remove from the model to improve comparability. However, we keep the other parts, including the tokenizer. We scale the bottleneck dimension down to 256 and adjust the hidden layers accordingly, to align the architecture more with the other diffusion-based models. Otherwise, we use the official implementation available at `https://github.com/MinkaiXu/TabDiff` under the MIT license.

**CDTD** (Mueller et al., 2025) – a continuous time diffusion that combines score matching (Song et al., 2021; Karras et al., 2022) with score interpolation (Dieleman et al., 2022) and leanable noise schedules. Based on the performance results in the original paper, we use the *by type* noise schedule, that is, we learn an adaptive noise schedule per feature type. We use the official implementation available at `https://github.com/muellermarkus/cdtd_simple` under the MIT license. To align architectures and improve comparability, we adjust the MLP dimensions.

None of the selected benchmark models accommodate the generation of missing values in numerical features out of the box. Therefore, to achieve a fair comparison, we endow each model with the simple means to generate missing values. To avoid manipulating a model's internals and therewith potentially disrupting the training dynamics, we confine ourselves to changing the data encoding. For each numerical feature that contains missing values, we introduce an additional binary missingness mask. We simply treat this mask as an additional categorical feature to be generated and mean-impute the missing values. After sampling, we overwrite the generated numerical values with `NaN` based on the generated missingness mask.

## D  TABCASCADE IMPLEMENTATION

Since we make use of two separate models instead of a single model, we use the same MLP architecture as for the baselines but scale various layers and components down to achieve $\approx$3 million parameters on the `adult` dataset. We add the conditioning information about $\mathbf{x}_{\text{low}}$ as an additive embedding to the bottleneck layer. Instead of parameterizing $\mathbf{u}_t^{\boldsymbol{\theta}}(\mathbf{x}_t|\mathbf{x}_{\text{low}})$ directly with a neural network $f^{\boldsymbol{\theta}}(\mathbf{x}_t, \mathbf{x}_{\text{low}}, t)$, we use the known form of the vector field to parameterize

$$\mathbf{u}_t^{\boldsymbol{\theta}}(\mathbf{x}_t|\mathbf{x}_{\text{low}}) = \dot{\boldsymbol{\gamma}}_t(\mathbf{x}_{\text{low}})f^{\boldsymbol{\theta}}(\mathbf{x}_t, \mathbf{x}_{\text{low}}, t). \tag{19}$$

We train $p_{\text{low}}^{\boldsymbol{\theta}}$ and $p_{\text{high}}^{\boldsymbol{\theta}}$ simultaneously using teacher forcing. That is, we train $p_{\text{high}}^{\boldsymbol{\theta}}$ using the real data instances, instead of the ones generated by $p_{\text{low}}^{\boldsymbol{\theta}}$. This enables an end-to-end training of two separate

models with a reduced time penalty. The training and generation processes are described in detail in Algorithm 1 and Algorithm 2 below.

For the DT encoder we set a max depth of 8 which on the `adult` dataset translates to an average of 66 distinct groups for each feature that are captured by $\mathbf{z}_{\text{num}}$. For the GMM encoder, we set the maximum number of components to 30 to keep the training time below 1 minute on the `adult` dataset. Empirical evidence shows that this does not effectively limit the estimated number of components, which typically lie below 30.

---

**Algorithm 1** Training

---

**# Pre-Training**

Learn feature-wise encoder $z_{\text{num}}^{(i)} = \text{Enc}_i(x_{\text{num}}^{(i)})$

**# Training**

Sample $\mathbf{x}_{\text{num}}, \mathbf{x}_{\text{cat}} \sim p_{\text{data}}$
Retrieve $z_{\text{num}}^{(i)} = \text{Enc}_i(x_{\text{num}}^{(i)}) \forall i$ and construct $\mathbf{x}_{\text{low}} = (\mathbf{x}_{\text{cat}}, \mathbf{z}_{\text{num}}) = [x_{\text{low}}^{(j)}]_{j=1}^{K_{\text{low}}}$
Construct mask for inflated and missing values in $\mathbf{x}_{\text{num}}$

**# Low-Resolution Model**

Train CDTD model (Mueller et al., 2025)

**# High-Resolution Model**

Sample $t \sim \mathcal{U}(0,1)$ and $\epsilon \sim \mathcal{N}(\mathbf{0}, \mathbf{I})$
Compute $\mathbf{x}_0$ using Equation (3)
Compute $\mathbf{x}_t = \boldsymbol{\gamma}_t(\mathbf{x}_{\text{low}})\mathbf{x}_1 + (1 - \mathbf{x}_{\text{low}})\mathbf{x}_0$
Form predictions $\mathbf{u}_t^{\boldsymbol{\theta}}(\mathbf{x}_t|\mathbf{x}_{\text{low}}) = \dot{\boldsymbol{\gamma}}_t(\mathbf{x}_{\text{low}})f^{\boldsymbol{\theta}}(\mathbf{x}_t, \mathbf{x}_{\text{low}}, t)$
Compute MSE between $\mathbf{u}_t^{\boldsymbol{\theta}}(\mathbf{x}_t|\mathbf{x}_{\text{low}})$ and the target (mask losses for missing and inflated values; see Equation (7))
Backpropagate.

---

---

**Algorithm 2** Generation

---

**# Low-Resolution Model**

Sample $\mathbf{x}_0^{(j)} \sim \mathcal{N}(\mathbf{0}, \mathbf{I}) \, \forall j$
**for** $t$ in $t_{\text{grid}}$ with step size $h$ **do**

 Predict $\Pr(x_{\text{low}}^{(j)} = c|\{\mathbf{x}_t^{(j)}\}_{j=1}^{K_{\text{cat}}}.t) \forall c \in \{0, 1, \ldots, C_j\} \, \forall j$

 Compute $\mu_t^{(j)} = \sum_{c=1}^{C_j} \Pr(x_{\text{low}}^{(j)} = c|\{\mathbf{x}_t^{(j)}\}_{j=1}^{K_{\text{low}}}.t) \cdot \mathbf{x}_1^{(j)}(c) \, \forall j$

 Compute $u_t^{(j)}(\mathbf{x}_t|\mathbf{x}_1) = \frac{\mu_t^{(j)} - \mathbf{x}_t^{(j)}}{\sigma^2(t)}$

 Take update step $\mathbf{x}_t^{(j)} = \mathbf{x}_t^{(j)} + h \cdot u_t^{(j)}(\mathbf{x}_t|\mathbf{x}_1) \, \forall j$
**end for**
Assign classes based on $\arg\max_c \Pr(x_{\text{low}}^{(j)} = c|\{\mathbf{x}_1^{(j)}\}_{j=1}^{K_{\text{low}}}.t = 1 - h) \forall c \in \{0, 1, \ldots, C_j\} \, \forall j$

**# High-Resolution Model**

Retrieve $\boldsymbol{\mu}(\mathbf{z}_{\text{num}}), \boldsymbol{\sigma}(\mathbf{z}_{\text{num}})$ and sample $\mathbf{x}_0$ using Equation (3)
Solve ODE $\mathbf{x}_{\text{num}} = \mathbf{x}_0 + \int_{t=0}^{t=1} \dot{\boldsymbol{\gamma}}(\mathbf{x}_{\text{low}})f^{\boldsymbol{\theta}}(\mathbf{x}_t, \mathbf{x}_{\text{low}}, t)\mathrm{d}t$

**# Post-Process Samples**

Overwrite $\mathbf{x}_{\text{num}}$ with inflated or missing values using Equation (2)
Return $\mathbf{x}_{\text{cat}}, \mathbf{x}_{\text{num}}$

---

# E  EVALUATION METRICS

**Univariate densities (Shape, WD, JSD).** To evaluate the quality of the column-wise, univariate densities, we mainly use the popular Shape metric, which is part of the SDMetrics library (version

0.22.0) of the Synthetic Data Vault (Patki et al., 2016). For numerical features, we use the Kolmogorov-Smirnov statistic $K_{\text{stat}} \in [0, 1]$ and compute the score as $1 - K_{\text{stat}}$ feature-wise. Note that $K_{\text{stat}}$ cannot be computed from observations with missing values. Therefore, we remove any rows with missing values in the numerical features beforehand. For categorical features, we compute the Total Variation Distance (TVD) based on the empirical frequencies of each category value, expressed as proportions $R_c$ and $S_c$ in the real and synthetic datasets, respectively. The TVD between real and synthetic datasets is then given as

$$\delta(R, S) = \frac{1}{2} \sum_{c \in \mathcal{C}} |R_c - S_c|.$$

Again, we let $1 - \delta(R, S)$ to ensure that an increasing score (up to 1) indicates improved sample quality. The average score over all features gives the Shape score reported in our results

To get a more nuanced impression about the univariate densities, we additionally report the Wasserstein distance (WD) for numerical features and the Jensen-Shannon divergence (JSD) for categorical features. Qualitatively, we expect them to convey the same information as the Shape metric.

**Bivariate densities (Trend).** To get a better idea of the accuracy of feature interactions in the synthetic data, we evaluate the Trend score, which is another metric provided by the SDMetrics library (version 0.22.0) of the Synthetic Data Vault (Patki et al., 2016). This metrics focuses on the sample quality in terms of accurate pair-wise correlations. Hence, the aim is to compute a score between every pair of features. For two numerical features, we can simply compute the Pearson correlation coefficient. We denote the score as

$$d_{i,j}^{\text{num}} = 1 - 0.5 \cdot |S_{i,j} - R_{i,j}|,$$

where $S_{i,j}$ and $R_{i,j}$ represent the Pearson correlation between features $i$ and $j$ computed on the synthetic and real data, respectively.

For two categorical features, we derive the score from the normalized contingency tables, i.e., from te proportion of samples in each possible combination of categories. To determine the difference between real and synthetic data, we can use the Total Variation Distance (TVD) such that

$$d_{i,j}^{\text{cat}} = 1 - 0.5 \sum_{c_i \in \mathcal{C}_i} \sum_{c_j \in \mathcal{C}_j} |S_{c_i,c_j} - R_{c_i,c_j}|,$$

where $\mathcal{C}_i$ and $\mathcal{C}_j$ are the set of categories of features $i$ and $j$ and $S_{i,j}, R_{i,j}$ are the cells from the normalized contigency table corresponding to these categories.

To be able to compute a comparable score when comparing features of different types, i.e., a numerical and a categorical feature, we first discretize the numerical feature into ten bins and then compute the TVD as explained above. For all scores, a higher score indicates better sample quality. The overall Trend score is the average over all pair-wise scores. Since this metric cannot accommodate missing values in numerical features, we again remove rows with such missing values beforehand.

**Joint density (Detection score).** While the other metrics so far focus on the sample quality in terms of univariate densities or pair-wise distributions, we are particularly interested in the overall quality of the full joint distribution. Following the typical approach in the literature (Bischoff et al., 2024; Mueller et al., 2025; Shi et al., 2025), we train a detection model to differentiate between fake and real samples, which make up the training data in equal proportions. This approach is also called a classifier two-sample test (C2ST) (Bischoff et al., 2024).

To ensure that the detection model is sensitive to small changes in the distribution, we choose LightGBM (Ke et al., 2017). Gradient-boosting models have shown remarkable performance on tabular datasets (Borisov et al., 2022). LightGBM has been particularly designed for improved efficiency, which is important for the evaluation of the detection score on larger datasets. Another advantage is that it naturally accommodates missings in numerical features. This allows the detection score to indirectly capture how well the generative model learned the missingness mechanism. To train LightGBM, we sample a synthetic dataset of the same size as the training set used for the generative model. The objective is to classify whether a given sample is real or synthetic. We use 5-fold cross-validation to estimate the out-of-sample performance, with a max depth = 5 and 500

boosting iterations. To get the final detection score, we first use the highest average AUC obtained over validation sets across boosting iterations, denoted by $\bar{A}$. The detection score is then computed as

$$\text{Detection Score} = 1 - (\max(0.5, \bar{A}) \cdot 2 - 1),$$

such that a score of 1 indicates that the model cannot distinguish fake and real samples at all. On the other extreme, a score of 0 indicates that the model can perfectly classify the samples into fake and real. This procedure mimics the detection metric in the SDMetrics library of the Synthetic Data Vault (Patki et al., 2016) but uses a much more powerful detection model.

**Downstream-task performance (Machine learning efficiency).** Machine learning efficiency (MLE; sometimes also called efficacy or utility) measures the usefulness of the synthetic data for the downstream prediction task, either binary classification or regression, associated with a given dataset. This represents a train-synthetic-test-real strategy: We train a predictor on the synthetic data and test the predictor's out-of-sample performance on the real test data. Similarly, we get the test set performance by training the predictor on the real training data. For regression tasks, we evaluate the RMSE and for classification tasks the AUC. Since our goal is to generate a realistic and faithful copy of the true data, we expect both models to perform similarly on the downstream task, regardless of which data has been used for training. Thus, only the relative comparison of the model performances matters, which we report using their absolute difference

$$\text{MLE Score} = |M_S - M_R|, \text{ with } M \in \{\text{AUC}, \text{RMSE}\}.$$

As the predictor, we again pick LightGBM (Ke et al., 2017) with a max depth of 5 and 500 boosting iterations because of its efficiency and strong predictive performance on tabular data. It also automatically accommodates missings in numerical features. Note that the generative model's ability to generate missing values is evaluated in two different ways: (1) LightGBM may rely directly on missing values to infer the target and (2) the generative model may place missing values incorrectly and thereby eradicates information that would be needed (and is available in the true training data) for the prediction task. Hence, there is a twofold negative impact of a generative model that is not able to accurately learn the missingness mechanism on the downstream task performance.

**Diversity (Distance to closest record share).** Our goal is to approximate the true generative process and provide a fair comparison to existing baselines. As such, we are, similar to previous work, not concerned with any privacy considerations. To obtain privacy guarantees, context-specific choices, for instance, with regards to the budget for differential privacy, must be made. Such in-processing privacy mechanisms as well as pre-processing and post-processing techniques are typically model agnostic but depend heavily on the dataset as well other considerations, such as legal and ethical questions. Hence, we investigate the distance to closest record (DCR) share only as a metric of diversity rather than privacy. Most importantly, it can inform about models which simply copy training samples, without actually learning the distribution

To ensure all features are on the same scale, we min-max-scale numerical features and one-hot encode categorical features. We allow for missing values in numerical features by using mean imputation and adding the missingness indicator to the one-hot encoded categorical features. For each synthetic sample we then find the nearest neighbor in the training set in terms of their $L_2$ distance (Zhao et al., 2021). Since the DCR is only meaningful when compared to some reference, we report the DCR share (Zhang et al., 2024b; Shi et al., 2025). Let $d_{\text{train}}^{(i)}$ and $d_{\text{test}}^{(i)}$ be the $L_2$ distance of the of the $i$-th synthetic sample to the closest training and test sample, respectively. Then we set

$$S^{(i)} = \begin{cases} 1 & \text{if } d_{\text{train}}^{(i)} < d_{\text{test}}^{(i)}, \\ 0 & \text{if } d_{\text{train}}^{(i)} > d_{\text{test}}^{(i)}, \\ 0.5 & \text{if } d_{\text{train}}^{(i)} = d_{\text{test}}^{(i)}, \end{cases}$$

such that synthetic samples being closer to the training samples than the test samples increase the score. The DCR share is then computed as an average over the scores $S^{(i)}$ obtained all synthetic samples. The optimal DCR share is 0.5.

**Fidelity and coverage ($\alpha$-Precision and $\beta$-Recall).** Precision and Recall metrics for generative model evaluation have been proposed by Sajjadi et al. (2018) and refined for tabular data by Alaa

et al. (2022). $\alpha$-Precision measures the probability that synthetic samples resides in the $\alpha$-support of the true distribution and therefore measures sample fidelity. $\beta$-Recall, on the other hand, measures the sample diversity or coverage. That is, what fraction of real samples reside in the $\beta$-support of the generative distribution. For both metrics, higher values indicate better sample quality. For estimation, we rely on the official implementation in the synthcity package (Qian et al., 2023) available at `https://github.com/vanderschaarlab/synthcity`. However, we need to make some minor adjustments, in exactly the same way as for the DCR computation, to accommodate the missing values in numerical features.

**Privacy (Membership inference attack).** For completeness, we also the provide scores of a membership inference attack (MIA; Shokri et al., 2017). We follow the implementation in the SynthEval package (Lautrup et al., 2024) available at `https://github.com/schneiderkamplab/syntheval/`.

Let $\mathcal{D}_{\text{train}}, \mathcal{D}_{\text{test}}$ and $\mathcal{D}_{\text{gen}}$ be the training set, test set, and generate data, respectively. First, we split $\mathcal{D}_{\text{test}}$ into $\mathcal{D}_{\text{test}}^{(\text{train})}$ (75%) and $\mathcal{D}_{\text{test}}^{(\text{test})}$ (25%). We then train a LightGBM classifier (Ke et al., 2017) on a training set made up of $\mathcal{D}_{\text{test}}^{(\text{train})}$ and an equally-sized subsample of $\mathcal{D}_{\text{gen}}$. The classifier is trained to predict which samples originated from the generative model. To retrieve score, we combine $\mathcal{D}_{\text{test}}^{(\text{test})}$ with an equally-sized subsample of $\mathcal{D}_{\text{train}}$, make the predictions, and compute the AUC score. We derive the MIA score as

$$\text{MIA Score} = 1 - (\max(0.5, \text{AUC}) \cdot 2 - 1),$$

such that a score of one indicates that an attack is not better than random guessing. The final score we report is an average over five repetitions of the above steps, to account for the uncertainty in the subsampling.

## F  ENCODERS

To encode each $x_{\text{num}}^{(i)}$ into its categorical low-resolution representation $z_{\text{num}}^{(i)}$, we propose two different encoder: (1) a Dirichlet Process Variational Gaussian Mixture Model and (2) a distributional regression tree. Below, we briefly elaborate on the respective implementations and explain our reasoning behind these choices as well as the differences between the two encoders.

### F.1  GAUSSIAN MIXTURE MODEL

An obvious choice for an encoder is a Gaussian Mixture Model (GMM) because it can approximate any density arbitrarily close. However, its classical variant requires pre-specification of the number of components $K$. This is not desirable, since it would require setting a potentially different $K$ for each feature. Instead, we rely on the Dirichlet Process Variational Gaussian Mixture Model (Bishop, 2006) as provided by the sklearn package. The combination with Dirichlet Process leads to a mixture of a theoretical infinite number of components. For practical purposes, this allows us to avoid specifying the number of components per feature and instead infer them directly from the data. We specify a weight concentration prior of 0.001, following settings in Synthetic Data Vault (Patki et al., 2016) package RDT (see `https://github.com/sdv-dev/RDT`). A low prior encourages the model to put most weight on few components, leading to fewer estimated components after training.

During training, the Variational GMM maximizes a variational lower bound to the maximum likelihood objective and does soft clustering of the data points. To assign a value $x_{\text{num}}^{(i)}$ to a discrete category $z_{\text{num}}^{(i)}$ after training and achieve a hard clustering, we let

$$z_{\text{num}}^{(i)} = \arg\max_k w_k \log p_k(x_{\text{num}}^{(i)}) = \arg\max_k \log w_k \mathcal{N}(x_{\text{num}}^{(i)}; \mu_k, \sigma_k^2),$$

where the $w_k$ are the mixture weights. A potential drawback for the GMM is that its components may substantially overlap (see Figure 5). For instance, it is possible that a small variance Gaussian lies in the middle of a high variance Gaussian if this benefits the overall fit. This can make the group derived from hard clustering disconnected on the real line. Also, it can lead to component mean to deviate from the actual mean within the cluster. To address these downsides, we investigate the use of a distributional regression tree instead.

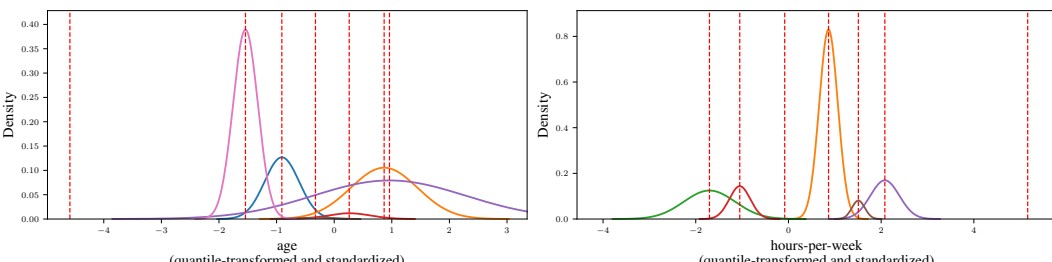

Figure 5: Gaussian components found by the GMM encoder (max components = 7, to align with the number of components found by DT) for two features in the `adult` dataset. The red vertical lines indicate the component means.

## F.2 DISTRIBUTIONAL REGRESSION TREE

Trees split the data into more homogeneous subgroups via binary splits. This can capture abrupt shifts and non-linear functions. Distributional trees (DT; Schlosser et al., 2019) utilize the non-parametric nature of trees and combine it with parametric distributions. The goal is to find homogeneous groups with respect to a parametric distributions such that the model captures abrupt changes in any distributional parameters, such as the mean and variance of a Gaussian distribution.

Training a DT can be interpreted as maximizing a weighted likelihood over $n$ observations:

$$\hat{\boldsymbol{\theta}}(x_{\text{num}}^{(i)}) = \max_{\boldsymbol{\theta} \in \Theta} \sum_{j=1}^{n} w_j(x_{\text{num}}^{(i)}) \cdot \ell(\boldsymbol{\theta}_j; x_{\text{num}}^{(i)}), \tag{20}$$

where $\boldsymbol{\theta}_j = (\mu_j, \sigma_j)$ are the parameters of the $j$th Gaussian component. Note that unlike the GMM, the tree-based approach leads to a hard clustering since $w_j(x_{\text{num}}^{(i)}) \in \{0, 1\}$ simply indicates the allocated terminal leaf for that data point. For each $x_{\text{num}}^{(i)}$, the fitting algorithm goes through the following steps:

- estimate $\hat{\boldsymbol{\theta}}$ via maximum likelihood,

- test for associations or instabilities of the score $\frac{\partial \ell}{\partial \boldsymbol{\theta}}(\hat{\boldsymbol{\theta}}; x_{\text{num}}^{(i)})$,

- choose split of $\text{supp}(x_{\text{num}}^{(i)})$ that yields the highest improvement in the log likelihood,

- repeat until convergence.

The DT exhibits various benefits when compared to the GMM encoder. It searches for a partitioning of $\text{supp}(x_{\text{num}}^{(i)})$ such that values falling into a given segment are more homogeneous with respect to the moments of the Gaussian distribution. Hence, it directly optimizes a hard clustering of data points and defines a Gaussian component only within the clusters. This substantially reduces the possible overlap of the Gaussian components compared to GMM, a feature which allows us to prove Theorem 1. For empirical evidence, compare Figure 6 to Figure 5. This is also an attractive property when determining a suitable Gaussian-based source distribution for flow matching: Sampling from the same Gaussian component guarantees samples being close in data space.

The level of granularity captured by $z_{\text{num}}^{(i)}$ is governed by the complexity of the encoder. DT allows us to specify a maximum tree depth but otherwise learns optimal number of components from the data. Thereby, it is also much faster to train than GMM. We investigate the effect of increasing max depth in additional ablation experiments in Appendix K.

Since no Python implementation of DT is available, we use the disttree R package and combine it with rpy2 to make it callable with Python.

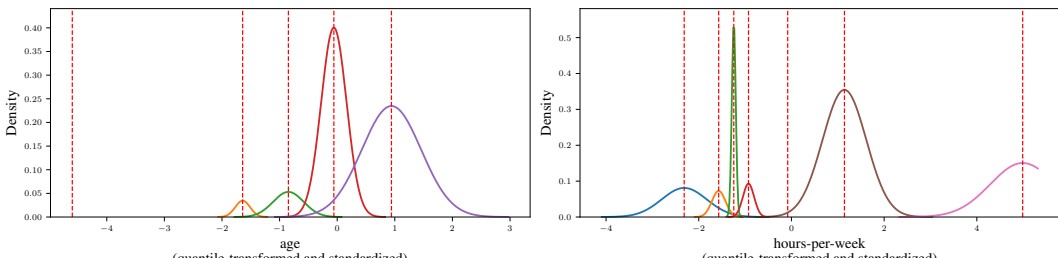

Figure 6: Gaussian components found by the DT encoder (max depth = 3) for two features in the `adult` dataset. The red vertical lines indicate the component means.

### F.3 PRACTICAL CONSIDERATIONS

In practice, $\sigma_k^2$ is never actually zero due to numerical precision. Therefore, if $\sigma_k^2 < \epsilon$, we check empirically whether $\mathrm{Var}[x_{\mathrm{num}}^{(i)}|z_{\mathrm{num}}^{(i)} = k] = 0$. If this is the case, we confirm $\mu_k$ to represent an inflated value.

Furthermore, many features may actually be integers instead of truly continuous values. To keep this ordinal structure, even integers with a smaller number of unique values are often treated as "continuous". In this case, since the granularity of $z_{\mathrm{num}}^{(i)}$ is governed by the complexity of the encoder, if we choose a complex encoder, it can happen that $z_{\mathrm{num}}^{(i)}$ recovers all unique values. But this is *not* a failure case. The consequence is only that the low resolution model already has access to *all* information about that feature and the high resolution model does not need to generate the feature at all. We can interpret this as a data=informed way of deciding when to treat an integer-valued feature as discrete versus (partially) continuous.

## G    POLYNOMIAL PARAMETERIZATION OF TIME SCHEDULE

We parameterize the feature-specific time schedules using the polynomial form proposed by Sahoo et al. (2023). Let $f_\phi : \mathbb{R}^m \times [0, 1] \to \mathbb{R}^d$, where $d$ is the number of features and $\mathbf{c} \in \mathbb{R}^m$ be a vector with conditioning information. We define $f_\phi$ as

$$f_\phi(\mathbf{c}, t) = \frac{\mathbf{a}_\phi^2(\mathbf{c})}{5}t^5 + \frac{\mathbf{a}_\phi(\mathbf{c})\mathbf{b}_\phi(\mathbf{c})}{2}t^4 + \frac{\mathbf{b}_\phi^2(\mathbf{c}) + 2'\mathbf{a}_\phi(\mathbf{c})\mathbf{d}_\phi(\mathbf{c})}{3}t^3 + \mathbf{b}_\phi(\mathbf{c})\mathbf{d}_\phi(\mathbf{c})t^2 + \mathbf{d}_\phi(\mathbf{c})t, \quad (21)$$

where multiplication and division operations are defined element-wise. The parameters $\mathbf{a}_\psi(\mathbf{c}), \mathbf{b}_\psi(\mathbf{c})$ and $\mathbf{d}_\psi(\mathbf{c}))$ are outputs of a neural network with parameters $\psi$ that maps $\mathbb{R}^m \to \mathbb{R}^d \to \mathbb{R}^d$ to construct a common embedding which is the input to separate linear layers that map to $\mathbf{a}_\psi(\mathbf{c}), \mathbf{b}_\psi(\mathbf{c})$ and $\mathbf{d}_\psi(\mathbf{c}))$, respectively. The network uses SiLU activation functions. We can then normalize to get

$$\gamma_t(\mathbf{c}) = \frac{f_\phi(\mathbf{c}, t)}{f_\phi(\mathbf{c}, 1)}, \quad (22)$$

such that $\gamma_t(\mathbf{c})$ is monotonically increasing for $t \in [0, 1]$ and has end points $\gamma_0(\mathbf{c}) = 0$ and $\gamma_1(\mathbf{c}) = 1$. Note that its time-derivative $\dot{\gamma}_t(\mathbf{c})$ is available in closed form.

## H    LEARNED TIME SCHEDULES

Below we display the learned feature-specific time schedules $\gamma_t(\mathbf{x}_{\text{low}})$ for each dataset for the TabCascade model with DT encoder (one line per feature). Since the time schedule is conditioned on $\mathbf{x}_{\text{low}}$ we picture $\mathbb{E}_{\mathbf{x}_{\text{low}}}[\gamma_t(\mathbf{x}_{\text{low}})]$ and $\text{Var}_{\mathbf{x}_{\text{low}}}[\gamma_t(\mathbf{x}_{\text{low}})]$. While on average a linear time schedule seems beneficial, the model does capture some heterogeneity across features.

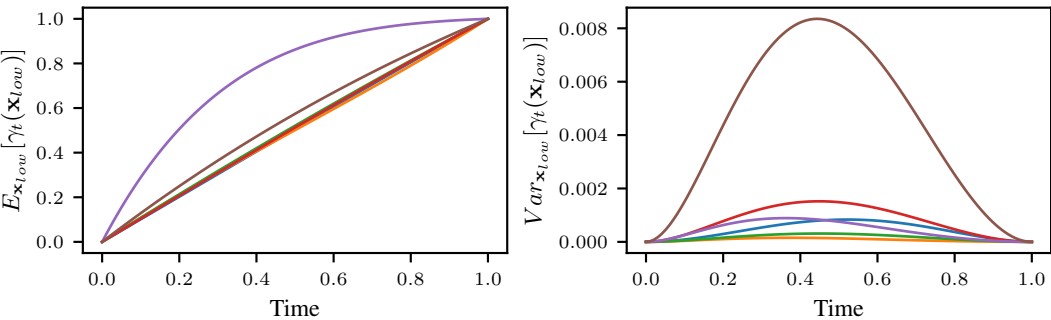

Figure 7: Learned time schedule for the `adult` dataset.

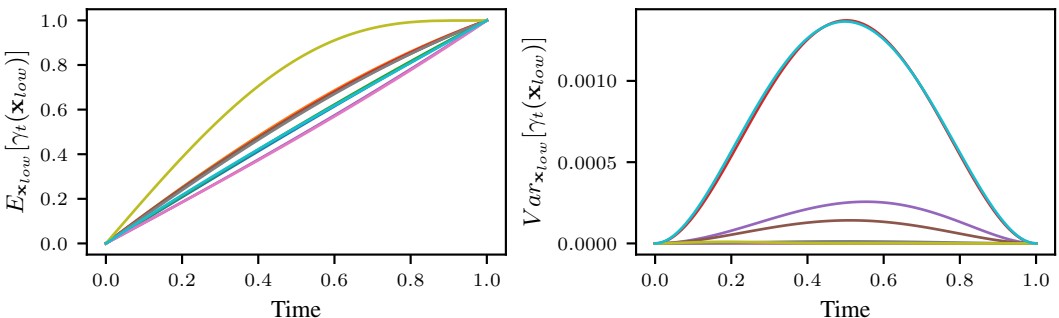

Figure 8: Learned time schedule for the `beijing` dataset.

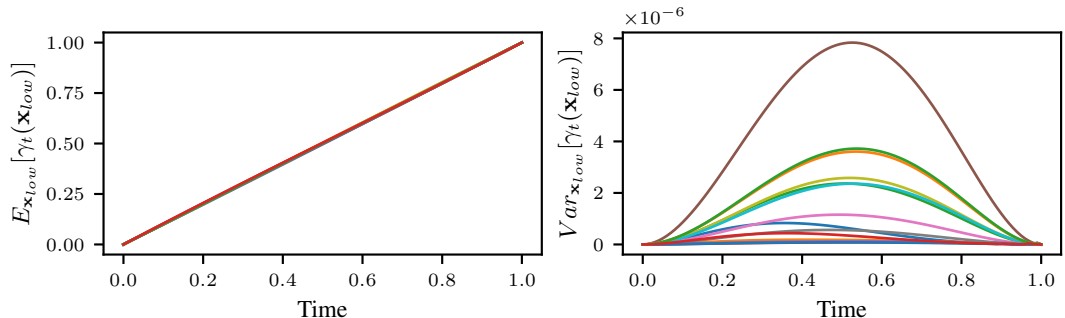

Figure 9: Learned time schedule for the `default` dataset.

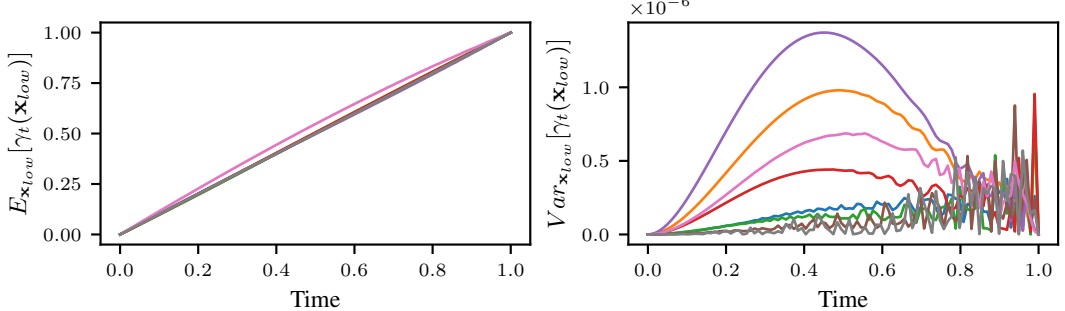

Figure 10: Learned time schedule for the `diabetes` dataset.

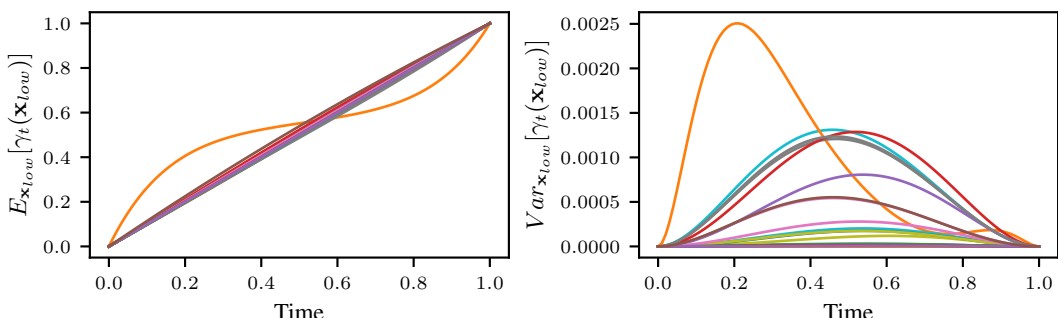

Figure 11: Learned time schedule for the `news` dataset.

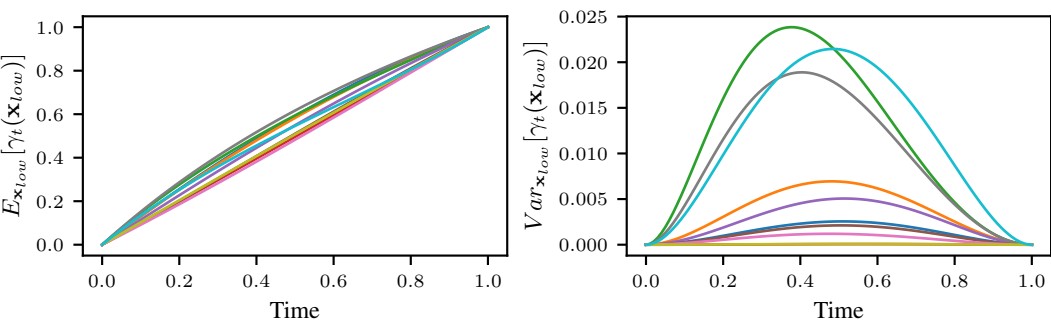

Figure 12: Learned time schedule for the `shoppers` dataset.

# I  QUALITATIVE COMPARISONS

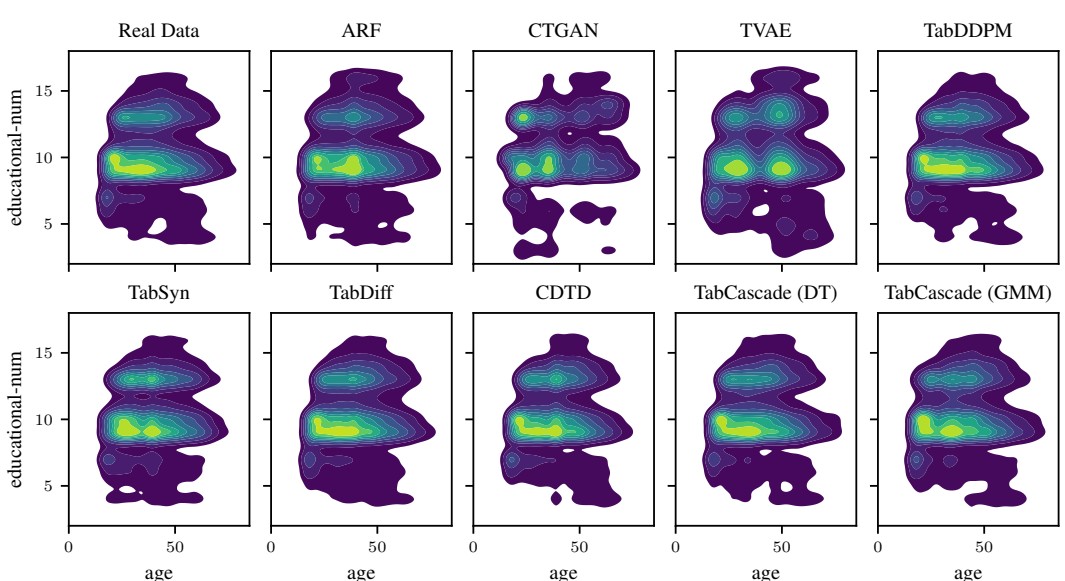

Figure 13: Example of bivariate density from the `adult` dataset.

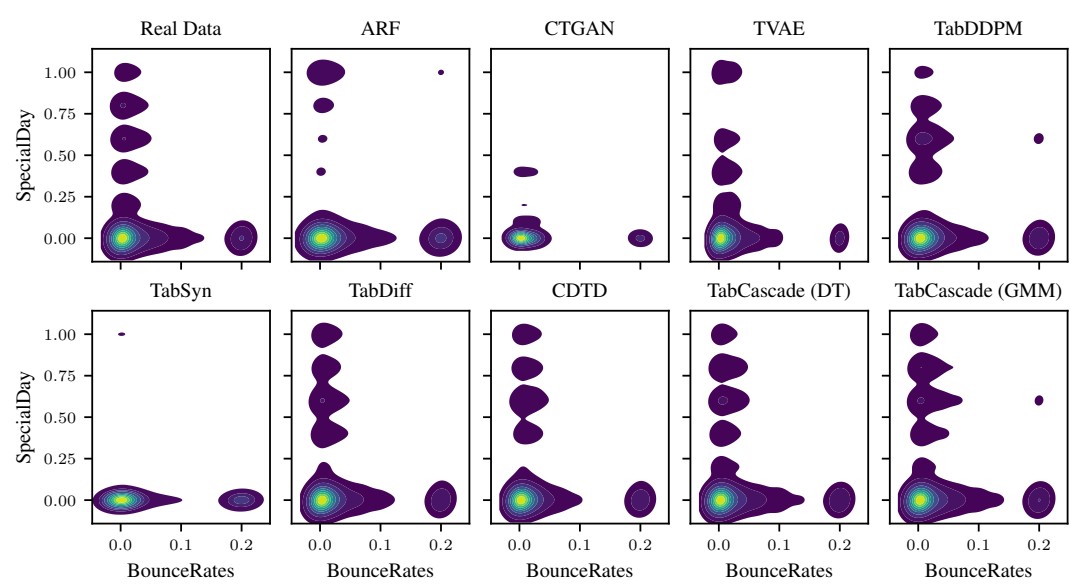

Figure 14: Example of bivariate density from the `shoppers` dataset.

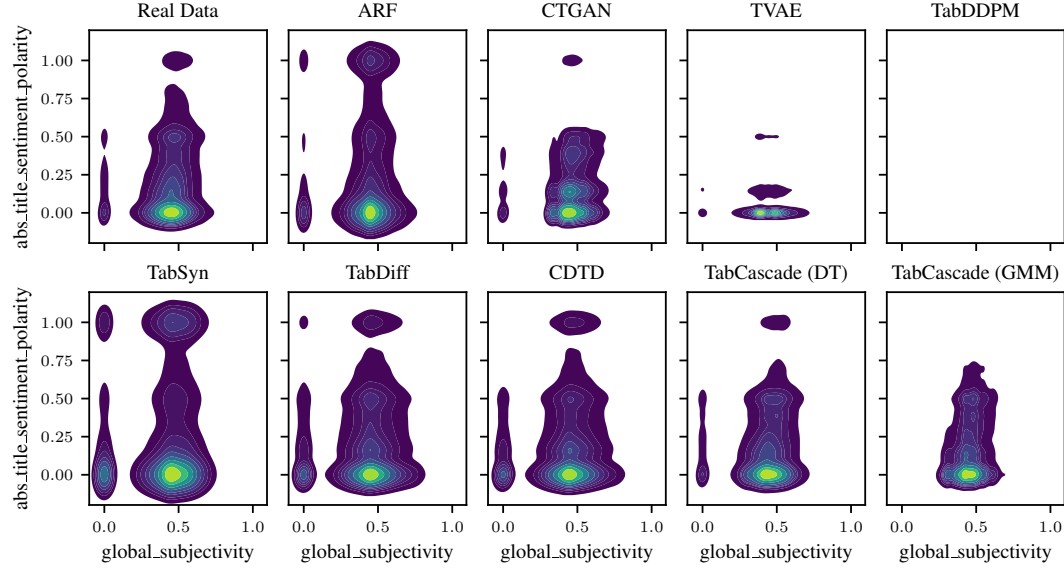

Figure 15: Example of bivariate density from the `news` dataset. TabDDPM produces NaNs for this dataset.

## J DETAILED MAIN RESULTS

Table 5: Comparison of **Detection scores**. **Bold** indicates the best and underline the second best result. We report the average across 3 training runs and 10 different generated samples each.

|  | adult | beijing | default | diabetes | news | shoppers |
|---|---|---|---|---|---|---|
| ARF | $0.350_{\pm0.011}$ | $0.061_{\pm0.002}$ | $0.052_{\pm0.004}$ | $0.288_{\pm0.009}$ | $0.000_{\pm0.000}$ | $0.118_{\pm0.004}$ |
| TVAE | $0.120_{\pm0.015}$ | $0.014_{\pm0.011}$ | $0.038_{\pm0.006}$ | $0.005_{\pm0.004}$ | $0.000_{\pm0.000}$ | $0.179_{\pm0.007}$ |
| CTGAN | $0.077_{\pm0.026}$ | $0.024_{\pm0.003}$ | $0.022_{\pm0.006}$ | $0.090_{\pm0.041}$ | $0.000_{\pm0.000}$ | $0.042_{\pm0.007}$ |
| TabDDPM | $0.725_{\pm0.013}$ | $\underline{0.103}_{\pm0.064}$ | $0.225_{\pm0.004}$ | - | - | $0.162_{\pm0.005}$ |
| TabSyn | $0.424_{\pm0.022}$ | $0.070_{\pm0.009}$ | $0.027_{\pm0.004}$ | $0.090_{\pm0.004}$ | $0.000_{\pm0.000}$ | $0.047_{\pm0.023}$ |
| TabDiff | $\underline{0.747}_{\pm0.014}$ | $0.099_{\pm0.008}$ | $\underline{0.227}_{\pm0.023}$ | $\underline{0.430}_{\pm0.005}$ | $0.000_{\pm0.000}$ | $\underline{0.200}_{\pm0.010}$ |
| CDTD | $0.622_{\pm0.009}$ | $0.080_{\pm0.002}$ | $0.190_{\pm0.008}$ | $0.310_{\pm0.052}$ | $0.000_{\pm0.000}$ | $0.181_{\pm0.005}$ |
| TabCascade (DT) | $\mathbf{0.891}_{\pm0.016}$ | $\mathbf{0.111}_{\pm0.003}$ | $\mathbf{0.579}_{\pm0.009}$ | $\mathbf{0.654}_{\pm0.030}$ | $\mathbf{0.001}_{\pm0.000}$ | $\mathbf{0.389}_{\pm0.016}$ |

Table 6: Comparison of **Shape scores**. **Bold** indicates the best and underline the second best result. We report the average across 3 training runs and 10 different generated samples each.

|  | adult | beijing | default | diabetes | news | shoppers |
|---|---|---|---|---|---|---|
| ARF | $0.985_{\pm0.000}$ | $0.946_{\pm0.001}$ | $0.948_{\pm0.001}$ | $\underline{0.978}_{\pm0.000}$ | $0.905_{\pm0.001}$ | $0.948_{\pm0.001}$ |
| TVAE | $0.893_{\pm0.008}$ | $0.891_{\pm0.030}$ | $0.905_{\pm0.007}$ | $0.869_{\pm0.012}$ | $0.856_{\pm0.018}$ | $0.934_{\pm0.010}$ |
| CTGAN | $0.902_{\pm0.012}$ | $0.909_{\pm0.002}$ | $0.908_{\pm0.012}$ | $0.925_{\pm0.012}$ | $0.916_{\pm0.001}$ | $0.908_{\pm0.003}$ |
| TabDDPM | $0.983_{\pm0.001}$ | $0.968_{\pm0.003}$ | $0.968_{\pm0.001}$ | - | - | $0.944_{\pm0.003}$ |
| TabSyn | $0.972_{\pm0.003}$ | $0.958_{\pm0.003}$ | $0.938_{\pm0.003}$ | $0.917_{\pm0.005}$ | $0.863_{\pm0.011}$ | $0.910_{\pm0.012}$ |
| TabDiff | $\mathbf{0.991}_{\pm0.001}$ | $\underline{0.971}_{\pm0.002}$ | $\underline{0.975}_{\pm0.003}$ | $0.969_{\pm0.001}$ | $\underline{0.927}_{\pm0.001}$ | $\underline{0.975}_{\pm0.001}$ |
| CDTD | $0.984_{\pm0.000}$ | $0.962_{\pm0.001}$ | $0.963_{\pm0.002}$ | $0.968_{\pm0.004}$ | $0.926_{\pm0.002}$ | $0.969_{\pm0.002}$ |
| TabCascade (DT) | $\underline{0.989}_{\pm0.001}$ | $\mathbf{0.976}_{\pm0.001}$ | $\mathbf{0.985}_{\pm0.002}$ | $\mathbf{0.986}_{\pm0.002}$ | $\mathbf{0.948}_{\pm0.001}$ | $\mathbf{0.981}_{\pm0.001}$ |

Table 7: Comparison of **Shape (cat) scores**, which evaluate categorical univariate densities only. **Bold** indicates the best and underline the second best result. We report the average across 3 training runs and 10 different generated samples each.

| | adult | beijing | default | diabetes | news | shoppers |
|---|---|---|---|---|---|---|
| ARF | $\mathbf{0.996}_{\pm 0.000}$ | $\mathbf{0.996}_{\pm 0.002}$ | $\mathbf{0.996}_{\pm 0.001}$ | $\mathbf{0.996}_{\pm 0.000}$ | $\mathbf{0.998}_{\pm 0.000}$ | $\mathbf{0.992}_{\pm 0.001}$ |
| TVAE | $0.896_{\pm 0.004}$ | $0.839_{\pm 0.022}$ | $0.883_{\pm 0.025}$ | $0.875_{\pm 0.012}$ | $0.888_{\pm 0.009}$ | $0.952_{\pm 0.008}$ |
| CTGAN | $0.893_{\pm 0.008}$ | $0.912_{\pm 0.022}$ | $0.899_{\pm 0.017}$ | $0.929_{\pm 0.010}$ | $0.988_{\pm 0.002}$ | $0.902_{\pm 0.014}$ |
| TabDDPM | $0.981_{\pm 0.002}$ | $0.988_{\pm 0.002}$ | $0.978_{\pm 0.002}$ | - | - | $0.939_{\pm 0.007}$ |
| TabSyn | $0.975_{\pm 0.008}$ | $0.990_{\pm 0.006}$ | $0.949_{\pm 0.005}$ | $0.916_{\pm 0.004}$ | $0.941_{\pm 0.020}$ | $0.916_{\pm 0.038}$ |
| TabDiff | $\underline{0.995}_{\pm 0.001}$ | $\underline{0.995}_{\pm 0.002}$ | $\underline{0.992}_{\pm 0.003}$ | $0.969_{\pm 0.001}$ | $\underline{0.997}_{\pm 0.001}$ | $\underline{0.991}_{\pm 0.001}$ |
| CDTD | $0.988_{\pm 0.001}$ | $0.994_{\pm 0.002}$ | $0.987_{\pm 0.003}$ | $0.982_{\pm 0.002}$ | $0.990_{\pm 0.001}$ | $0.989_{\pm 0.001}$ |
| TabCascade (DT) | $0.989_{\pm 0.001}$ | $0.995_{\pm 0.002}$ | $0.987_{\pm 0.003}$ | $\underline{0.986}_{\pm 0.002}$ | $0.993_{\pm 0.000}$ | $0.984_{\pm 0.002}$ |

Table 8: Comparison of **Shape (num) scores**, which evaluates numerical univariate densities only. **Bold** indicates the best and underline the second best result. We report the average across 3 training runs and 10 different generated samples each.

| | adult | beijing | default | diabetes | news | shoppers |
|---|---|---|---|---|---|---|
| ARF | $0.969_{\pm 0.001}$ | $0.941_{\pm 0.001}$ | $0.914_{\pm 0.002}$ | $0.913_{\pm 0.001}$ | $0.877_{\pm 0.001}$ | $0.913_{\pm 0.002}$ |
| TVAE | $0.890_{\pm 0.016}$ | $0.896_{\pm 0.032}$ | $0.921_{\pm 0.008}$ | $0.847_{\pm 0.011}$ | $0.846_{\pm 0.021}$ | $0.920_{\pm 0.012}$ |
| CTGAN | $0.915_{\pm 0.024}$ | $0.909_{\pm 0.001}$ | $0.914_{\pm 0.009}$ | $0.911_{\pm 0.018}$ | $0.894_{\pm 0.002}$ | $0.914_{\pm 0.008}$ |
| TabDDPM | $\underline{0.985}_{\pm 0.001}$ | $0.966_{\pm 0.004}$ | $0.960_{\pm 0.001}$ | - | - | $0.948_{\pm 0.002}$ |
| TabSyn | $0.968_{\pm 0.006}$ | $0.955_{\pm 0.003}$ | $0.929_{\pm 0.005}$ | $0.921_{\pm 0.011}$ | $0.840_{\pm 0.015}$ | $0.905_{\pm 0.017}$ |
| TabDiff | $0.984_{\pm 0.001}$ | $\underline{0.968}_{\pm 0.002}$ | $\underline{0.962}_{\pm 0.002}$ | $\underline{0.971}_{\pm 0.002}$ | $0.906_{\pm 0.001}$ | $\underline{0.962}_{\pm 0.002}$ |
| CDTD | $0.978_{\pm 0.001}$ | $0.959_{\pm 0.001}$ | $0.946_{\pm 0.002}$ | $0.918_{\pm 0.016}$ | $\underline{0.906}_{\pm 0.002}$ | $0.953_{\pm 0.004}$ |
| TabCascade (DT) | $\mathbf{0.989}_{\pm 0.002}$ | $\mathbf{0.975}_{\pm 0.002}$ | $\mathbf{0.984}_{\pm 0.001}$ | $\mathbf{0.986}_{\pm 0.001}$ | $\mathbf{0.935}_{\pm 0.002}$ | $\mathbf{0.978}_{\pm 0.001}$ |

Table 9: Comparison of **Wasserstein (WD) distances**, which we use to evaluate numerical univariate densities only. **Bold** indicates the best and underline the second best result. We report the average across 3 training runs and 10 different generated samples each.

| | adult | beijing | default | diabetes | news | shoppers |
|---|---|---|---|---|---|---|
| ARF | $0.007_{\pm 0.000}$ | $0.017_{\pm 0.001}$ | $0.017_{\pm 0.002}$ | $0.021_{\pm 0.000}$ | $0.079_{\pm 0.031}$ | $0.017_{\pm 0.001}$ |
| TVAE | $0.021_{\pm 0.003}$ | $0.036_{\pm 0.005}$ | $0.014_{\pm 0.003}$ | $0.026_{\pm 0.003}$ | $0.047_{\pm 0.006}$ | $0.019_{\pm 0.006}$ |
| CTGAN | $0.022_{\pm 0.005}$ | $0.034_{\pm 0.006}$ | $0.011_{\pm 0.002}$ | $0.020_{\pm 0.004}$ | $\underline{0.036}_{\pm 0.013}$ | $0.019_{\pm 0.003}$ |
| TabDDPM | $\mathbf{0.002}_{\pm 0.000}$ | $\underline{0.005}_{\pm 0.001}$ | $0.004_{\pm 0.000}$ | - | - | $0.010_{\pm 0.001}$ |
| TabSyn | $0.006_{\pm 0.001}$ | $0.009_{\pm 0.001}$ | $0.012_{\pm 0.003}$ | $0.027_{\pm 0.010}$ | $0.666_{\pm 0.627}$ | $0.040_{\pm 0.027}$ |
| TabDiff | $0.002_{\pm 0.000}$ | $0.005_{\pm 0.001}$ | $\underline{0.004}_{\pm 0.000}$ | $\underline{0.007}_{\pm 0.001}$ | $0.043_{\pm 0.015}$ | $\underline{0.006}_{\pm 0.001}$ |
| CDTD | $0.004_{\pm 0.000}$ | $0.007_{\pm 0.001}$ | $0.006_{\pm 0.001}$ | $0.025_{\pm 0.006}$ | $0.050_{\pm 0.023}$ | $0.010_{\pm 0.001}$ |
| TabCascade (DT) | $\underline{0.002}_{\pm 0.000}$ | $\mathbf{0.004}_{\pm 0.001}$ | $\mathbf{0.004}_{\pm 0.001}$ | $\mathbf{0.004}_{\pm 0.000}$ | $\mathbf{0.023}_{\pm 0.010}$ | $\mathbf{0.004}_{\pm 0.001}$ |

Table 10: Comparison of **Jensen-Shannon divergences (JSD)**, which we use to evaluate categorical univariate densities only. **Bold** indicates the best and underline the second best result. We report the average across 3 training runs and 10 different generated samples each.

| | adult | beijing | default | diabetes | news | shoppers |
|---|---|---|---|---|---|---|
| ARF | $\mathbf{0.010}_{\pm 0.001}$ | $0.018_{\pm 0.007}$ | $\underline{0.020}_{\pm 0.002}$ | $\mathbf{0.013}_{\pm 0.001}$ | $0.036_{\pm 0.006}$ | $\underline{0.024}_{\pm 0.003}$ |
| TVAE | $0.144_{\pm 0.008}$ | $0.295_{\pm 0.019}$ | $0.141_{\pm 0.023}$ | $0.194_{\pm 0.017}$ | $0.203_{\pm 0.007}$ | $0.068_{\pm 0.006}$ |
| CTGAN | $0.148_{\pm 0.015}$ | $0.097_{\pm 0.031}$ | $0.147_{\pm 0.030}$ | $0.112_{\pm 0.013}$ | $0.068_{\pm 0.028}$ | $0.134_{\pm 0.019}$ |
| TabDDPM | $0.027_{\pm 0.003}$ | $0.012_{\pm 0.003}$ | $0.033_{\pm 0.004}$ | - | - | $0.073_{\pm 0.010}$ |
| TabSyn | $0.041_{\pm 0.010}$ | $0.022_{\pm 0.009}$ | $0.079_{\pm 0.010}$ | $0.112_{\pm 0.003}$ | $0.088_{\pm 0.020}$ | $0.110_{\pm 0.033}$ |
| TabDiff | $\underline{0.011}_{\pm 0.001}$ | $0.015_{\pm 0.006}$ | $\mathbf{0.019}_{\pm 0.002}$ | $0.042_{\pm 0.001}$ | $\mathbf{0.031}_{\pm 0.005}$ | $\mathbf{0.022}_{\pm 0.002}$ |
| CDTD | $0.022_{\pm 0.001}$ | $\mathbf{0.009}_{\pm 0.003}$ | $0.032_{\pm 0.004}$ | $0.032_{\pm 0.002}$ | $\underline{0.033}_{\pm 0.006}$ | $0.024_{\pm 0.003}$ |
| TabCascade (DT) | $0.018_{\pm 0.002}$ | $\underline{0.011}_{\pm 0.002}$ | $0.033_{\pm 0.006}$ | $\underline{0.026}_{\pm 0.002}$ | $0.038_{\pm 0.006}$ | $0.026_{\pm 0.003}$ |

Table 11: Comparison of **Trend scores**. **Bold** indicates the best and underline the second best result. We report the average across 3 training runs and 10 different generated samples each.

|  | adult | beijing | default | diabetes | news | shoppers |
|---|---|---|---|---|---|---|
| ARF | $0.969_{\pm0.001}$ | $0.977_{\pm0.001}$ | $0.952_{\pm0.003}$ | $\mathbf{0.962}_{\pm0.000}$ | $0.951_{\pm0.004}$ | $0.956_{\pm0.001}$ |
| TVAE | $0.782_{\pm0.012}$ | $0.925_{\pm0.016}$ | $0.835_{\pm0.008}$ | $0.761_{\pm0.024}$ | $0.881_{\pm0.016}$ | $0.934_{\pm0.006}$ |
| CTGAN | $0.765_{\pm0.017}$ | $0.943_{\pm0.005}$ | $0.816_{\pm0.006}$ | $0.818_{\pm0.016}$ | $0.885_{\pm0.009}$ | $0.855_{\pm0.010}$ |
| TabDDPM | $0.971_{\pm0.002}$ | $0.991_{\pm0.001}$ | $0.953_{\pm0.009}$ | - | - | $0.931_{\pm0.005}$ |
| TabSyn | $0.943_{\pm0.006}$ | $0.984_{\pm0.003}$ | $0.903_{\pm0.010}$ | $0.848_{\pm0.019}$ | $0.905_{\pm0.004}$ | $0.879_{\pm0.018}$ |
| TabDiff | $\mathbf{0.982}_{\pm0.001}$ | $\underline{0.991}_{\pm0.001}$ | $\mathbf{0.968}_{\pm0.008}$ | $\underline{0.940}_{\pm0.002}$ | $0.958_{\pm0.003}$ | $\underline{0.972}_{\pm0.001}$ |
| CDTD | $0.971_{\pm0.002}$ | $0.988_{\pm0.001}$ | $0.936_{\pm0.019}$ | $0.916_{\pm0.008}$ | $\underline{0.958}_{\pm0.005}$ | $0.971_{\pm0.002}$ |
| TabCascade (DT) | $\underline{0.976}_{\pm0.003}$ | $\mathbf{0.992}_{\pm0.001}$ | $\underline{0.964}_{\pm0.006}$ | $0.936_{\pm0.003}$ | $\mathbf{0.971}_{\pm0.003}$ | $\mathbf{0.975}_{\pm0.002}$ |

Table 12: Comparison of **Trend (mixed) scores**, which evaluate only the dependencies across feature types. **Bold** indicates the best and underline the second best result. We report the average across 3 training runs and 10 different generated samples each.

|  | adult | beijing | default | diabetes | news | shoppers |
|---|---|---|---|---|---|---|
| ARF | $0.959_{\pm0.001}$ | $0.924_{\pm0.002}$ | $\mathbf{0.964}_{\pm0.006}$ | $\underline{0.928}_{\pm0.001}$ | $0.919_{\pm0.010}$ | $0.958_{\pm0.001}$ |
| TVAE | $0.705_{\pm0.021}$ | $0.726_{\pm0.054}$ | $0.759_{\pm0.019}$ | $0.716_{\pm0.032}$ | $0.749_{\pm0.031}$ | $0.921_{\pm0.011}$ |
| CTGAN | $0.685_{\pm0.024}$ | $0.797_{\pm0.015}$ | $0.715_{\pm0.019}$ | $0.708_{\pm0.038}$ | $0.728_{\pm0.021}$ | $0.803_{\pm0.018}$ |
| TabDDPM | $0.966_{\pm0.002}$ | $0.963_{\pm0.003}$ | $0.928_{\pm0.018}$ | - | - | $0.909_{\pm0.006}$ |
| TabSyn | $0.932_{\pm0.005}$ | $0.943_{\pm0.006}$ | $0.867_{\pm0.019}$ | $0.830_{\pm0.047}$ | $0.813_{\pm0.009}$ | $0.852_{\pm0.025}$ |
| TabDiff | $\mathbf{0.977}_{\pm0.001}$ | $\underline{0.967}_{\pm0.004}$ | $\underline{0.957}_{\pm0.015}$ | $\mathbf{0.937}_{\pm0.004}$ | $\underline{0.919}_{\pm0.008}$ | $\underline{0.965}_{\pm0.001}$ |
| CDTD | $0.960_{\pm0.004}$ | $0.953_{\pm0.006}$ | $0.892_{\pm0.038}$ | $0.829_{\pm0.019}$ | $0.914_{\pm0.011}$ | $0.962_{\pm0.002}$ |
| TabCascade (DT) | $\underline{0.968}_{\pm0.006}$ | $\mathbf{0.974}_{\pm0.002}$ | $0.948_{\pm0.013}$ | $0.872_{\pm0.003}$ | $\mathbf{0.939}_{\pm0.009}$ | $\mathbf{0.968}_{\pm0.003}$ |

Table 13: Comparison of **MLE**. Per dataset, **bold** indicates the best and underline the second best result. We report the average (and standard deviation) across 3 training runs and 10 different generated samples each.

|  | adult | beijing | default | diabetes | news | shoppers |
|---|---|---|---|---|---|---|
| ARF | $0.019_{\pm0.003}$ | $0.102_{\pm0.007}$ | $0.014_{\pm0.003}$ | $\underline{0.031}_{\pm0.014}$ | $0.115_{\pm0.045}$ | $0.052_{\pm0.014}$ |
| TVAE | $0.077_{\pm0.018}$ | $0.288_{\pm0.060}$ | $0.017_{\pm0.007}$ | $0.063_{\pm0.015}$ | $0.061_{\pm0.067}$ | $0.026_{\pm0.011}$ |
| CTGAN | $0.094_{\pm0.016}$ | $0.256_{\pm0.018}$ | $0.039_{\pm0.007}$ | $0.081_{\pm0.030}$ | $\mathbf{0.013}_{\pm0.003}$ | $0.116_{\pm0.015}$ |
| TabDDPM | $0.018_{\pm0.005}$ | $0.046_{\pm0.003}$ | $\mathbf{0.007}_{\pm0.005}$ | - | - | $0.014_{\pm0.007}$ |
| TabSyn | $0.029_{\pm0.003}$ | $0.097_{\pm0.015}$ | $0.034_{\pm0.019}$ | $0.093_{\pm0.017}$ | $3.286_{\pm2.633}$ | $0.044_{\pm0.013}$ |
| TabDiff | $\underline{0.015}_{\pm0.002}$ | $0.054_{\pm0.004}$ | $\underline{0.009}_{\pm0.004}$ | $\mathbf{0.023}_{\pm0.018}$ | $0.082_{\pm0.026}$ | $0.021_{\pm0.006}$ |
| CDTD | $0.016_{\pm0.003}$ | $\underline{0.037}_{\pm0.004}$ | $0.009_{\pm0.005}$ | $0.053_{\pm0.016}$ | $0.147_{\pm0.036}$ | $\underline{0.011}_{\pm0.006}$ |
| TabCascade (DT) | $\mathbf{0.007}_{\pm0.001}$ | $\mathbf{0.035}_{\pm0.004}$ | $0.009_{\pm0.004}$ | $0.036_{\pm0.015}$ | $\underline{0.054}_{\pm0.046}$ | $\mathbf{0.009}_{\pm0.005}$ |

Table 14: Comparison of $\alpha$-**Precision scores**. Per dataset, **bold** indicates the best and underline the second best result. We report the average (and standard deviation) across 3 training runs and 10 different generated samples each.

|  | adult | beijing | default | diabetes | news | shoppers |
|---|---|---|---|---|---|---|
| ARF | $0.991_{\pm0.003}$ | $0.933_{\pm0.003}$ | $0.957_{\pm0.004}$ | $0.976_{\pm0.002}$ | $0.898_{\pm0.005}$ | $0.964_{\pm0.006}$ |
| TVAE | $0.766_{\pm0.021}$ | $0.697_{\pm0.165}$ | $0.772_{\pm0.078}$ | $0.261_{\pm0.070}$ | $0.139_{\pm0.061}$ | $0.938_{\pm0.025}$ |
| CTGAN | $0.804_{\pm0.077}$ | $0.806_{\pm0.009}$ | $0.825_{\pm0.006}$ | $0.878_{\pm0.055}$ | $\underline{0.930}_{\pm0.008}$ | $0.922_{\pm0.075}$ |
| TabDDPM | $0.928_{\pm0.012}$ | $0.964_{\pm0.005}$ | $0.907_{\pm0.007}$ | - | - | $0.767_{\pm0.017}$ |
| TabSyn | $0.970_{\pm0.023}$ | $\underline{0.982}_{\pm0.013}$ | $0.941_{\pm0.040}$ | $0.926_{\pm0.047}$ | $0.611_{\pm0.172}$ | $0.851_{\pm0.070}$ |
| TabDiff | $\mathbf{0.995}_{\pm0.001}$ | $0.973_{\pm0.004}$ | $0.975_{\pm0.007}$ | $0.826_{\pm0.010}$ | $\mathbf{0.972}_{\pm0.008}$ | $\underline{0.983}_{\pm0.006}$ |
| CDTD | $\underline{0.993}_{\pm0.002}$ | $\mathbf{0.993}_{\pm0.003}$ | $\underline{0.978}_{\pm0.005}$ | $\underline{0.979}_{\pm0.013}$ | $0.851_{\pm0.012}$ | $0.982_{\pm0.006}$ |
| TabCascade (DT) | $0.981_{\pm0.003}$ | $0.980_{\pm0.007}$ | $\mathbf{0.986}_{\pm0.003}$ | $\mathbf{0.993}_{\pm0.004}$ | $0.907_{\pm0.008}$ | $\mathbf{0.987}_{\pm0.007}$ |

Table 15: Comparison of $\beta$-**Recall scores**. Per dataset, **bold** indicates the best and underline the second best result. We report the average (and standard deviation) across 3 training runs and 10 different generated samples each.

| | adult | beijing | default | diabetes | news | shoppers |
|---|---|---|---|---|---|---|
| ARF | $0.420_{\pm 0.004}$ | $0.285_{\pm 0.006}$ | $0.362_{\pm 0.007}$ | $0.329_{\pm 0.006}$ | $0.114_{\pm 0.003}$ | $0.423_{\pm 0.006}$ |
| TVAE | $0.196_{\pm 0.018}$ | $0.098_{\pm 0.058}$ | $0.247_{\pm 0.032}$ | $0.179_{\pm 0.079}$ | $0.049_{\pm 0.025}$ | $0.483_{\pm 0.010}$ |
| CTGAN | $0.162_{\pm 0.039}$ | $0.148_{\pm 0.013}$ | $0.304_{\pm 0.032}$ | $0.178_{\pm 0.063}$ | $0.339_{\pm 0.031}$ | $0.326_{\pm 0.016}$ |
| TabDDPM | $0.525_{\pm 0.008}$ | $0.396_{\pm 0.007}$ | $0.553_{\pm 0.004}$ | - | - | $0.664_{\pm 0.024}$ |
| TabSyn | $0.397_{\pm 0.014}$ | $0.311_{\pm 0.019}$ | $0.346_{\pm 0.026}$ | $0.176_{\pm 0.024}$ | $0.035_{\pm 0.016}$ | $0.301_{\pm 0.056}$ |
| TabDiff | $0.477_{\pm 0.003}$ | $0.373_{\pm 0.007}$ | $0.482_{\pm 0.005}$ | $0.274_{\pm 0.010}$ | $0.366_{\pm 0.018}$ | $0.476_{\pm 0.007}$ |
| CDTD | $0.573_{\pm 0.004}$ | $0.441_{\pm 0.006}$ | **$0.603_{\pm 0.008}$** | **$0.561_{\pm 0.017}$** | **$0.517_{\pm 0.010}$** | **$0.728_{\pm 0.007}$** |
| TabCascade (DT) | **$0.595_{\pm 0.009}$** | **$0.540_{\pm 0.004}$** | $0.562_{\pm 0.006}$ | $0.517_{\pm 0.004}$ | $0.478_{\pm 0.010}$ | $0.688_{\pm 0.007}$ |

Table 16: Comparison of **DCR share scores**. Per dataset, **bold** indicates the best and underline the second best result. We report the average (and standard deviation) across 3 training runs and 10 different generated samples each.

| | adult | beijing | default | diabetes | news | shoppers |
|---|---|---|---|---|---|---|
| ARF | $0.815_{\pm 0.002}$ | $0.801_{\pm 0.002}$ | $0.793_{\pm 0.004}$ | $0.806_{\pm 0.002}$ | $0.785_{\pm 0.003}$ | $0.800_{\pm 0.004}$ |
| TVAE | $0.800_{\pm 0.004}$ | $0.816_{\pm 0.019}$ | $0.792_{\pm 0.016}$ | $0.787_{\pm 0.009}$ | $0.815_{\pm 0.015}$ | $0.817_{\pm 0.006}$ |
| CTGAN | $0.781_{\pm 0.003}$ | **$0.777_{\pm 0.013}$** | $0.783_{\pm 0.003}$ | $0.778_{\pm 0.004}$ | $0.783_{\pm 0.002}$ | $0.784_{\pm 0.006}$ |
| TabDDPM | $0.799_{\pm 0.005}$ | $0.795_{\pm 0.002}$ | $0.800_{\pm 0.003}$ | - | - | $0.856_{\pm 0.020}$ |
| TabSyn | **$0.780_{\pm 0.003}$** | $0.780_{\pm 0.003}$ | **$0.780_{\pm 0.004}$** | **$0.775_{\pm 0.002}$** | **$0.780_{\pm 0.005}$** | **$0.780_{\pm 0.005}$** |
| TabDiff | $0.786_{\pm 0.003}$ | $0.787_{\pm 0.003}$ | $0.786_{\pm 0.003}$ | $0.777_{\pm 0.002}$ | $0.782_{\pm 0.002}$ | $0.782_{\pm 0.005}$ |
| CDTD | $0.863_{\pm 0.002}$ | $0.823_{\pm 0.002}$ | $0.851_{\pm 0.005}$ | $0.837_{\pm 0.002}$ | $0.818_{\pm 0.004}$ | $0.955_{\pm 0.004}$ |
| TabCascade (DT) | $0.871_{\pm 0.006}$ | $0.845_{\pm 0.002}$ | $0.839_{\pm 0.004}$ | $0.799_{\pm 0.002}$ | $0.805_{\pm 0.003}$ | $0.937_{\pm 0.004}$ |

Table 17: Comparison of **MIA scores**. Per dataset, **bold** indicates the best and underline the second best result. We report the average (and standard deviation) across 3 training runs and 10 different generated samples each.

| | adult | beijing | default | diabetes | news | shoppers |
|---|---|---|---|---|---|---|
| ARF | $0.977_{\pm 0.009}$ | **$0.994_{\pm 0.005}$** | $0.978_{\pm 0.010}$ | $0.995_{\pm 0.004}$ | $0.993_{\pm 0.006}$ | $0.985_{\pm 0.008}$ |
| TVAE | $0.987_{\pm 0.006}$ | $0.991_{\pm 0.006}$ | $0.980_{\pm 0.010}$ | $0.994_{\pm 0.004}$ | $0.990_{\pm 0.008}$ | $0.980_{\pm 0.012}$ |
| CTGAN | **$0.994_{\pm 0.004}$** | $0.992_{\pm 0.005}$ | **$0.988_{\pm 0.006}$** | $0.992_{\pm 0.004}$ | $0.991_{\pm 0.005}$ | $0.988_{\pm 0.012}$ |
| TabDDPM | $0.971_{\pm 0.010}$ | $0.992_{\pm 0.006}$ | $0.975_{\pm 0.010}$ | - | - | $0.979_{\pm 0.012}$ |
| TabSyn | $0.985_{\pm 0.007}$ | $0.994_{\pm 0.005}$ | $0.978_{\pm 0.005}$ | $0.995_{\pm 0.003}$ | **$0.993_{\pm 0.005}$** | **$0.990_{\pm 0.008}$** |
| TabDiff | $0.981_{\pm 0.006}$ | $0.993_{\pm 0.005}$ | $0.975_{\pm 0.009}$ | $0.995_{\pm 0.003}$ | $0.991_{\pm 0.005}$ | $0.985_{\pm 0.010}$ |
| CDTD | $0.969_{\pm 0.010}$ | $0.992_{\pm 0.005}$ | $0.979_{\pm 0.008}$ | **$0.996_{\pm 0.003}$** | $0.992_{\pm 0.005}$ | $0.974_{\pm 0.011}$ |
| TabCascade (DT) | $0.960_{\pm 0.008}$ | $0.991_{\pm 0.006}$ | $0.949_{\pm 0.010}$ | $0.980_{\pm 0.004}$ | $0.990_{\pm 0.006}$ | $0.956_{\pm 0.014}$ |

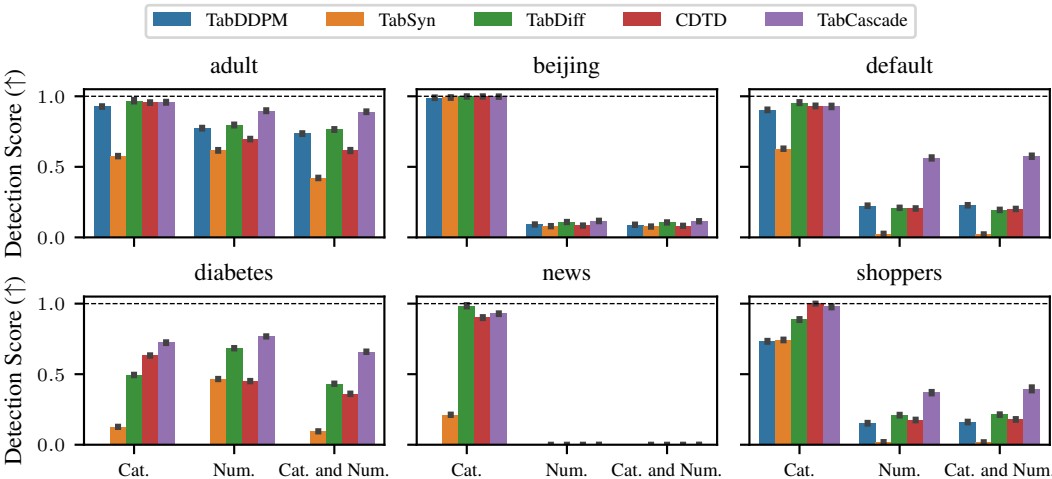

Figure 16: Detection scores for all diffusion-based models and all datasets. The Cat. score considers only categorical features, the Num. score only numerical features.

## K  FURTHER ABLATION EXPERIMENTS ON THE DT ENCODER COMPLEXITY

We thoroughly investigate the effect of the complexity of the DT encoder. Specifically, we vary the maximum depth of the DT encoder from 3 to 9. Figure 18 shows the impact of increasing max depth on the proportion of masked inputs to the high-resolution model. For comparison, Figure 17 shows the same for increasing the complexity of the GMM encoder. For features that are integer-valued with few unique values, increasing max depth can lead to cases where each unique value is treated as a separate component. In these cases, the feature would be entirely generated by the low-resolution model.

Further, we investigate the effect of max depth on various sample quality metrics. Table 18 gives the average results over all datasets with 10 different synthetic samples each. For each setting, we adjusted the model parameters to ≈1 million parameters for the high-resolution model and ≈2 million parameters for the low-resolution model on the `adult` dataset. We emphasize that the effect of max depth may be different for different architectures but an exhaustive evaluation of all combinations is prohibitively expensive.

Increasing max depth increases the number of Gaussian components. This appears to make samples substantially more realistic in the eyes of the gradient-boosting-based detection model whereas it has a less pronounced effect on the other metrics. The best choice for max depth also depends on which metrics are deemed to be most relevant in a given modeling context. If, for instance, $\alpha$-Precision and $\beta$-Recall are presumed to be important than the Detection Score, than more favorable results could be achieved by lowering max depth to 5.

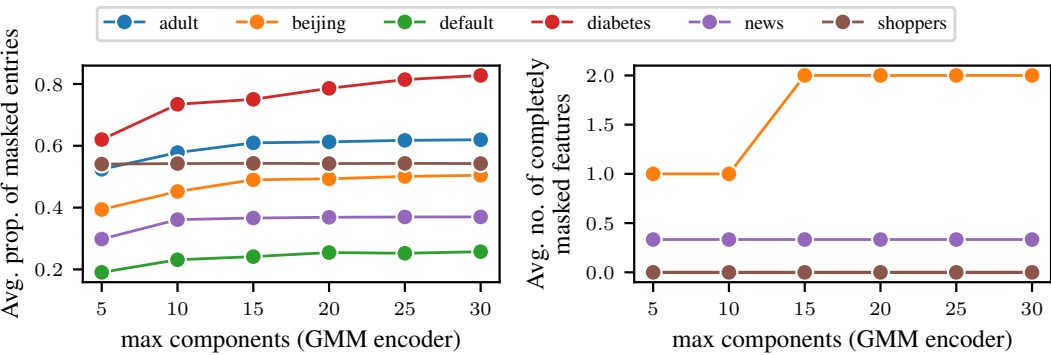

Figure 17: Effect of increasing the maximum possible number of Gaussian components on the average (over three training seeds) proportion of masked inputs to $p_{\text{high}}^{\theta}$ and the average number of completely masked features.

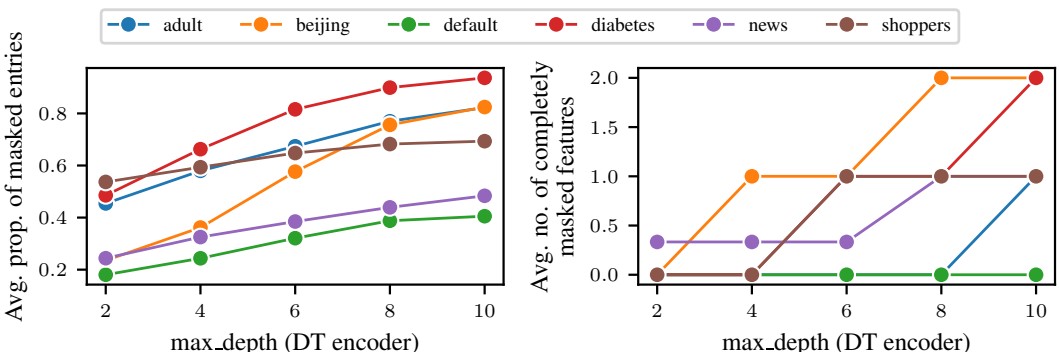

Figure 18: Effect of increasing the maximum tree depth on the average (over three training seeds) proportion of masked inputs to $p_{\text{high}}^{\theta}$ and the average number of completely masked features.

Table 18: The effect of max depth for the DT encoder on various evaluation metrics averaged over datasets. The standard deviation captures variance across the datasets. Grey indicates the max depth used for the main results.

| Max. Depth | Shape (num) | WD (num) | Trend | Trend (mixed) | Detection Score | MLE | $\alpha$-Precision | $\beta$-Recall | DCR Share |
|---|---|---|---|---|---|---|---|---|---|
| 3 | $0.969_{\pm 0.021}$ | $0.007_{\pm 0.006}$ | $0.964_{\pm 0.029}$ | $0.937_{\pm 0.050}$ | $0.391_{\pm 0.308}$ | $0.019_{\pm 0.014}$ | $0.974_{\pm 0.024}$ | $0.583_{\pm 0.068}$ | $0.866_{\pm 0.062}$ |
| 4 | $0.972_{\pm 0.019}$ | $0.006_{\pm 0.006}$ | $0.962_{\pm 0.029}$ | $0.935_{\pm 0.051}$ | $0.391_{\pm 0.301}$ | $0.018_{\pm 0.015}$ | $0.985_{\pm 0.011}$ | $0.579_{\pm 0.069}$ | $0.859_{\pm 0.058}$ |
| 5 | $0.973_{\pm 0.018}$ | $0.006_{\pm 0.006}$ | $0.962_{\pm 0.026}$ | $0.934_{\pm 0.048}$ | $0.391_{\pm 0.295}$ | $0.027_{\pm 0.020}$ | $0.986_{\pm 0.012}$ | $0.575_{\pm 0.072}$ | $0.851_{\pm 0.052}$ |
| 6 | $0.974_{\pm 0.017}$ | $0.006_{\pm 0.006}$ | $0.962_{\pm 0.025}$ | $0.933_{\pm 0.048}$ | $0.418_{\pm 0.322}$ | $0.026_{\pm 0.019}$ | $0.985_{\pm 0.010}$ | $0.566_{\pm 0.074}$ | $0.848_{\pm 0.052}$ |
| 7 | $0.974_{\pm 0.019}$ | $0.006_{\pm 0.006}$ | $0.967_{\pm 0.019}$ | $0.939_{\pm 0.042}$ | $0.416_{\pm 0.325}$ | $0.027_{\pm 0.023}$ | $0.981_{\pm 0.014}$ | $0.568_{\pm 0.073}$ | $0.849_{\pm 0.052}$ |
| 8 | $0.975_{\pm 0.020}$ | $0.006_{\pm 0.007}$ | $0.969_{\pm 0.018}$ | $0.943_{\pm 0.037}$ | $0.446_{\pm 0.350}$ | $0.017_{\pm 0.013}$ | $0.972_{\pm 0.035}$ | $0.563_{\pm 0.077}$ | $0.849_{\pm 0.050}$ |
| 9 | $0.973_{\pm 0.020}$ | $0.006_{\pm 0.006}$ | $0.968_{\pm 0.018}$ | $0.942_{\pm 0.038}$ | $0.436_{\pm 0.337}$ | $0.016_{\pm 0.014}$ | $0.966_{\pm 0.042}$ | $0.559_{\pm 0.079}$ | $0.849_{\pm 0.049}$ |

## L   TRAINING AND SAMPLING TIMES

Table 19: Training times in minutes. For diffusion-based models, the training time was capped at 30 minutes.

|  | ARF | TVAE | CTGAN | TabDDPM | TabSyn | TabDiff | CDTD | TabCascade (DT) | TabCascade (GMM) |
|---|---|---|---|---|---|---|---|---|---|
| adult | 11.4 | 20.0 | 36.2 | 9.5 | 14.4 | 30.0 | 6.0 | 10.7 | 11.2 |
| beijing | 10.6 | 21.5 | 35.3 | 8.1 | 13.2 | 30.0 | 5.6 | 11.4 | 11.2 |
| default | 14.7 | 25.1 | 44.1 | 11.9 | 19.4 | 30.0 | 6.6 | 11.7 | 11.9 |
| diabetes | 56.0 | 29.5 | 101.8 | 30.0 | 16.2 | 30.0 | 8.0 | 12.6 | 13.7 |
| news | 38.7 | 41.7 | 68.2 | 21.1 | 30.0 | 30.0 | 9.2 | 17.1 | 16.9 |
| shoppers | 3.6 | 24.1 | 39.2 | 10.4 | 14.3 | 30.0 | 6.2 | 11.1 | 11.1 |

Table 20: Sample times in seconds per 1000 samples. TabDDPM produces NaNs for diabetes and news datasets.

|  | ARF | TVAE | CTGAN | TabDDPM | TabSyn | TabDiff | CDTD | TabCascade (DT) | TabCascade (GMM) |
|---|---|---|---|---|---|---|---|---|---|
| adult | 1.55 | 0.14 | 0.24 | 7.08 | 0.53 | 3.62 | 2.55 | 0.69 | 2.06 |
| beijing | 1.09 | 0.14 | 0.23 | 5.32 | 0.55 | 2.24 | 3.76 | 0.62 | 2.09 |
| default | 2.43 | 0.18 | 0.29 | 10.19 | 0.56 | 3.47 | 6.38 | 0.76 | 2.14 |
| diabetes | 4.46 | 0.22 | 0.32 | - | 0.53 | 24.11 | 3.54 | 0.87 | 2.26 |
| news | 6.76 | 0.34 | 0.44 | - | 0.60 | 7.69 | 5.26 | 1.17 | 2.52 |
| shoppers | 1.71 | 0.18 | 0.25 | 7.45 | 0.54 | 3.20 | 2.90 | 0.70 | 2.05 |

