# OpenReview forum: "Cascaded Flow Matching for Heterogeneous Tabular Data with Mixed-Type Features"
_ICLR.cc/2026/Conference — Submitted to ICLR 2026_

### Official Review · Reviewer_oVg1 · 2025-10-26

**Soundness:** 2
**Presentation:** 2
**Contribution:** 2
**Rating:** 4
**Confidence:** 4

**Summary:**

This paper's main aim is to answer the research question “How can we generate realistic heterogeneous tabular data that includes mixed-type numerical features (continuous values with discrete point masses such as missings or inflated values), by reducing transport costs and improving fidelity via a cascaded flow-matching framework that leverages low-resolution information?”

**Strengths:**

1. **[Important] Novel model design.** Cascaded factorisation of tabular generation. The paper proposes TabCascade, which factors the joint into a low-resolution model and a high-resolution flow-matching model, which is conceptually attractive.
2. **Theoretical claim.** With a DT encoder, the authors seem to provide convincing evidence that data-dependent coupling lowers an upper bound on transport cost compared to independent couplings.
3. **[Important] Empirical gains on standard metrics.** On six datasets, TabCascade (DT) achieves state-of-the-art Detection scores (C2ST) and competitive/better Shape/Trend scores.

**Weaknesses:**

1. **[Important] Two-stage factorisation may under-capture cross-type dependencies.** Since $x_{\text{cat}}$ and $x_{\text{num}}$ do not seem to be generated jointly (high-res is conditioned on low-res outputs), subtle dependencies might be missed.
2. **[Important] Limited coverage of benchmark generators.** Many competitive models are missing in the current results, such as foundation models, CTSyn [1] and TabPFN [2]. I would suggest the authors refer to relevant literature [3, 4] for a broader context of the benchmark setups. Existing coverage seems limited to reach conclusive results.
3. **Privacy is only analysed superficially (DCR share) and no guarantees are claimed.** For sensitive tabular data, the absence of *any* privacy mechanism or DP ablation limits practical adoption.
4. **Metric dependence and detector sensitivity.** The strong wins are most pronounced for Detection score (gradient-boosted C2ST); while Shape/Trend also improves, they are already near ceiling for many baselines. This raises questions about how general the gains are across orthogonal metrics and downstream utility.

[1] Lin, Xiaofeng, et al. "Ctsyn: A foundational model for cross-tabular data generation." *arXiv preprint arXiv:2406.04619* (2024).

[2] Hollmann, Noah, et al. "Accurate predictions on small data with a tabular foundation model." *Nature* 637.8045 (2025): 319-326.

[3] Ma, Junwei, et al. "TabPFGen--Tabular Data Generation with TabPFN." *arXiv preprint arXiv:2406.05216* (2024).

[4] Margeloiu, Andrei, et al. "Tabebm: A tabular data augmentation method with distinct class-specific energy-based models." *Advances in Neural Information Processing Systems* 37 (2024): 72094-72144.

**Questions:**

Please refer to "Weaknesses" section.

---

> ### Author Response · Authors · 2025-11-21
>
> > Strengths: model design, theoretical claim, empirical gains
>
> Thank you for your positive evaluation on these three strengths of the paper.
>
> >  Since $x_{\text{cat}}$ and $x_{\text{num}}$ do not seem to be generated jointly (high-res is conditioned on low-res outputs), subtle dependencies might be missed.
>
> Perhaps a part of our exposition was not clear enough, but both parts of the data are generated from their joint distribution. We use a valid factorization of this joint into a marginal distribution for the categorical part of the data and a conditional distribution for the numerical part. All dependencies are still present in this factorization
>
> The presence of the latent variable $z_{\text{num}}$ complicates the discussion a bit. It captures some information about $x_{\text{num}}$ and is generated jointly with $x_{\text{cat}}$. Therefore, some dependencies are captured in the first part of the cascade by the dependencies between $x_{\text{num}}$ and $z_{\text{num}}$. It is true, that more subtle dependencies are not captured in this stage. However, the high-resolution model $p_{\text{high}}$ will next learn them. Overall, we have a valid factorization of the joint density: $p(x_{\text{cat}}, x_{\text{num}} ) = \sum_z p(x_{\text{num}} | x_{\text{cat}}, z_{\text{num}}) p(x_{\text{cat}}, z_{\text{num}})$.
>
> We added a Trend (mixed) metric to the main results in Table 1 that investigates this issue, as it only evaluates dependencies between categorical and numerical features. TabCascade also performs very well on this metric compared to the baselines.
>
> > Many competitive models are missing in the current results, such as foundation models, CTSyn [1] and TabPFN [2]. I would suggest the authors refer to relevant literature [3, 4] for a broader context of the benchmark setups. Existing coverage seems limited to reach conclusive results.
>
> We strongly disagree that our benchmark models were chosen to reach conclusive results. Note in particular that CDTD and TabDiff are very recent publications in ICLR 2025 and are clearly considered SOTA. None of papers on diffusion models for tabular data benchmark against foundational models. They are generally considered as a different model class and therefore not directly comparable. Similar to previous work, we do not believe that it is reasonable to compare a foundational model that has been trained for weeks to diffusion models which are trained from scratch in a fraction of the time. They fundamentally serve a different purpose. The scope of this paper is the improvement of *diffusion-based* generative models, and our method has merit within that context. It is not our aim to contribute an improvement in the context of foundational models. Lastly, while TabEBM [1] looks interesting, this is a *class-conditioned* model and cannot be directly or fairly compared to the unconditional generation models in our paper.
>
> - [1] Margeloiu, Andrei, et al. "Tabebm: A tabular data augmentation method with distinct class-specific energy-based models." Advances in Neural Information Processing Systems 37 (2024): 72094-72144.
>
> > For sensitive tabular data, the absence of any privacy mechanism or DP ablation limits practical adoption.
>
> Thanks for bringing up this important point. Privacy guarantees are not our goal in this paper.
>
> We note that none of the previous papers that proposed our benchmark models directly target privacy, even though all of them propose generative models for tabular data. Second, we avoid making claims about privacy. Guarding against privacy is typically considered a separate task, warranting additional mechanisms such as differential privacy, etc., which are highly context-specific. These techniques can be applied to any model, and therefore also ours. Specific choices will depend on ones definition of privacy, the modeling problem, and its constraints at hand.
>
> That being said, we added membership inference attack (MIA) scores to the main results in Table 1 to provide a more suitable privacy metric. We re-framed DCR share as a metric about sample diversity instead. Section 5.2 now also includes a short discussion on privacy concerns.

---

> ### Author Response · Authors · 2025-11-21
>
> > The strong wins are most pronounced for Detection score (gradient-boosted C2ST); while Shape/Trend also improves, they are already near ceiling for many baselines. This raises questions about how general the gains are across orthogonal metrics and downstream utility.
>
> We agree in that there is a need for better metrics for synthetic tabular data evaluation. However, we have little choice here: If we would have excluded Shape and Trend results, then we would be missing some very popular metrics in the literature. Therefore, we decided to emphasize the role of the detection score, which actually evaluates the whole joint distribution. Table 1 now displays the expected performance across metrics, making more average gains of our framework visible. Downstream utility is evaluated using machine learning efficiency (MLE), a very common metric in the literature. This is also improved by our model's focus on the details of the data distribution, which produces more accurate (but therefore less private) samples.
>
> ----
> We thank you for all your comments, and we believe that we have improved the paper by making the various changes mentioned above. We hope that we sufficiently answered your questions and you are willing to update your scores for this paper. Please let us know in case some points are still unclear, or if further changes are needed.

---

> ### Comment · Reviewer_oVg1 · 2025-11-26
>
> I would like to thank the authors for the effort they have put into their responses. I have carefully reviewed their replies, as well as the discussion with the other reviewers.
>
> With respect to my initial comments, the authors’ response has addressed most of them. I only have a few remaining points that may require further clarification.
>
> 1. Many competitive models are missing in the current results
>
> I generally agree with the reviewer that foundation models may be beyond the context of their work. However, I have to kindly note that the existing coverage remains somewhat limited. The authors argue that "class-conditioned" models are unfair to compare with. However, if I understand correctly, the currently included TVAE and CTGAN are class-conditioned as well, which are trained with the target feature as the condition. Therefore, I am unsure if the reasons convince me.
>
> In addition, my intention in citing those references was not to insist that the authors include specific methods such as TabPFN/CTSyn/TabEBM. Instead, I wanted to draw the authors’ attention to the broader literature on tabular generators. Beyond the methods already mentioned, there are even more lightweight yet highly competitive generators that could be considered, such as SMOTE [1].
>
> To be clear, I fully understand that, given the limited timeframe, it is practically difficult to incorporate additional methods. Therefore, this point did not substantially affect my overall evaluation of the paper. I would strongly encourage the authors to explicitly acknowledge this limitation in the manuscript and leave more extensive comparisons to future work.
>
> [1] Chawla, Nitesh V., et al. "SMOTE: synthetic minority over-sampling technique." Journal of artificial intelligence research 16 (2002): 321-357.
>
> 2. while Shape/Trend also improves, they are already near the ceiling for many baselines
>
> I believe there may have been some misunderstanding of my original comments. I was not disregarding Shape/Trend metrics due to the ceiling performance. Instead, I do see their unique values, and my point was to expand the evaluations to more challenging datasets. As the performance gains on ceiling performance could be very limited in distinguishing truly stronger/weaker models. The "challenging datasets" bit has been done by some very well-established benchmarks in this domain, like TabZilla [2] and TabArena [3].
>
> Again, I am not asking the authors to provide a benchmark-level evaluation of their method, which is, for sure, impractical. However, given the current ceiling performance of almost all the methods, it is really prohibitive to reach the conclusion for "which are better models". This is just a suggestion in my humble opinion, as the current results are not strong enough to support the authors' claim of the proposed model.
>
> [2] McElfresh, Duncan, et al. "When do neural nets outperform boosted trees on tabular data?." Advances in Neural Information Processing Systems 36 (2023): 76336-76369.
>
> [3] Erickson, Nick, et al. "Tabarena: A living benchmark for machine learning on tabular data." arXiv preprint arXiv:2506.16791 (2025).
>
> Overall, I would like to thank the authors once again for their efforts in engaging with the discussion. I believe the above points are important for the strength of the work and the validity of its claims.

---

> > ### Author Response · Authors · 2025-11-27
> >
> > Thank you for your response and further comments. We discuss them below.
> >
> > ---
> >
> > > I generally agree with the reviewer that foundation models may be beyond the context of their work. However, I have to kindly note that the existing coverage remains somewhat limited. The authors argue that "class-conditioned" models are unfair to compare with. However, if I understand correctly, the currently included TVAE and CTGAN are class-conditioned as well, which are trained with the target feature as the condition. Therefore, I am unsure if the reasons convince me.
> >
> > Yes, the CTGAN is in principle a conditional model. However, details matter here: The CTGAN conditioning is *random*. That is, at each training step, a new selection of conditioning features are sampled (one for each observation in the batch (see Algorithm 1 in [1]). This is fundamentally different from a conditional diffusion model, for example, such a model would typically pick a fixed target feature $y$ and condition on these values throughout the whole training process. In these cases, you would have to decide on the conditioning feature, and the choice can have a dramatic impact on the model's performance. Therefore, we, for example, did not use the original TabDDPM setup (which is conditional on $y$) but its unconditional variant. This makes the benchmarks comparable by eliminating the (arbitrary) choice of conditioning variable. Due to the CTGAN randomization technique during training, one does not have to make similar choices for that case. The TVAE model is unconditional. The encoder takes the form $q(z_j | \rvr_j)$, where $\rvr_j$ is a complete data row (see page 6 in [1]).
> >
> > - [1] Modeling Tabular Data using Conditional GAN.
> >
> > > In addition, my intention in citing those references was not to insist that the authors include specific methods such as TabPFN/CTSyn/TabEBM. Instead, I wanted to draw the authors’ attention to the broader literature on tabular generators.
> >
> > This was indeed not clear to us, we are sorry for the confusion. We are happy to add these models (and additional references) as related work in the final document. However, depending on the length, since these are not directly relevant to our model class, we may need to move the corresponding subsection into the appendix.
> >
> > > I would strongly encourage the authors to explicitly acknowledge this limitation in the manuscript and leave more extensive comparisons to future work.
> >
> > We updated the document (see new text in red) to acknowledge the existence of the foundational models in the section on our selected baselines. We also mention SMOTE. Note, however, that previous work has shown that SMOTE becomes inefficient very quickly when increasing the number of samples or features, due to its reliance on nearest neighbor computations. For evidence, we refer to Table 29 in [2]. Even on smaller datasets, such as the adult dataset, the sampling time is considerable. Since we simulate missings, we would need to add many categorical features to set of variables, which would prolong sampling for SMOTE quite a bit on our benchmark datasets, making it much less attractive in practice. This combined with the earlier documented poor performance (see Table 1 in [2]) made us decide to exclude SMOTE as a benchmark. We have added this motivation to the paper.
> >
> >
> > - [2] Continuous Diffusion for Mixed-Type Tabular Data.

---

> ### Author Response · Authors · 2025-11-27
>
> > [...] my point was to expand the evaluations to more challenging datasets. As the performance gains on ceiling performance could be very limited in distinguishing truly stronger/weaker models.
>
> We are sorry that we missed the point here. There is indeed not much improvement possible if a metric is near its ceiling value. Note however, that this does not hold for all metrics we use, especially the detection score.
>
> Please note that datasets that are are difficult in a regression or classification setting, do not necessarily have worse Shape scores when it comes to data generation. Similar to WD and JSD scores, regression and classification focus on univariate (conditional) densities. We expect those univariate densities to be rather accurate and have corresponding metrics close to the ceiling for all recent generative models, regardless of the dataset. However, what may become more challenging with different datasets is for the model to learn *all* dependencies accurately, i.e., to learn the joint density. In our opinion, the most valuable metric to evaluate the learned joint density is the detection score, which we report and on which we clearly outperform the baselines. We want to emphasize that we used the most popular benchmark datasets in the field (please see the data collection section at [https://github.com/Diffusion-Model-Leiden/awesome-diffusion-models-for-tabular-data](https://github.com/Diffusion-Model-Leiden/awesome-diffusion-models-for-tabular-data). TabArena, for example, includes the diabetes dataset and the shoppers dataset we are using. This shows that we picked these dataset due to their popularity, not due to our model's performance on those. Not all included datasets in that paper would be eligible for our framework. We require both continuous *and* categorical features, as we focus on mixed-type data generation (our model can also work on only continuous features but this does not hold for all baselines). We also focus on datasets with >10k observations due to the large size of the neural network backend. These criteria would only make 18/51 datasets of the TabArena benchmark eligible. That being said, we have mentioned our limited data selection as a limitation towards the end of the paper, referring to your comment on "near-ceiling" performance on the current selection.
>
> > This is just a suggestion in my humble opinion, as the current results are not strong enough to support the authors' claim of the proposed model.
>
> We do still believe that our claim is correct. First, notice that TabCascade outperforms the baselines on the detection metric \textit{by a lot}, not just marginally. Second, previous work used a comparable number of datasets. For instance,
>
> - TabDDPM [3] uses 9 mixed-type datasets,
> - TabSyn [4] uses 6,
> - TabDiff [5] uses 7,
> - Tabbyflow [6] uses 6.
>
> Naturally, the datasets that are popular to benchmark against gradient boosting algorithms need not be the same that are popular in the field of generating synthetic tabular data.
>
> - [3] Kotelnikov, et al. TabDDPM: Modelling Tabular Data with Diffusion Models.
> - [4] Zhang, et al. Mixed-Type Tabular Data Synthesis with Score-based Diffusion in Latent Space.
> - [5] Shi, et al. TabDiff: a Mixed-type Diffusion Model for Tabular Data Generation.
> - [6] Guzmán-Cordero, et al. Exponential Family Variational Flow Matching for Tabular Data Generation.
>
> ---
>
> We thank you again for your time spent reviewing our work and making valuable comments. **We updated our document once again** and believe that our paper has improved due to this review process. It would be very helpful to us if you could specifically **point us towards further changes you would like us to make to the document, if any, that would induce you to increase your score.**

---

### Official Review · Reviewer_5gde · 2025-10-29

**Soundness:** 2
**Presentation:** 3
**Contribution:** 2
**Rating:** 2
**Confidence:** 4

**Summary:**

The paper introduces Cascaded Flow Matching, where the authors leverage ideas from cascaded diffusion and apply it to flow matching by first generating low resolution features (categorical features) then generating high resolution information (continuous features) conditioned on the generated categorical features. Experiments highlight that their generative paradigm provides competitive results.

**Strengths:**

**Originality**. The paper integrate concepts from cascaded diffusion onto tabular data. As far as I know, this is the first paper that applies a multi-resolution generation concept to tabular data (low = categorical and high = numerical).

**Quality**. Idea is interesting and results demonstrated are competitive.

**Clarity**. Overall, the paper is easy to follow as it conveys the message and method well.

**Significance**. Tabular data generation is important in many aspects such as privacy preservation.

**Weaknesses:**

In the work, the authors claim that:"this is the first work to address mixed-type feature generation, i.e., features following a mixture of categorical and continuous distributions". However, this is not expressed very clearly. There are methods that unify the data representation including TabRep [1] that explores various encoding, TabbyFlow [4] that represents heterogeneous data types using a general exponential family distribution, TabSYN that projects the data onto a latent space, and StaSy that applies continuous diffusion to a unified data space via one-hot encoding.

The results seem to be underwhelming compared to the baselines reported in the paper. Not-so-recent baselines including TabRep and TabbyFlow that explores flow matching on tabular data should also be included for comparisons. Considering that TabRep-Flow and TabbyFlow both outperform TabDiff across the board, the margins for TabCascade will shrink.

What is the motivation of using flow matching in the framework? TabRep and TabbyFlow leverages its sampling speed. If so, are there experiments to demonstrate this?

One of the most important use-cases of tabular relational data generation is privacy preservation. DCR experiments are conducted and demonstrates that it underperforms against its competitors. Additionally, existing literature in tabular data generation [1] and computational privacy [2] [3] have also highlighted the inadequacy of DCR in evaluating privacy preservation. Hence, its important to assess the privacy preservation via Membership Inference Attacks too.

[1] Si, Jacob, et al. "TabRep: Training Tabular Diffusion Models with a Simple and Effective Continuous Representation." arXiv preprint arXiv:2504.04798 (2025).

[2] Georgi Ganev and Emiliano De Cristofaro. The inadequacy of similarity-based privacy metrics: Privacy attacks against "truly anonymous" synthetic datasets, 2024.

[3] Joshua Ward, Chi-Hua Wang, and Guang Cheng. Data plagiarism index: Characterizing the privacy risk of data-copying in tabular generative models, 2024.

[4] Guzmán-Cordero, Andrés, Floor Eijkelboom, and Jan-Willem van de Meent. "Exponential Family Variational Flow Matching for Tabular Data Generation."

**Questions:**

Please see weaknesses.

---

> ### Author Response · Authors · 2025-11-21
>
> > Strengths: originality, quality, clarity and significance
>
> Thank you for your positive evaluation of our paper on these aspects.
>
> > In the work, the authors claim that: "this is the first work to address mixed-type feature generation, i.e., features following a mixture of categorical and continuous distributions". However, this is not expressed very clearly.
>
> Please note that mixed-type *data*, i.e., a dataset where some features are categorical and some are numerical, differs from a mixed-type *feature*, i.e., a single feature whose distribution is a mixture of a categorical and a continuous distribution. We now expand upon this point in the problem statement. We acknowledge that this choice of words is unfortunate and adjusted the text in the contribution part of the introduction slightly to make this point clearer. We also added a new Figure 1 to illustrate the concept of a mixed-type feature distribution.
>
> That being said, the models you mentioned can deal with mixed-type *data* but not explicitly with mixed-type *features*. For instance, TabSyn is able to encode a mixed-type feature into a latent space but the decoder will not know anything about the underlying structure, i.e., it being a mixture of a discrete and a continuous distribution. For that reason, the existing benchmark models, without any adjustments, cannot generate missings or account for inflated values in numerical features, which is one of the main contributions of our model.
>
>
> > The results seem to be underwhelming compared to the baselines reported in the paper.
>
> It is unclear to us to which results you are referring here. To make the performance of our model easier to grasp, we reworked the result section. We change the main result table (see Table 1) to report averages over datasets, training seeds and sampling seeds (in accordance with the request of another reviewer). TabCascade leads to considerable improvements in terms of average performance. The detection scores and accuracy of the numerical univariate densities in particular are SOTA. There are, however, trade-offs, which we also illustrate. In particular, our cascaded model implies that dependencies between $x_{\text{num}}$ and $x_{\text{cat}}$ become more difficult to learn. This is reflected in a slightly lower Trend (mixed) score on average. A more powerful generative model also necessarily leads to a lower DCR share, as it is expected to generate more accurate samples, thereby reducing the distance to closest training records. This is a direct consequence of our model being designed to also fit the details of continuous distributions (see Figure 3).

---

> ### Author Response · Authors · 2025-11-21
>
> > Not-so-recent baselines including TabRep and TabbyFlow that explores flow matching on tabular data should also be included for comparisons. Considering that TabRep-Flow and TabbyFlow both outperform TabDiff across the board, the margins for TabCascade will shrink.
>
> Please note that TabbyFlow does *not* outperform TabDiff in general, e.g. only very marginally in their Table 2. Also, TabbyFlow barely outperforms TabSyn in Table 3 in [1] and is actually outperformed by TabDiif in Table 4. Thus, the statement that TabbyFlow outperforms TabDiff across the board is not correct. Also, the devil lies in the details. TabbyFlow uses a transformer backend as well as a [Runge-Kutta ODE solver](https://github.com/andresguzco/ef-vfm/blob/40a4b96c850169c1611231783a7f6250d58afc9e/ef_vfm/models/flow_model.py#L97) instead of an Euler sampler. Hence, much of the gain compared to, e.g., TabDDPM and TabSyn, can be tracked to a difference in backends and solvers. This is not a fair comparison. Lastly, we are asking the generative models to also generate missingness indicators. This leads to a more complex (and higher-dimensional) data distribution and therefore complicates the task of learning a generative model. The performance of models in other papers therefore does not directly translate to our results.
>
> We tried to include Tabbyflow as an additional benchmark. While training works well, the sampling code that is available on GitHub is faulty. Without any changes, sampling is simply not possible. With minor changes to the code, the sampling procedure seems to work but the sample quality is abysmal. These problems are already documented in GitHub issues by other users, see [https://github.com/andresguzco/ef-vfm/issues/1](https://github.com/andresguzco/ef-vfm/issues/1) and [https://github.com/andresguzco/ef-vfm/issues/2](https://github.com/andresguzco/ef-vfm/issues/2). Therefore, we can currently not add [1] to the paper. We contacted the authors about solving the issues with their available code.
>
> Regarding TabRep, this is not a published paper but only a workshop submission and as such not thoroughly peer-reviewed. Relative to a vanilla flow matching model, it changes the encoding of categorical data, without any changes to the underlying diffusion framework. By mapping categorical data to points in a circle, there still exist only discrete true states. From a theoretical standpoint, this implies that the score $\nabla_x \log p(x)$ is not properly defined and usually different approximations would need to be used, see, e.g., the suggestion in [2].
>
> Nevertheless, we implemented TabRep-Flow using the official implementation available at [https://github.com/jacobyhsi/TabRep](https://github.com/jacobyhsi/TabRep). This builds upon the TabSyn codebase, which we already successfully implemented. However, it appears that TabRep, maybe due to the novel categorical encoding, does not work well with our set of missingness indicators, which are treated as binary categorical features. The model massively overestimates the propensity of entries to be missing. This of course leads to a very subpar sample quality as it severely skews the data distribution of the non-missing entries considerably. Therefore, the average performance does not reach the performance of the other baselines at all. We illustrate this in the table below, which is a direct extension of the average results in Table 1 in the updated paper. TabRep clearly learns something, as the DCR Share is rather low, so the samples are quite close in data space. However, sample quality in terms of actual statistical similarity is subpar, in particular for categorical data. We decided not to include these results in the paper. We will reach out to the authors to sort out potential issues related to missingness indicators.
>
>
> |                    |   Ours (DT) |   TabRep |
> |:-------------------|------------:|---------:|
> | Detection Score    |       0.437 |    0.001 |
> | Shape              |       0.978 |    0.725 |
> | Shape (cat)        |       0.989 |    0.572 |
> | Shape (num)        |       0.974 |    0.882 |
> | WD (num)           |       0.007 |    0.314 |
> | JSD (cat)          |       0.025 |    0.448 |
> | Trend              |       0.969 |    0.578 |
> | Trend (mixed)      |       0.945 |    0.43  |
> | MLE                |       0.025 |    0.299 |
> | $\alpha$-Precision |       0.972 |    0.035 |
> | $\beta$-Recall     |       0.563 |    0.012 |
> | DCR Share          |       0.85  |    0.785 |
> | MIA Score          |       0.971 |    0.986 |
>
>
>
> - [1] Guzmán-Cordero, et al. Exponential Family Variational Flow Matching for Tabular Data Generation.
> - [2] Dieleman, et al. Continuous diffusion for categorical data.

---

> ### Author Response · Authors · 2025-11-21
>
> > What is the motivation of using flow matching in the framework?
>
> Flow matching is a very general framework. Depending on your modeling choices, this can include linear probability paths, which allow for fewer sampling steps and therefore increased sampling speed. We do not aim for increased sampling speed, instead our goal is to obtain more realistic samples. Hence, we are not using linear paths but "paths of least resistance" by letting the learnable $\gamma_t$ decide how we interpolate between samples from the two distributions. We provide the sampling times per 1000 samples in Appendix L for completeness but they are not the focus of our work.
>
> Instead, we chose the flow matching approach since we aim for changing the prior distribution based on the DT encoder. Flow matching allows for convenient changes to that distribution. This helps us conceptualize the idea formally and simplifies the exposition to the reader. In principle, the same idea could be transformed into an equivalent continuous-time diffusion or score matching model.
>
>
> > One of the most important use-cases of tabular relational data generation is privacy preservation. DCR experiments are conducted and demonstrates that it underperforms against its competitors.
>
> Due to the trade-off between accuracy and privacy, these results are expected. We are proposing a more powerful generative model that explicitly focuses on improving the details of the learned numerical distributions by additional conditioning information. This necessarily leads to a lower DCR share. In Section 5.1 on evaluation metrics, we now added clarifications that we, like all previous work, do not directly target privacy. Privacy considerations are highly context-dependent and may rely, e.g., on differential privacy or post-processing techniques. These techniques are general and can also be applied to our model. Since none of the benchmark models take privacy into account, accounting for it in our model would not yield a fair comparison.
>
>
> > Additionally, existing literature in tabular data generation [1] and computational privacy [2] [3] have also highlighted the inadequacy of DCR in evaluating privacy preservation. Hence, its important to assess the privacy preservation via Membership Inference Attacks too.
>
> Thank you for these references. Our focus is not to design a private model but a model that is able to generate mixed-type features with high accuracy. Thank you also for your comments on DCR. First, please note that we are not reporting DCR but DCR share, which is relative to a holdout set and therefore much more meaningful than the raw DCR. To accommodate your comment, we reframed DCR share as diversity rather than a privacy metric. Second, we added Membership Inference Attack (MIA) scores to our main results in Table 1 for completeness. It can be seen that the decrease in privacy as indicated by the MIA score is marginal. Similar to all benchmark models, we do not make any privacy claims or guarantees.
>
> ---
> We thank you for all your comments, and we believe that we have improved the paper by making the various changes mentioned above. We hope that we sufficiently answered your questions and you are willing to update your scores for this paper. Please let us know in case some points are still unclear, or if further changes are needed.

---

### Official Review · Reviewer_SS8m · 2025-11-01

**Soundness:** 2
**Presentation:** 3
**Contribution:** 2
**Rating:** 2
**Confidence:** 2

**Summary:**

The paper proposes TabCascade, a cascaded flow matching framework for heterogeneous tabular data with mixed type features. Low resolution information, that is categorical variables and a discretized view of numeric variables, is generated first, and a high resolution conditional flow matching model then fills in continuous numeric details. The high resolution model uses a guided conditional probability path with feature specific time schedules and a data dependent Gaussian source distribution, together with a theorem showing that the tree based encoder can lower a transport cost bound. Experiments on six public datasets show strong detection scores and competitive Shape and Trend metrics, with an ablation over decision tree depth. However, the study does not include direct flow matching baselines, so it is hard to tell how much of the reported gains come from the cascade versus simply switching from diffusion to flow matching.

**Strengths:**

### **Strengths**

- Clear and practical framing for mixed type numeric features, with an ancestral sampling procedure that decides coarse states first and fills in numeric details when needed (eq 2.)
- Guided conditional probability path with feature specific time schedules and a compact supervised target for the velocity field (eq 5 and 6).
- Theorem showing reduced transport cost under the tree encoder, which gives non trivial support for the chosen coupling, see Appendix A.
- Consistent improvements on detection scores and competitive Shape and Trend across six datasets, with a simple depth ablation.

**Weaknesses:**

### **Weaknesses**

- Too many changes at once, limited component ablation. The system introduces several factors at the same time, switch from diffusion to flow matching, data dependent source with mean and variance from z, learned feature specific time schedules, and the cascaded split with a strong low resolution model. The study does not sufficiently isolate the contribution of each factor. The current ablation varies only the depth of the tree encoder, which mainly changes the difficulty of the high resolution stage rather than the learning rule itself.
- Missing flow matching baselines. The core high resolution component is a flow matching model, but Section 5 compares against diffusion models and non diffusion models, not against direct flow matching baselines. At minimum, include a straight flow matching baseline for tabular data with a linear path and a standard normal source. In addition, please compare to recent flow matching baselines suggested by the community (eg [1, 2]), or explain why they are not compared against.
- Encoder dependence and leakage of difficulty. With deeper trees, integer valued features can be fully captured at low resolution, effectively removing them from the high resolution task, which can inflate joint realism without demonstrating stronger continuous modeling. A per dataset analysis of how often the high resolution stage is masked would help.


### References

- [1] **Exponential Family Variational Flow Matching for Tabular Data Generation** - Andrés Guzmán-Cordero, Floor Eijkelboom, Jan-Willem van de Meent
- [2] **Generating and Imputing Tabular Data via Diffusion and Flow-based Gradient-Boosted Trees** - Alexia Jolicoeur-Martineau, Kilian Fatras, Tal Kachman

**Questions:**

### **Questions**

- Add a direct flow matching baseline for tabular data with a linear path and a standard normal source, and also a rectified flow baseline, both under the same architecture and training budget as your high resolution model, then report the main metrics in Tables 1 to 3 to separate the effect of the cascade from the effect of the objective. Ideally also compare against existing FM approaches.
- Provide component wise ablations that keep the encoder fixed and switch off each ingredient in turn, no learned time schedules, set gamma t equal to t, no data dependent source, set mu equal to zero and sigma equal to the identity, keep the cascade but replace the high resolution flow matching with diffusion as in CDTD with the same model size and budget, and keep the high resolution flow matching but remove the cascaded conditioning to test unconditional flow matching, then report all three main metrics.
- Report, for each dataset, the fraction of features and rows for which the high resolution model is masked by z, and how this fraction changes with encoder depth, to calibrate how much work is delegated to the low resolution stage.

---

> ### Author Response · Authors · 2025-11-21
>
> > Strengths: framing, guided conditional probability paths, theoretical result, and performance improvements
>
> Thank you for your positive evaluation on these important components of our paper.
>
> > Missing flow matching baselines.
>
> Thank you for the suggestions for other benchmarks in the flow matching domain. We will discuss these suggestions in turn,
>
> About [1]: We tried to include it as an additional benchmark. While training works well, the sampling code that is available on GitHub is faulty. Without any changes, sampling is simply not possible. With minor changes to the code, the sampling procedure seems to work but the sample quality is abysmal. These problems are already documented in GitHub issues by other users, see [https://github.com/andresguzco/ef-vfm/issues/1](https://github.com/andresguzco/ef-vfm/issues/1) and [https://github.com/andresguzco/ef-vfm/issues/2](https://github.com/andresguzco/ef-vfm/issues/2). Therefore, we can, unfortunately, currently not add [1] to the paper. We contacted the authors about solving the issues with their available code.
>
> You also suggested [2] as an additional benchmark. However, ForestDiffusion is prohibitively expensive for all but very small tabular datasets (the median of the number of observations in their datasets is only 540). Even for these tiny datasets, they required 10-20 CPUs with 64-256 GB of memory (see Appendix B.3 in [2]). With their suggested hyperparameters, a single training run on the adult dataset failed to complete after training for multiple hours, which far exceeds the training budget of all other models *combined*. Similar observations have been made in the CDTD paper (see footnote on page 7 in [3]). Note that ForestDiffusion requires the estimation of $D \cdot T$ different models, where $D$ is the number of features and $T$ the number of diffusion time steps. For the adult dataset this makes 750 distinct models under default hyperparameters. We guess this might be the reason why none of the cited reference papers benchmark against ForestDiffusion [2].
>
>
> - [1] Exponential Family Variational Flow Matching for Tabular Data Generation - Andrés Guzmán-Cordero, Floor Eijkelboom, Jan-Willem van de Meent
> - [2] Generating and Imputing Tabular Data via Diffusion and Flow-based Gradient-Boosted Trees - Alexia Jolicoeur-Martineau, Kilian Fatras, Tal Kachman
> - [3] Continuous Diffusion for Mixed-Type Tabular Data - Markus Mueller, Kathrin Gruber, Dennis Fok
>
> > Too many changes at once, limited component ablation
>
> This also relates to one of your questions. See below, where we answer these questions.
>
> >  The core high resolution component is a flow matching model, but Section 5 compares against diffusion models and non diffusion models, not against direct flow matching baselines.
>
> Flow matching and diffusion models are simply a different perspective on the same model class. For this argument, please see, e.g., page 31 in [Principles of Diffusion Models](https://www.arxiv.org/abs/2510.21890). As such, one model formulation could be translated into another, with specific design choices. We chose the flow matching framework for ease of exposition to the reader due to convenient notation when conceptualizing and changing the coupling mechanism.
>
> > Add a direct flow matching baseline for tabular data [...]
>
> We added a vanilla Flow Matching high-resolution model to the ablation experiments in Table 2. We also added a model that uses the CDTD model as the high-resolution model, instead of our proposed flow-matching variant. Yet another model introduces the cascade separately into the vanilla CDTD model to investigate the effect of the cascade alone. Overall, the ablation results indicate that simply introducing a cascade and latent variables is not enough. Our data-dependent coupling is crucial and allows us to reap the benefits of the latent variables.
>
>
> > With deeper trees, integer valued features can be fully captured at low resolution, effectively removing them from the high resolution task, which can inflate joint realism without demonstrating stronger continuous modeling. A per dataset analysis of how often the high resolution stage is masked would help.
>
> This can indeed happen. However, this should be seen as feature of our model and not a bug. You can interpret it as a data-dependent way of deciding whether an integer-valued feature should be treated as categorical or numerical. This still constitutes stronger continuous modeling if the metrics corresponding to numerical features improve.
>
> > Provide component wise ablations [...]
>
> Thank you for this important suggestion. We completely reworked the results and ablation sections. In Table 2, we now start from the vanilla CDTD model and gradually introduce components until we reach the TabCascade specification.

---

> ### Author Response · Authors · 2025-11-21
>
> > keep the high resolution flow matching but remove the cascaded conditioning to test unconditional flow matching
>
> Thank you for the suggestion, but we note that this is not feasible. In that case, we would model $p(x_{\text{num}})$ without any conditioning. This is not a valid factorization of the joint distribution $p(x_{\text{num}}, x_{\text{cat}})$. We therefore did not include this specification in the ablation experiments.
>
>
> > Report, for each dataset, the fraction of features and rows for which the high resolution model is masked by z, and how this fraction changes with encoder depth, to calibrate how much work is delegated to the low resolution stage.
>
> We added Figures 17 and 18 in the Appendix to display the proportion of masked inputs to the high resolution model as well as the average number of completely masked features. It is unclear to us what you are referring to with 'calibration'. Our encoder-based approach is a data-dependent process of delegating work between low and high resolution models.
>
>
> ---
>
> We thank you for all your comments, and we believe that we have improved the paper by making the various changes mentioned above. We hope that we sufficiently answered your questions and you are willing to update your scores for this paper. Please let us know in case some points are still unclear, or if further changes are needed.

---

### Official Review · Reviewer_5EfX · 2025-11-03

**Soundness:** 3
**Presentation:** 1
**Contribution:** 2
**Rating:** 2
**Confidence:** 4

**Summary:**

Mixed-type tabular data generation with cascaded flow matching so it condition on categorical features and latent continuous features.

"To the best of our knowledge, this is the first work to address mixed-type feature generation,
i.e., features following a mixture of categorical and continuous distributions, within diffusion-based
models." This is an overclaim. You mentioned yourself prior reference of work doing mixed-type generation. You don't need this claim for your work to be relevant. Okay, after reading more I get what you are trying to see, please rephrase it to talked about diffusion cascade because right now it looks like an overclaim.

Missing reference:
- https://arxiv.org/abs/2309.09968
handles mixed-type generation and missing data using xgboost
- https://openreview.net/forum?id=LFCSTy6MYe#discussion
uses kernel density integral quantization (KDI) which sounds quite similar to your use of distributional trees

Problem statement Inflated values: Why be limited to dirac (aka binary categorical feature)? There are multi-class categories. I'm not sure I get why this paragraph is needed.

"Previous diffusion models for tabular data can be trained on numerical features with missing values, but are
not designed to generate such instances." Nobody wants to generate data with missing values, its not useful. Then you have to discard those samples anyways if using something like linear regression or logistic classification (which is what people use for small data in medecine/psychology).

"he simplicity of learning categorical features": Can you show Figure 2 with other methods included, e.g., TabDDPM, ForestDiffusion? This fits the figure and would strengthen the argument. One thing though to keep in mind is that categorical data can be very hard to model properly, it could just that for this data, getting the category right is easy. In retrospect, to be honest, I dont buy that claim either that categorical features are easier, it really depends on the dataset. I really dont buy it, you need a stronger argument or more proof or to remove that paragraph. Maybe multiple datasets with multiple methods.

General comment:
- I feel like there is a lot of flafla text that could be trimmed down and a lot of unnecessary math equations that could be removed and replaced with a figure or one paragraph.
- its overcomplicated, make it more simple
- results sections need major rework

Can you explain why you need a fifth-degree polynomial for the time schedule. This seems extremely overengineering, like a simple linear line from 0 to 1 would work. And why do we need feature-specific path?

The coupling and factorization make sense. I can see how it could help produce better data. I like the idea of learn z from GMM or DT.

10% MNAR is extremely low. Real world data that data scientists deal with have 25-50% MNAR. Having an example with 25 or 50% would be important because its closer to the real world and will test the methods to their limit.

 SDMetrics shape and trend are not great metrics, they are not accounting for the whole distribution. The other metrics are good. But you need a distribution metric to really tackle distance in distribution. You can use something like the Wasserstein distance (see https://arxiv.org/abs/2309.09968). It's important to have such a metric. In my opinion it would be better to show your results as the average across datasets or the rank, this way you can have a single figure with all the metrics. You can leave the current tables to the appendix. Also right now it makes you look like you chose the best metrics to show in the paper and left the rest in appendix; from looking at the appendix, it sure looks that way. Make one table with all metrics in the paper, everything else is appendix info. Please also add a small table showing the N, N_cat_features, N_cont_features of each dataset included. I'm sorry to say this, but 6 datasets is not a lot with tabular data. I know that this is extra work, but ideally having more datasets would really help. When you switch to average or ranking, it will make it easy to add new datasets without getting tables that are too big.

The ablation is too small, you need a real ablation where components are removed including whether to go cascaded or not, GMM vs distributional trees vs Quantile-Transform, linear schedule versus your super complicated feature-dependent 5-degree polynomial, etc.

If the authors make the major changes requested, I'll revisit my score.

**Strengths:**

cascaded structure is promising

**Weaknesses:**

- There is a lot of flafla text that could be trimmed down and a lot of unnecessary math equations that could be removed and replaced with a figure or one paragraph.
- its overcomplicated for no reason, explain in a simpler way
- results sections need major rework
- showing only good numbers in the paper and leaving bad ones in the appendix
- lack of true ablation

See the "Summary"

**Questions:**

Can you explain why you need a fifth-degree polynomial for the time schedule. This seems extremely overengineering, like a simple linear line from 0 to 1 would work. And why do we need feature-specific path?

---

> ### Author Response · Authors · 2025-11-21
>
> > "To the best of our knowledge, this is the first work to address mixed-type feature generation, i.e., features following a mixture of categorical and continuous distributions, within diffusion-based models." This is an overclaim. You mentioned yourself prior reference of work doing mixed-type generation.
>
> We believe that this comment is due to confusion between mixed-type features and mixed-type data. We believe that we are not "overclaiming": Mixed-type data, i.e., a dataset where some features are categorical and some are numerical, differs from mixed-type features, i.e., a single feature whose distribution is a mixture of a categorical and a continuous distribution. We tried to avoid this confusion by expanding upon this point in the problem statement and adding a new Figure 1 for explanation. In sum: yes, there exist models that generate mixed-type tabular *data* but none of them can explicitly accommodate mixed-type *features*. TabCascade is, to the best of our knowledge, the first diffusion model that considers that.
>
>
> > [1] handles mixed-type generation and missing data using xgboost
>
> Thank you for pointing us towards this paper. Another reviewer made the same suggestion. However, ForestDiffusion is prohibitively expensive for all but very small tabular datasets (the median of the number of observations in their datasets is only 540). Even for these tiny datasets, they required 10-20 CPUs with 64-256 GB of memory (see Appendix B.3 in [1]). With their suggested hyperparameters, a single training run on the adult dataset failed to complete after training for multiple hours, which far exceeds the training budget of all other models *combined*. Similar observations have been made in the CDTD paper (see footnote on page 7 in [2]). Note that ForestDiffusion requires the estimation of $D \cdot T$ different models, where $D$ is the number of features and $T$ the number of diffusion time steps. For the adult dataset this makes 750 distinct models under default hyperparameters. We guess this might be the reason why none of the cited reference papers benchmark against ForestDiffusion.
>
>
> - [1] Generating and Imputing Tabular Data via Diffusion and Flow-based Gradient-Boosted Trees - Alexia Jolicoeur-Martineau, Kilian Fatras, Tal Kachman
> - [2] Continuous Diffusion for Mixed-Type Tabular Data - Markus Mueller, Kathrin Gruber, Dennis Fok
>
>
> > [...] uses kernel density integral quantization (KDI) which sounds quite similar to your use of distributional trees
>
> Thank you for bringing KDI to our attention. We implemented it and it is indeed able to identify inflated values. However, KDI only identifies a very limited number of components, and this translates into increased transport costs in our model relative to the distributional regression tree [DT] that we propose. Also, unlike Gaussian Mixture Models [GMM] and DT, KDI does not directly fit Gaussian components, which we need for our data-dependent coupling. We therefore decided to not add KDI as another encoder. Since KDI leads to an even lower number of components than GMM, the performance of TabCascade using the KDI encoder will be even lower.
>
> > Problem statement Inflated values: Why be limited to dirac (aka binary categorical feature)? There are multi-class categories. I'm not sure I get why this paragraph is needed.
>
> Indeed, there could be multiple inflated values that could be treated as separate discrete classes, each with formalized by a different Dirac delta distribution. We added a sentence to clarify the generalizability of our setup. This paragraph is needed to motivate what a mixed-type feature distribution looks like.
>
> > Nobody wants to generate data with missing values, its not useful. Then you have to discard those samples anyways if using something like linear regression or logistic classification (which is what people use for small data in medecine/psychology).
>
> We disagree with this statement. In general it is valuable to be able to synthesize datasets as they appear in practice, so including missing data, inflated zeros, and other variables with combinations of continuous and discrete features. We already give some examples in the paper of missing values that convey special meaning and may need to be analyzed, e.g., by social scientists or medical professionals. Typically, practitioners would need to model missing values explicitly to extract their meaning, for a classical example, please see the popular Tobit model [https://en.wikipedia.org/wiki/Tobit_model](https://en.wikipedia.org/wiki/Tobit_model). Finally, missing values need not be discarded in statistical analysis by default. Our generated data (including missings) could provide datasets useful for the development and testing of imputation methods. This is only possible if missing values can actually be synthesized.

---

> ### Author Response · Authors · 2025-11-21
>
> > Can you show Figure 2 with other methods included?
> > [...] I dont buy that claim either that categorical features are easier, it really depends on the dataset.
>
> Thank you for this suggestion! We do believe that we document a general pattern here, rather than a result that only holds for a single case. To document this, we changed Figure 2. It now shows the average over all diffusion-based models and all datasets, and the average performance of TabCascade across all datasets. We report the results on each diffusion model and each dataset in Figure 16 in the Appendix. Of course, as always, model performance depends on the dataset. However, Figure 16 in the Appendix shows that our claim is a common pattern across tabular datasets.
>
>
> > There is a lot of flafla text that could be trimmed down and a lot of unnecessary math equations that could be removed and replaced with a figure or one paragraph.
>
> We respectfully, but strongly disagree. In designing a generative model, we have to be precise. This requires mathematical notation. Where necessary, we provided intuitive explanations in plain English. All used notation is necessary to properly define the generative model from scratch and make the entire method reproducible. If you feel differently, please point us towards paragraphs that you deem unnecessarily complex or overly math-heavy so that we can alleviate the issue.
>
>
> > its overcomplicated for no reason, explain in a simpler way
>
> Could you be more specific about which parts you deem overcomplicated? Other reviewers did not raise this issue, and some complimented us on the expositional clarity.  We carefully chose our introduced notation and (in our view) did not add unnecessary complexity. Our text includes intuitive explanations of our design elements in addition to the formal definitions. We would be happy to further improve the writing if needed.
>
>
> > Can you explain why you need a fifth-degree polynomial for the time schedule. This seems extremely overengineering, like a simple linear line from 0 to 1 would work.
>
> We require a function that interpolates between $\gamma_0=0$ and $\gamma_1=1$ in a very *flexible* way and we require closed-form access to its time-derivative $\dot{\gamma}_t$. We chose a fifth-degree polynomial because [1] used it in a different context for interpolation and achieved good results. Note that a linear line would not work, as fixing the two points $\gamma_0=0$ and $\gamma_1=1$ already fully defines the line. Hence, there would be nothing left for the model to learn and adjust.
>
> - [1] Subham Sekhar Sahoo, Aaron Gokaslan, Chris De Sa, and Volodymyr Kuleshov. Diffusion Models With Learned Adaptive Noise.
>
>
> > And why do we need feature-specific path?
>
> This is required because features in tabular data are highly heterogeneous. There is no reason to believe that a single, common path would perform well for *all* features. Therefore, we define *feature-specific* paths. Note, however, that our approach is strictly more flexible and includes the possibility of all features following the same path (and possibly a linear one as well). However, Appendix H illustrates that there *does* exist variance in the optimal paths across features.
>
> > The coupling and factorization make sense. I can see how it could help produce better data. I like the idea of learn z from GMM or DT.
>
> Thank you for this positive comment on some of the core ideas in our paper!
>
>
> > 10% MNAR is extremely low. Real world data that data scientists deal with have 25-50% MNAR. Having an example with 25 or 50\% would be important because its closer to the real world and will test the methods to their limit.
>
> We agree that the missingness rate can vary in practice, depending on the context, field, etc. Typical examples include economic survey data or psychological questionnaires. In many contexts, a 10\% missingness rate would already be considered very high. However, we do agree that more extreme scenarios should be tested. We add a comparison of TabCascade to TabDiff, which performs very well in the main results, on missingness rates of 10%, 25% and 50% (see Table 3). The performance gap of TabCascade remains consistent.

---

> ### Author Response · Authors · 2025-11-21
>
> > SDMetrics shape and trend are not great metrics, they are not accounting for the whole distribution. The other metrics are good. But you need a distribution metric to really tackle distance in distribution.
>
> We agree that Shape and Trend are not the best metrics. However, they are still the most popular ones by far! Note also that Shape and Trend metrics are partially built around the Total Variation Distance, which does take into account the full univariate, categorical distributions. Furthermore, the Wasserstein distance is only meaningful for numerical features, as it requires a meaningful distance between data points. We now added Wasserstein Distance and Jensen-Shannon divergence to our main results. On both, TabCascade performs best or very competitively (see Table 1). We also added Shape scores for numerical and categorical features only. We hope that these additional results convince you that TabCascade performs well on a wide range of metrics.
>
>
> > In my opinion it would be better to show your results as the average across datasets or the rank,
>
> Thank you for this suggestion. We added a new Table 1 to the paper to show these averages and moved the detailed results to the appendix. Table 1 gives a holistic overview of the models' performance across metrics. The conclusion stays the same: TabCascade has the best overall performance.
>
>
> > Please also add a small table showing the N, N_cat_features, N_cont_features of each dataset included.
>
> This was already included, please see Appendix B.
>
>
> > [...] but 6 datasets is not a lot with tabular data.
>
> Of course, there are always more datasets we could add. However, our experimental setup requires extensive compute as we need to train each model three times on the same dataset. This is unlike previous papers, as they do not generate missing values. Due to this much larger computational burden we cannot add an unlimited number of datasets. Further, previous work used a comparable number of datasets. For instance,
>
> - TabDDPM [2] uses 9 mixed-type datasets
> - TabSyn [3] uses 6
> - TabDiff [4] uses 7
> - Tabbyflow [5] uses 6
>
> Hence, we believe that the current choice of the most popular datasets used in previous papers gives a good indication of our model's capabilities.
>
> - [2] Kotelnikov, et al. TabDDPM: Modelling Tabular Data with Diffusion Models.
> - [3] Zhang, et al. Mixed-Type Tabular Data Synthesis with Score-based Diffusion in Latent Space.
> - [4] Shi, et al. TabDiff: a Mixed-type Diffusion Model for Tabular Data Generation.
> - [5] Guzmán-Cordero, et al. Exponential Family Variational Flow Matching for Tabular Data Generation.
>
> > lack of true ablation
>
> We agree that the ablation study was a bit limited. We now add an extensive ablation study, also considering the requests of the other reviewers. It gives insight into every single component we proposed, also in comparison to the vanilla CDTD and Flow Matching approaches.
>
> ----
> We thank you for all your comments, and we believe that we have improved the paper by making the various changes mentioned above. We hope that you are willing to update your scores for this paper. Please let us know in case some points are still unclear, or if further changes are needed.

---

> > ### Comment · Reviewer_5EfX · 2025-11-21
> >
> > I would still remove that claim or significantly rephrase it. Feature and Variables means the same thing. Maybe say 'mixed-type latent feature'. I would personally just say that you propose this, not claim it as the first to ever do it since you never know if others have not done something like this before. But I'm fine if you insist on the claim as long as its rephrased. Its not good enough yet, a feature is 1 variable. You need to clarify more since now your "feature" is 1 latent numeric feature and 1 categorical feature. And its a conditional distribution, I'm not a fan of the framing. Why not call it a novel conditional parametrization or something like that?
> >
> > I'm fine with the limitations of tree-based methods, its not the end of the world if you dont compare to it. I'm just saying that its important to mention it in your discussion about handling mixed-type data since the handling of mixed data is a big part of how you sell your approach in the intro and its an alternative approach of doing it.
> >
> > About KDI, showing that KDI is worse is good, its an interesting ablation and put things in context.
> >
> > You really dont want missing data in your datasets. Its a bit problem because then you are to get rid of samples with missing variables to do linear regression or use multiple imputation. Otherwise you are stuck with tree-based methods that automatically split missing values or neural networks with generally single imputation (which leads to incorrect prediction confidence). The only case I can think of would be censoring to emulate when someone dies and you cant get their prediction, but even then idk, it seems not ideal either.
> > "Finally, missing values need not be discarded in statistical analysis by default." As a former data scientist, yes, if you use linear regression you will have to discard it unless you use multiple imputation which blows up the inference cost. I completely disagree that generating missing data could be useful. I think that if someone wanted to do it anyways, they cold just  add a NA category for categorical data and put a 0 for continuous missing data with an extra indicator function 0/1 to whether they are missing. So any generative model that dont handle NA, can still generate NA if its truly desired. But right now the fact that you are forced to have NAs being generated is a big problem for most usecases.
> >
> > Regarding missing data, I worked with humans, so often they miss one test or questionnaire at some point, its inevitable that you end up with a lot of missing data.
> >
> > I dont understand why linear schedule doesn't work. I have trained plenty flow-matching models with plain old uniform[0,1] linear schedule. The whole point of flow-matching in the first place from my pov was to get rid of the complex schedules coming from the SDE. Have something simpler but good. Why always make things so complicated for no reason.
> >
> > Popular metrics are not better and yes its distribution-based but 1d! Thank you for adding those.
> >
> > Training your model 3 times should be cheap. Its small tabular data. If its too slow, just add like 10 small datasets with 1k or less observations. And its important, because I expect your model to be able to handle small data.
> >
> > I'm sorry, I know that you have provided efforts in trying to improve the paper, but not enough. You have not succeeded in convincing me. Most importantly, I dont think that this approach will a high impact considering the NAs generation and its overcomplicated for no reason. I will not change my score.

---

> > > ### Author Response · Authors · 2025-11-24
> > >
> > > Thank you for the additional comments and quick response. We believe that for your two main concerns there mainly seems to be some misunderstanding. Perhaps we have not been clear enough in the paper or in our earlier reply. Below we first consider these two points. Next we will briefly comment on your other remaining concerns.
> > >
> > > ## Mixed-type data vs mixed-type features
> > >
> > > Current approaches for tabular data generation in the literature (at most) allow for two types of variables: categorical and continuous. The combination of the two in a single dataset is referred to as mixed-type **data**. However, in practice one often has variables that are both at the same time: they follow a continuous distribution on a part of the domain, but also have discrete outcomes with a positive probability mass. We (and the literature) label such features as mixed-type **features** (but we are happy to adopt a different naming convention if you have one in mind).
> > >
> > > Examples are features with missing values (there is a non-zero probability for the outcome "missing") or features with inflated zeros, e.g., individual level expenditure on a certain product category (there is a non-zero probability for the outcome exactly equal to 0 mixed with a continuous distribution for the strictly positive outcomes).
> > >
> > > Such features can of course not be treated as categorical (as they are continuous on a part of the domain), but treating them as continuous in SOTA methods will lead to poor generative performance (we show this in our paper). The relatively poor performance remains even when the misssings are imputed, and a separate "missingness indicator" is constructed. These methods are simply not designed to generate the discrete outcomes with a sufficiently large probability and capture all the dependencies across variables. TabCascade explicitly does allow for these mixed-type features.
> > >
> > > With this important distinction in mind we do still believe that we are the first to consider mixed-type **features** in diffusion models for tabular data generation. In our paper we have now toned down this claim a bit by adding "to the best of our knowledge".

---

> > > ### Author Response · Authors · 2025-11-24
> > >
> > > ## Usefulness of generating missings
> > >
> > > We agree that the presence of missing values is often a nuisance in applied work. In this sense, generating data with missings may seem to be meaningless. However, first note that missing values is just one example of mixed-type features. Our method also allows for all other cases of mixed-type features. In practice one will often observe such features. We already briefly mentioned an example above.
> > >
> > > Also, missing data cannot be discarded at all times. As another example, consider the case of intermittently missing data. That is, you observe a time series, e.g., for patients, but some data at certain time steps is missing. Such instances should be explicitly modeled and may be informative about the previous or future part of the time-series [1]. For instance, if the propensity for a missing at time $t$ is higher for severely sick patients, then a missing entry may indicate a downturn in patient health in the subsequent period for $s > t$. In the general case, we refer to the following statement:  "When the missing data mechanism is non-ignorable, inferences based on only the observed data will not be valid." [2]. See also [3]. Lastly, you may need to model missings separately when doing causal inference in observational data [4], which, as you agreed in your comment, comes often with missings.
> > >
> > > - [1] Joint Modeling of Longitudinal and Survival Data with Missing and Left-Censored Time-Varying Covariates. https://pmc.ncbi.nlm.nih.gov/articles/PMC4189992/
> > > - [2] A Bayesian model for longitudinal count data with non-ignorable dropout. https://pmc.ncbi.nlm.nih.gov/articles/PMC2975948/
> > > - [3] Statistical Analysis with Missing Data. Roderick J. A. Little, Donald B. Rubin
> > > - [4]  Causal inference with confounders missing not at random. https://academic.oup.com/biomet/article/106/4/875/5573228
> > >
> > > Next, we believe that the ability to generate data with the missingness pattern as observed in original data is also useful by itself. Such generated data allows for replication of research with sensitive data where the specific choices for processing the missings are also important. Another added value is testing methods that "correct for" missings. As you mention yourself, there are different choices to be made in practice when it comes to dealing with missing values and a proper evaluation of the benefits or costs of such choices requires relevant data.
> > >
> > > Your example of missings in questionnaires enforces this point. Missings are indeed very common in such data. If you want to model these missing values as part of a statistical model and you want to train that model on synthetic data, *then your generative model must be able to generate realistic missing values*. For example, say you want to develop a novel imputation method for banking data. Such data is highly sensitive, so you will not be allowed to use the real data for development. In such a case, synthetic data samples could be used. However, since you want to develop an imputation model, that synthetic data actually *needs to include missing values*.
> > > Our results show that TabCascade generates such missing values better than previous models. We may disagree on how valuable this use case is, but hopefully you agree that this purpose may be of value to some other researchers.
> > >
> > > We want to stress again that missings are just one *example* of a mixed-type feature and therewith of a single use case. If you want, you can completely disregard our claim about the usefulness of missing values. In that case, just consider the missing values we introduce as actual, non-missing observations of some arbitrary inflated points in the data space. In this case, our results show that our cascaded approach still outperforms the baselines in general sample quality.

---

> ### Author Response · Authors · 2025-11-24
>
> ## Other comments
>
> > Mixed-type latent features
>
> Perhaps we have already answered this through our replies above. Note that there are no mixed-type latent features in TabCascade. The introduced latent variables are all categorical.
>
> > I'm just saying that its important to mention it in your discussion about handling mixed-type data since the handling of mixed data is a big part of how you sell your approach in the intro and its an alternative approach of doing it.
>
> We agree and added a footnote on page 7. We will add the reference to the related work section as well.
>
> > (About an alternative way of encoding missings):
> > I think that if someone wanted to do it anyways, they cold just add a NA category for categorical data and put a 0 for continuous missing data with an extra indicator function 0/1 to whether they are missing.
>
> Yes, they can. That is how we endow all baselines with the ability to generate missings. **But this is not an equivalent specification!** In this case, the model does *not* know that a certain missingness indicator belongs to a certain feature. Our model, however, *does* know and therefore can generate missing values better and more efficiently, which is the whole point of the paper!
>
> > Training your model 3 times should be cheap.
>
> Training is cheap, the computational burden is the evaluation. The evaluation is called $3 \cdot 10$ times and takes a considerable amount of time (> 1 hours) on some complex datasets due to metrics relying on nearest neighbor computations.
>
> > If its too slow, just add like 10 small datasets with 1k or less observations.
>
> We hope that you will agree that benchmarking on tiny datasets with multi-million parameter models makes little sense. In practice, tabular datasets are often much larger in size than the ones we considered here [5]. Considering many small datasets therefore has little to no additional value. If you are willing to increase your score accordingly, we will add a few more (but larger) datasets in the upcoming week.
>
> - [5] Analyzing Pitfalls and Filling the Gaps in Tabular Deep Learning Benchmarks.
>
> > I dont understand why linear schedule doesn't work. I have trained plenty flow-matching models with plain old uniform[0,1] linear schedule. The whole point of flow-matching in the first place from my pov was to get rid of the complex schedules coming from the SDE. Have something simpler but good. Why always make things so complicated for no reason.
>
> We hoped that our previous explanations would already have explained this. Let us try again. A linear schedule is *fixed* by design, that is, the model cannot adjust it at all. It does work, as we show in the ablation in Table 3. This matches your personal experiences with Flow Matching [FM] models. However, our model, with a learnable, non-linear schedule works better. This is shown in the same table. The point of FM is to simplify *and* generalize. This generalization makes it easier to change different design elements, for a motivation see the excellent talk by the FM author on [YouTube](https://www.youtube.com/watch?v=5ZSwYogAxYg&t=3347s). **We insist that we do not make things complicated for no reason**, we gave multiple, valid reasons why learnable schedules are important for *heterogeneous* tabular data and the results show that this is indeed the case.
>
> ---
>
> We again hope that our replies above are convincing. It would be great if you could update your score accordingly, or let us know what *specific* changes would be required for you to do so. In case you are not yet convinced, please let us know your argumentation. **We hope that you will not base your evaluation on your belief about the usefulness of missing data generation.** Again, this is only *one* use case of our model.

---

### Author Response · Authors · 2025-11-21
**Summary of responses**

We thank all reviewers for their invested time and their valuable comments.

Reviewers indicated to like
- the idea of adding latents to the generative process, the cascaded structure and the proposed data-dependent coupling (reviewer 5EfX),
- the clear and practical framing for mixed type numeric features (reviewer SS8m) as well as the Theorem showing reduced transport cost (reviewers SS8m and oVg1),
- the novel idea, results (reviewers 5gde and oVg1) and overall clarity (reviewer 5gde).

In response to the reviewer comments, we made the following changes to the updated paper (changed text is highlighted in blue in the pdf):

- We completely reworked those sections (sections 5.2 and 5.3) in the updated paper.
- All main results are now aggregated as averages in Table 1.
- We added multiple new metrics, including Wasserstein distance, Jensen-Shannon divergence and membership inference attack scores.
- Multiple, component-wise ablation experiments have been added in Table 2.
- We conducted additional ablation experiments that examine the impact of an increased missingness rate, reported in Table 3.
- We added a new Figure 1 to better illustrate the overall framework and clarify the concept of a mixed-type feature distribution (versus a mixed-type data distribution).
- We added new figures (Figure 17 and 18 in the Appendix) to illustrate the proportion of masked entries in the inputs to the high-resolution model.

We hope these additions address the concerns raised by the reviewers.

---

### Meta-Review · Area_Chair_o12c · 2025-12-15

**Summary:**

Reviewers agree that the main idea (using a cascaded approach) is conceptually interesting and that the model shows good empirical performance. Reviewers also highlight Theorem 1 as an interesting theoretical contribution.

The main weaknesses raised by the reviewers are as follows:
- [W1] Insufficient comparison against existing baselines (5EfX, SS8m, 5gde, oVg1)
- [W2] Insufficient ablation studies (5EfX, SS8m)
- [W3] Unclear claims regarding the novelty and contributions of the work (5EfX, 5gde)
- [W4] Concerns about the overall empirical setup and significance of the results (5EfX, 5gde, oVg1)
- [W5] Concerns regarding the claimed privacy-preserving aspects of the approach (5gde, oVg1)

**Reviewer Concerns:**

- [W1] While the authors discuss the unsuitability of some suggested baselines, this point is still outstanding and not fully addressed.
- [W2] The authors add additional ablations to the submission in Section 5.2. However, a more careful study and discussion is likely needed to fully alleviate these concerns.
- [W3] The authors largely address this concern.
- [W4] This is largely unaddressed. While there is some discussion regarding this in the rebuttal, without significant changes to the empirical results, reviewers seem unlikely to update their beliefs on this point.
- [W5] While this work is not focused on privacy, the authors still conduct experiments on this, and the discussion does not seem to fully alleviate the reviewer's comments.

**Reviewer Scores:**

- 5EfX states explicitly their unwillingness to change their score.
- RSS8m may have increased their score by 1-2 points given the additional ablations and vanilla FM baseline.
- 5gde may have increased their score by 1-2 points given the discussion and additional results.
- oVg1 participated in the discussion and seems unlikely to increase their score.

Overall, there is a general consensus from the reviewers that this paper is interesting and has the potential to be a strong contribution, but is not ready for publication in its current form. I highly encourage the authors to revise their work according to this feedback and resubmit at a later date.

---

### Decision · Program_Chairs · 2026-01-26

Reject